# A cofactor-induced repressive type of transcription factor condensation can be induced by synthetic peptides to suppress tumorigenesis

Yang Tang[1,7], Fan Chen [2,7], Gemin Fang [3,7], Hui Zhang[4], Yanni Zhang[3], Hanying Zhu[3], Xinru Zhang[1], Yi Han[1], Zhifa Cao[1], Fenghua Guo[4], Wenjia Wang[5], Dan Ye [5], Junyi Ju [1], Lijie Tan[5], Chuanchuan Li[5], Yun Zhao [2], Zhaocai Zhou [5,6✉], Liwei An [1✉] & Shi Jiao [5✉]

## Abstract

**Transcriptional factors (TFs) act as key determinants of cell death and survival by differentially modulating gene expression. Here, we identified many TFs, including TEAD4, that form condensates in stressed cells. In contrast to YAP-induced transcription-activating condensates of TEAD4, we found that co-factors such as VGLL4 and RFXANK alternatively induced repressive TEAD4 condensates to trigger cell death upon glucose starvation. Focusing on VGLL4, we demonstrated that heterotypic interactions between TEAD4 and VGLL4 favor the oligomerization and assembly of large TEAD4 condensates with a nonclassical inhibitory function, i.e., causing DNA/chromatin to be aggregated and entangled, which eventually impede gene expression. Based on these findings, we engineered a peptide derived from the TEAD4-binding motif of VGLL4 to selectively induce TEAD4 repressive condensation. This "glue" peptide displayed a strong antitumor effect in genetic and xenograft mouse models of gastric cancer via inhibition of TEAD4-related gene transcription. This new type of repressive TF phase separation exemplifies how cofactors can orchestrate opposite functions of a given TF, and offers potential new antitumor strategies via artificial induction of repressive condensation.**

**Keywords** Repressive Condensation; TEAD4; VGLL Family of Proteins; Glue Peptide; Cancer
**Subject Categories** Cancer; Chromatin, Transcription & Genomics

## Introduction

Liquid–liquid phase separation (LLPS) have been broadly studied to explain the formation of protein condensates. For example, high degrees of aggregation and condensation of biomolecules are frequently participated in many conditions, such as degenerative diseases and multiple cancers (Molliex et al, 2015; Patel et al, 2015; Shin and Brangwynne, 2017). Multiple researchers have revealed the fundamental concept and aspects of LLPS (Bergeron-Sandoval et al, 2016; Hyman et al, 2014), biomolecular condensates (Banani et al, 2017) and their biological functions (Alberti and Hyman, 2021; Boeynaems et al, 2018; Lyon et al, 2021). Biomolecular condensates have been found in a wide area of subcellular locations as the nucleus (Sabari et al, 2020). Some nuclear condensates, which form from proteins binding to DNA or RNA, play critical roles in the maintenance of gene structure, chromatin folding, transcriptional activity and even proliferation signaling, and other activities (Lafontaine et al, 2021; Peng et al, 2020; Sabari, 2020). Currently, however, we do not explain the exact role of phase separation in tumorigenesis, a process closely associated with genetic aberrations that often dysregulate gene transcription (Bradner et al, 2017). Moreover, it still remains unknown whether different material properties of biomolecules functionally impact the gene expression and the biological effects in cancer.

Transcription regulation is essential for both cell survival and cell death (Galluzzi Vitale et al, 2018; Tang et al, 2019). Transcriptional activity has been increasingly evidenced to be spatiotemporally regulated by phase separation, a process in which multivalent interactions of multiple proteins and/or nucleic acids drive the formation of condensates (Boeynaems et al, 2018; Hnisz et al, 2017; Hyman et al, 2014; Shin and Brangwynne, 2017). For example, the condensation of proteins including RNA polymerase II (Pol II) (Kwon et al, 2013; Lu et al, 2018), transcription factors (TFs) (Boija et al, 2018; Chong et al, 2018) and coactivators (Sabari et al, 2018) could regulate gene transcription. TFs are defined as sequence-specific DNA-binding proteins regulating gene transcription. Many TFs can form multivalent

[1]Department of Medical Ultrasound, Department of Stomatology, Shanghai Tenth People's Hospital, Tongji University Cancer Center, Tongji University School of Medicine, Shanghai 200072, China. [2]State Key Laboratory of Cell Biology, Shanghai Institute of Biochemistry and Cell Biology, Center for Excellence in Molecular Cell Science, Chinese Academy of Sciences, University of Chinese Academy of Sciences, Shanghai 200031, China. [3]Institutes of Physical Science and Information Technology, Anhui University, Hefei 230601, China. [4]Department of General Surgery, Hua'shan Hospital, Fudan University Shanghai Medical College, Shanghai 200040, China. [5]State Key Laboratory of Genetic Engineering, School of Life Sciences, Zhongshan Hospital, Fudan University, Shanghai 200438, China. [6]Collaborative Innovation Center for Cancer Personalized Medicine, School of Public Health, Nanjing Medical University, Nanjing 211166, China. [7]These authors contributed equally: Yang Tang, Fan Chen, Gemin Fang. ✉E-mail: zhouzhaocai@fudan.edu.cn; lwan@tongji.edu.cn; jiaoshi@fudan.edu.cn

interactions with other TFs and/or cofactors, subsequently leading to the formation of condensates, most of which are thought to enhance TF activity by promoting a compartmentalized enrichment of TFs (Wagh et al, 2021). For example, Hippo pathway transcription factor TEAD4 is implicated in condensates formed by YAP/TAZ to promote gene transcription (Cai et al, 2019; Franklin and Guan, 2020; Lu et al, 2020; Wei et al, 2021; Yu et al, 2021). However, it remains unclear whether TFs can undergo repressive condensation, i.e., to lower their activity in some cases such as in cell death.

Glucose is the main energy source for cancer cells maintaining a rapid proliferation of cancer cells. Under the glucose deprivation, the growth of cancer cells is inhibited and regulated by associated TFs including c-Myc, p53 and related pathways. In this aspect, glucose starvation is emerging as an effective therapy to inhibit tumor growth. To develop a new way of combatting cancer, we are interested in figuring out whether a given TF can be switched on and off through cofactor-induced activating or repressive condensation in response to stimuli such as glucose starvation.

Regarding transcription cofactors, it is not unusual for a TF to have a family of binding proteins with similar interaction domain partners, yet with some of these partners possibly promoting but others repressing its activity. For example, VGLL1–4 proteins do not harbor any DNA-binding domain, but rather show their transcriptional regulatory roles through binding TEAD4 via their Tondu (TDU) domain(s). To date, most studies have identified VGLL1–3 as transcriptional co-activators (Faucheux et al, 2010; Gunther et al, 2004; Halperin et al, 2013; Maeda et al, 2002), whereas VGLL4 is defined as a transcriptional repressor inhibiting YAP-induced tumorigenesis (Guo et al, 2013; Jiao et al, 2017; Jiao et al, 2014; Koontz et al, 2013). Given the lack of apparent functional domains other than TDUs in the VGLL family of proteins, a daunting question is how these proteins achieve opposing functions via competing with YAP/TAZ for binding TEADs in a seemingly identical manner.

In this work, we performed a high throughput screening for TFs undergoing condensation in a context of limited glucose supply. Our results revealed condensation of TFs as a widespread phenomenon in cells challenged by glucose limitation. Of note, we identified TEAD4 as a pivotal player in chromatin deformation and thereby in orchestrating gene transcription for cell death. This process was found to occur via VGLL4-mediated oligomerization and repressive condensation of TEAD4. Compared to the YAP/TAZ-mediated transcriptionally activating TEAD4 condensates (Cai et al, 2019; Lu et al, 2020), we propose that transcriptional cofactors can switch on and off the activity of a TF by inducing activating or repressive condensation. We demonstrated that VGLL4 induces repressive condensation of TEAD4 against YAP-induced activation, thereby shutting down gene transcription and triggering cell death. Based on these findings, we further developed a linker peptide "glue" to force repressive condensation of TEAD4, which strongly reversed tumor progression.

# Results

## A group of TFs form condensates in cells upon glucose deprivation

Despite the emergence of glucose metabolic enzymes and transporter inhibitors, the efficiency of targeting tumor glucose metabolism is being challenged. It is well established that cancer cells heavily rely on glucose to overly proliferate, and the tumor microenvironment is constantly short of glucose supply. In fact, glucose starvation has been emerging as a therapy to suppress tumor growth. In this regard, we aimed to determine what happened to TFs in cells facing glucose deprivation. Thus, we performed a high-content fluorescent-spot-based screen in cells subjected to glucose deprivation using a panel of plasmids encoding 759 Flag- or Gal4-tagged human TFs (Fig. 1A and Dataset EV1). We calculated the fraction of condensed fluorescent spot in HEK293FT cells transfected with the individual TFs and deprived of glucose (Fig. 1B). Typically, the standard cutoff values for finding hits are z-scores with a +/−3-fold change. Setting the cut-off to >3-fold, 7 TFs—namely TEAD4, EWSH, RFXDC1, GTF2A1L, C16orf5, ZNF800, and ELF1—were identified as hits when using this screening (Fig. 1B), and were found to form more strongly fluorescent spots in response to glucose withdrawal compared to the untreated condition (Fig. 1B).

To determine whether these fluorescent spots were phase-separated condensates, we treated cells with 1,6-hexanediol (1,6-Hex), which inhibits weak hydrophobic protein-protein interactions required for LLPS-associated droplet formation (Duster et al, 2021; Ulianov et al, 2021) (Fig. 1C; Appendix Fig. S1A). Glucose deprivation clearly typically induced formation of bigger spots in cells transfected with TFs identified using the above screen (in Fig. 1B), indicating that these condensates were formed in a manner dependent on a cell-death-related stress signal (Fig. 1C; Appendix Fig. S1A). Notably, no obvious or much smaller spots were observed in 1,6-Hex-treated cells, suggesting that the TF condensates may have undergone a LLPS in glucose-deprived cells (Fig. 1C; Appendix Fig. S1A).

As TEAD4 has been well characterized to undergo LLPS to form condensates that stimulate transcription and cell growth with YAP/TAZ coactivators (Cai et al, 2019; Lu et al, 2020), it was intriguing to note such TEAD4 condensation in a stressed condition shutting off transcriptional activity. To further validate this observation, we established HEK293FT cells stably expressing TEAD4 fused at its C-terminus with GFP and reproducibly captured such TEAD4 condensation upon depriving the cells of glucose, which could be dramatically reversed by treating the cells with 1,6-Hex (Fig. 1D). Moreover, fluorescence recovery after photobleaching (FRAP) of the TEAD4 condensates yielded a τ value of about 5.9 s with a mobile fraction of about 71% (Fig. 1E), supporting the idea that glucose deprivation induces a highly dynamic and rapid fluorescence recovery of TEAD4 condensates. Also, when re-supplementing the glucose-deprived cells with glucose, a reduced quantity of TEAD4 condensates was observed (Appendix Fig. S1B), suggesting a reversibility of TEAD4 condensation. At last, we confirmed an endogenous formation of TEAD4 condensates in the glucose-deprived cells (Appendix Fig. S1C).

We then asked whether formation of fluidic TEAD4 condensates depends on YAP/TAZ, a transcriptional coactivator that has been reported to undergo phase separation to form condensates (Cai et al, 2019; Franklin and Guan, 2020; Yu et al, 2021). To address this issue, we compared the TEAD4 condensates in both wild-type (WT) and YAP-knockout (YAPKO) cells subjected or not subjected to glucose deprivation (Fig. 1F; Appendix Fig. S1D). We consistently observed larger TEAD4 condensates upon depriving WT cells of glucose, but no difference between TEAD4 condensates in YAPKO cells or YAPKO cells rescued by YAP regardless of glucose treatment (Fig. 1F), indicating no effect of YAP on glucose-deprivation-induced TEAD4 condensation. We also generated TAZ-knockout (TAZKO) cell lines, and found a lack of effect similar to that of YAP (Appendix Fig. S1E). Using an in vitro LLPS

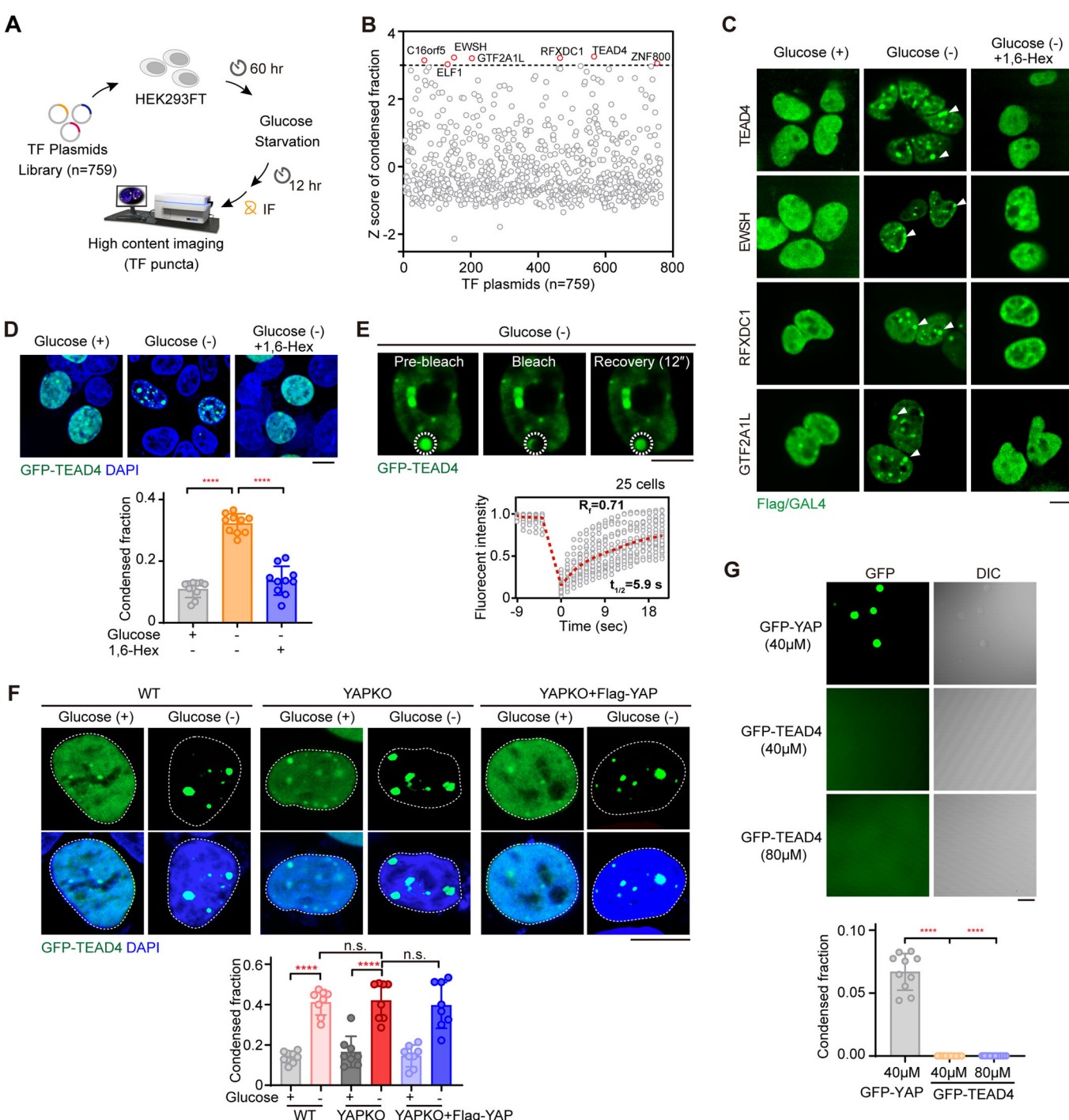

**5588** *The EMBO Journal* Volume 43 | Issue 22 | November 2024 | 5586 – 5612

assay, we observed the formation of YAP droplets (Fig. 1G), confirming previous studies (Cai et al, 2019; Franklin and Guan, 2020; Yu et al, 2021). However, the solutions containing purified TEAD4 protein at two tested concentrations remained clear during the imaging process (Fig. 1G), in keeping with the recent study showing that TEAD4 cannot form phase-separated condensates on its own (Yu et al, 2021) in vitro. Therefore, we speculated that condensation of TEAD4 in vivo seems to be triggered by its heterotypic interactions with other proteins.

## Identification of co-factors that induce repressive condensation of TEAD4

Since condensation of TEAD4 seems to be triggered by its heterotypic interactions with other proteins, we next deciphered the TEAD4 interactome to identify the possible regulators of glucose-deprivation-induced condensation of TEAD4. To this end, we performed a proximity-dependent biotin identification (BioID) assay, which utilizes the *E. coli* biotin ligase enzyme BirA, fused to

**Figure 1. Condensation of TFs is widespread upon glucose deprivation.**

(A) Schematic presentation of nuclear condensate screening. HEK293FT cells were transfected with a library of transcription factor (TF, $n = 759$) plasmids encoding Flag- or Gal4-tagged TF proteins (Dataset EV1). TF condensate was then identified by carrying out immunofluorescence (IF) staining for Flag/Gal4 antibodies after treating the cells with (+) or without (-) glucose for 12 h. Subsequently, using a high-content confocal microscopy and Columbus™ image data storage and analysis system, each cell was imaged and the areas and densities of condensates in the cells were analyzed to calculate the condensed fraction. (B) Z score analysis of the condensed fraction for each TF in HEK293FT cells upon upon their being starved of glucose. The cutoff value was set to 3 standard deviations. (C) Representative images of TF condensates in HEK293FT cells transfected with the indicated plasmids. After being deprived of glucose, HEK293FT cells were treated for 2 h with 1,6-hexanediol (1,6-Hex), a disruptor of condensation formation ($n = 3$). White arrowheads indicated nuclear condensates. 1,6-Hex, 0.25% v/v (same below). Scale bar, 10 μm. (D) Fluorescence images of GFP-TEAD4 condensates in glucose starvation-treated HEK293FT cells with or without 1,6-Hex. Representative images (upper) and quantification of TEAD4 condensed fraction (lower) are shown. The quantification graph represents the data collected from 10 cells ($n = 10$). Data shown as means ± SD represent the representative results from two independent experiments. The data were analyzed using one-way ANOVA, followed by the Tukey's post-hoc test. ****$p < 0.0001$. Scale bar, 10 μm. (E) FRAP analysis of TEAD4 condensates in HEK293FT cells upon their being deprived of glucose. White circles denote the photobleached spots and nucleoplasm (upper). Three images were taken, during pre-bleach, bleaching and fluorescence recovery (upper). The duration of each FRAP analysis experiment was about 20 s. For each photobleached spot, the fluorescence recovery curve was traced (lower), and this displayed graph represents the data collected from 25 cells expressing GFP-TEAD4 ($n = 25$). $t_{1/2}$: fluorescence recovery time; Rf: mobile fraction. Scale bar, 10 μm. (F) Fluorescence images of GFP-TEAD4 condensates in wild-type (WT) cells, YAP-knockout (YAPKO) cells, and YAPKO cells rescued with YAP, with or without glucose deprivation for 12 h. The quantification graph represents the data collected from 8 cells ($n = 8$). Quantification of the TEAD4 condensed fraction (bottom) is shown. Data shown as means ± SD represent the representative results from two independent experiments. The data were analyzed using one-way ANOVA, followed by the Tukey's post-hoc test. n.s., no significance; ****$p < 0.0001$. Scale bar, 10 μm. (G) Droplet formation assay, using differential interference microscopy (DIC), for purified GFP-tagged YAP or TEAD4 proteins. Quantification of condensed fraction is shown and collected from 10 figures ($n = 10$). Data shown as means ± SD represent the representative results from two independent experiments. The data were analyzed using one-way ANOVA, followed by the Tukey's post-hoc test. ****$p < 0.0001$. Scale bar, 10 μm. See also Appendix Fig. S1. Source data are available online for this figure.

the indicated protein, to label both stable or/and transiently associated proteins within ~10 nm (Roux et al, 2012). To identify the proximal TEAD4-binding proteins, a BirA-fused TEAD4 construct was generated and transfected into HEK293FT cells in triplicate. After addition of biotin, biotinylated proteins were isolated and analyzed using quantitative tandem mass spectrometry. A total of 192 proteins were identified as TEAD4-interacting proteins (Dataset EV2), of which the previously reported TEAD4-binding partners YAP (Vassilev et al, 2001; Wu et al, 2008; Zhang et al, 2008), TAZ (Lei et al, 2008), VGLL1 (Pobbati et al, 2012), VGLL4 (Jiao et al, 2014) and FAM181A (Bokhovchuk et al, 2020) were enriched most significantly (Fig. 2A, left).

To identify the key regulators of TEAD4 condensation, we designed an siRNA library targeting the identified 192 genes for a second round of validation. Knockdowns achieved using 20 individual siRNAs showed decreased TEAD4 condensation capacity (Fig. 2A,B), suggesting that these 20 genes might be involved in the regulation of TEAD4 condensation. Meanwhile, out of the 192 candidate genes, there were 7 hits whose knockdown increased the viability of HGC-27, a human gastric cancer (GC) cell line (Fig. 2A,B), suggesting that these genes might have antitumor functions. We then identified 7 genes, including those for VGLL4, ARID3B, RFXANK, YY1, CTCF, EMSY, and LDOC1, possibly responsible for both TEAD4 condensation and cell viability (Fig. 2C). Notably, VGLL4 was previously reported to interact with TEAD4 and restrain its transcriptional activity (Appendix Figs. S2A–C) (Jiao et al, 2014; Lin et al, 2016; Zhang et al, 2014), and LDOC1 was found to interact with TEAD4 in a yeast two-hybrid screen (Luck et al, 2020). Resembling the suppressive regulatory effects of VGLL4 (Jiao et al, 2014; Lin et al, 2016; Zhang et al, 2014), individual knockdowns of *LDOC1*, *EMSY* or *RFXANK* significantly enhanced the transcription of TEAD4 target genes *CTGF and CYR61* (Appendix Figs. S2A,B).

We then set out to determine whether VGLL4 or RFXANK per se acts as an inducer of TEAD4 condensation. As shown in Fig. 2D, transfection of either VGLL4 or RFXANK into HEK293FT cells strongly enhanced the ability of TEAD4 to form condensates, with this ability abrogated for cells treated with 1,6-Hex (Fig. 2D). Conversely, knockdown of VGLL4 or

RFXANK markedly decreased TEAD4 condensation capacity (Appendix Fig. S2D). Moreover, FRAP experiments on the VGLL4-induced condensation of TEAD4 yielded a τ value of about 4.19 s with a mobile fraction of about 77% (Fig. 2E), indicating a rapid fluorescence recovery of VGLL4-generated TEAD4 condensates.

Like TEAD4 proteins, VGLL4 protein on its own was found to be fully soluble and not form protein droplets in vitro (Appendix Fig. S2E). However, droplets rapidly formed and assembled upon mixing VGLL4 and TEAD4 proteins together (Fig. 2F). Notably, these droplets fusing from two smaller droplets increased in size, and reached their big diameter in <~2 h, whereas higher concentration of NaCl promoted the droplets dissolution (Fig. 2F). Resembling the case for VGLL4, in vitro droplet formation assay showed that RFXANK indeed induced TEAD4 to form condensates (Fig. 2G). Overexpression of VGLL4 not only inhibited the transcription of TEAD4 target gene *CTGF*, but also induced apoptosis (Fig. 2H,I), findings confirming previous studies (Jiao et al, 2017; Jiao et al, 2014; Zhang et al, 2014). Importantly, treating cells with 1,6-Hex abrogated the regulatory effects of VGLL4 on both *CTGF* expression and apoptosis (Fig. 2H,I). Similar observations were obtained for RFXANK overexpression (Appendix Figs. S2F,G), suggesting that VGLL4- or RFXANK-mediated TEAD4 condensation led to TEAD4-related transcriptional repression and cell apoptosis in vivo.

Taken together, these results indicate that VGLL4 and RFXANK can directly induce TEAD4 LLPS both in vitro and in vivo, and these condensates may function as repressors of transcription to eventually induce tumor cell apoptosis.

## VGLL4 triggers condensation of TEAD4 by inducing its oligomerization

The VGLL family of proteins contain TDU domains harboring a conserved "VXXHF" TEAD4-binding motif (Appendix Fig. S3A). Of these VGLLs, the death promoter VGLL4 contains two TDU domains while the growth promoter VGLL1 contains only one TUD domain (Appendix Fig. S3A). To further dissect the mechanism of VGLL4-mediated TEAD4 phase separation, we re-inspected the previously determined crystal structure of the

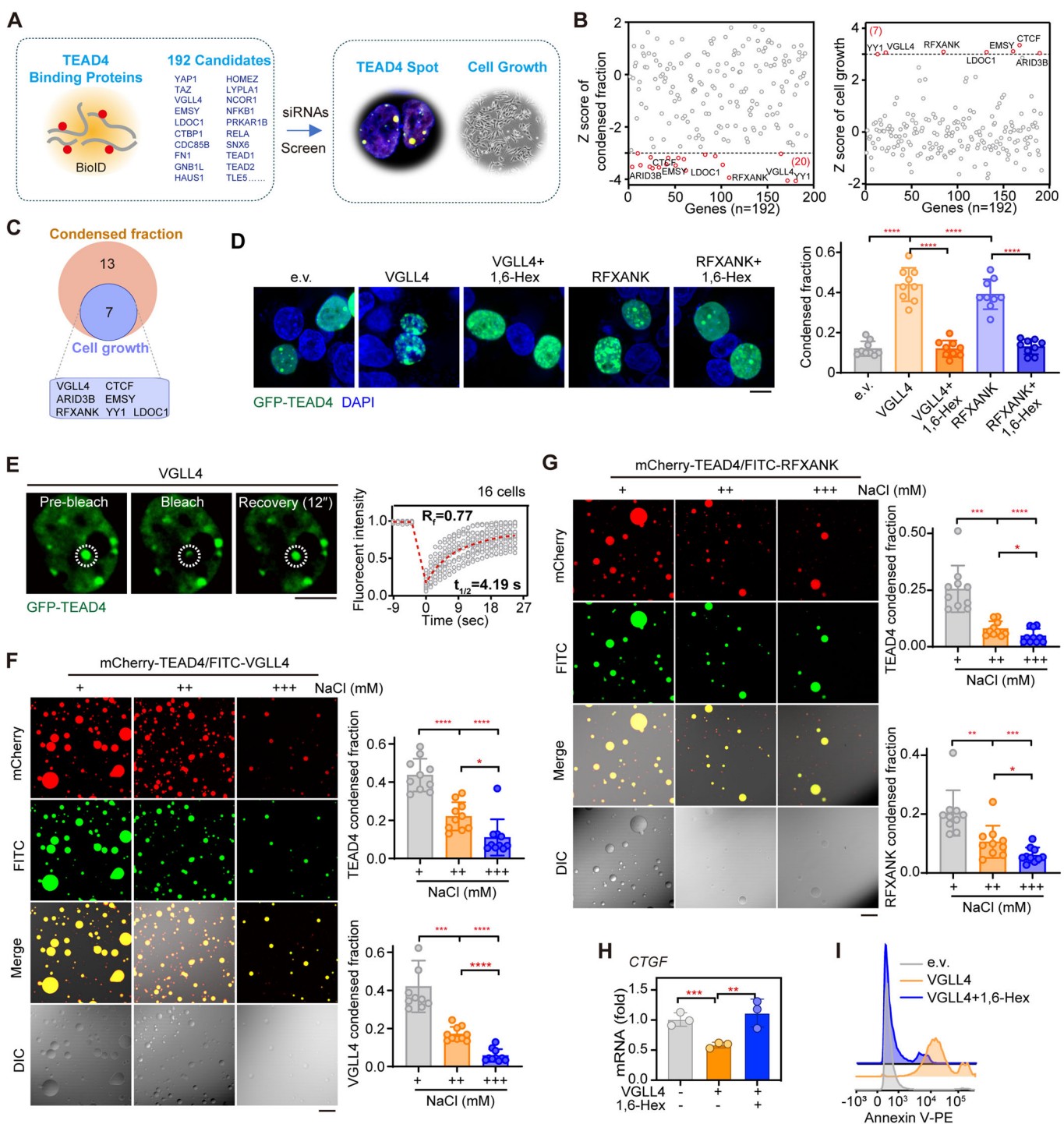

VGLL4-TEAD4 complex (Jiao et al, 2014) and found that each of the two TDU domains of one VGLL4 molecule binds one TEAD4 molecule, forming a 1:2 heterotrimer (Fig. 3A). Together with the crystal packing observed between TEAD4 molecules, these analyses implied that TEAD4 may form oligomers in cells expressing VGLL4. We then speculated that oligomerization of TEAD4 is a prerequisite for glucose-deprivation-induced TEAD4 condensation. To test this possibility, we performed co-immunoprecipitation (co-IP) experiments using two samples of TEAD4 molecules, tagged with different epitopes, in the context of glucose limitation (Appendix Fig. S3B). Indeed, self-association of TEAD4 was found to be significantly increased upon glucose deprivation, whereas 1,6-Hex treatment could weaken this effect. (Appendix Fig. S3B). We also observed a marked co-localization of mCherry-tagged TEAD4 with GFP-tagged TEAD4 in glucose-deprivation-generated TEAD4 condensates (Fig. 3B). Furthermore, native-PAGE revealed a smear of exogenous or endogenous TEAD4 aggregates of high molecular mass that was clearly enhanced after 6 h of glucose deprivation

**Figure 2. Identification of molecular inducers of TEAD4 condensation.**

(A) Schematic diagram of a strategy involving two rounds of screening for molecular inducers of TEAD4 phase separation. The first round utilized a proximity-based labeling system (BioID), which identified 192 candidates as TEAD4-interacting proteins (left). For the second round, individual siRNAs targeting the 192 candidates were used for two types of screening in parallel (based on TEAD4 condensate formation and cell viability, respectively) to identify regulators of TEAD4 LLPS (right). (B) Z score analysis of siRNA screening results, namely TEAD4 condensed fraction and cell growth. The cutoff values were $<-3$ (TEAD4 condensed fraction, left) and $>3$ (cell growth, right), respectively. (C) Venn diagram analysis showing the regulators of TEAD4 condensate formation and cell growth. Of the 192 candidates, 7 were identified as having a significant probability of being a regulator of TEAD4 LLPS. (D) Fluorescence images and quantification of TEAD4 condensates in VGLL4- or RFXANK-overexpressing HEK293FT cells that had been treated with or without 1,6-Hex. The quantification graph represents the data collected from 9 cells ($n = 9$). Quantification of TEAD4 condensed fraction (right) is shown. Data shown as means ± SD represent the representative results from two independent experiments. The data were analyzed using one-way ANOVA, followed by the Tukey's post-hoc test. ****$p < 0.0001$. Scale bar, 10 µm. (E) FRAP analysis of TEAD4 condensates in HEK293FT cells transfected with VGLL4. The graph represents the data collected from 16 cells ($n = 16$). White circles indicated the bleaching condensates. $t_{1/2}$: fluorescence recovery time; Rf: mobile fraction. Scale bar, 10 µm. (F) In vitro droplet formation assay for VGLL4-TEAD4 mixtures in the presence of different concentrations of NaCl ($n = 10$). NaCl (+), 100 mM; NaCl (++), 250 mM; NaCl (+++), 500 mM. Quantifications of, respectively, TEAD4 and VGLL4 condensed fractions are shown (right). The quantification graph represents the data collected from 10 images. Data shown as means ± SD represent the representative results from two independent experiments. The data were analyzed using one-way ANOVA, followed by the Tukey's post-hoc test. *$p < 0.05$; ***$p < 0.001$; ****$p < 0.0001$. Scale bar, 10 µm. (G) In vitro droplet formation assay for RFXANK-TEAD4 mixtures in the presence of different concentrations of NaCl. NaCl (+), 100 mM; NaCl (++), 250 mM; NaCl (+++), 500 mM. Quantifications of, respectively, TEAD4 and RFXANK condensed fractions are shown (right). The quantification graph represents the data collected from 10 images. Data shown as means ± SD represent the representative results from two independent experiments. The data were analyzed using one-way ANOVA, followed by the Tukey's post-hoc test. *$p < 0.05$; **$p < 0.01$; ***$p < 0.001$; ****$p < 0.0001$. Scale bar, 10 µm. (H) *CTGF* mRNA levels in VGLL4-overexpressing HEK293FT cells treated with or without 1,6-Hex (three biological replicates). Data shown as means ± SD represent the representative results from two independent experiments. Significance was assessed using one-way ANOVA, followed by the Tukey's post-hoc test. **$p < 0.01$, ***$p < 0.001$. (I) Annexin V staining of VGLL4-overexpressing cells treated with or without 1,6-Hex (three biological replicates, $n = 3$). e.v., empty vector. See also Appendix Fig. S2. Source data are available online for this figure.

(Fig. 3C), indicating that oligomerization of TEAD4 may promote the association of its condensates.

As oligomerization is proposed as a general driving force for LLPS of proteins (Carter et al, 2021), we asked whether TEAD4 LLPS also occurs via VGLL4-mediated oligomerization of TEAD4. As shown in Fig. 3D, overexpression of VGLL4 in cells indeed contributed to an increase in TEAD4 oligomerization. However, VGLL1, which has only one TDU domain and interacts with the TEAD4 molecule in a 1:1 ratio (Pobbati et al, 2012), did not affect TEAD4 oligomerization (Fig. 3D). Also, expression of VGLL1 failed to trigger TEAD4 condensation (Fig. 3E; Appendix S3C). Importantly, purified VGLL4 protein, but not VGLL1 protein, induced TEAD4 condensation (Fig. 3F), suggesting a need for at least two TDU domains to trigger TEAD4 oligomerization and condensation.

To test whether the number of TDUs truly is a key factor in promoting TEAD4 condensation, we generated a VGLL4 mutant (VGLL4$^{mut}$) in which one TDU domain was deleted, and a VGLL1 mutant (VGLL1$^{mut}$) in which an extra TDU domain was added at the C-terminus (Fig. 3G, upper). Notably, our co-IP assay showed VGLL4$^{mut}$ not promoting TEAD4 oligomerization compared to wild-type VGLL4 (Appendix Fig. S3D). By contrast, VGLL1$^{mut}$ with two TDU domains significantly increased TEAD4 oligomerization compared to wild-type VGLL1 (Appendix Fig. S3D). We also performed dynamic light scattering (DLS) to monitor TEAD4: strikingly, here, the VGLL species each harboring two TDUs (wild-type VGLL4, VGLL1$^{mut}$) induced a peak of TEAD4 oligomerization, while those with only one TDU (VGLL4$^{mut}$, wild-type VGLL1) did not show such an effect (Fig. 3G, lower). A DSS cross-linking experiment also confirmed these findings (Fig. 3H). Consistently, wild-type VGLL4 but not VGLL4$^{mut}$ can promote TEAD4 condensation (Fig. 3I; Appendix Fig. S3E). Moreover, wild-type VGLL4 protein but not VGLL4$^{mut}$ protein triggered TEAD4 droplet formation in an in vitro droplet formation assay (Fig. 3J), confirming the necessity of VGLL4 having two TDUs for it to promote TEAD4 oligomerization and condensation.

Overall, these data highlight the idea that VGLL4 triggers TEAD4 condensation by inducing its oligomerization, and that this process requires two TDU domains of VGLL4.

## TEAD4 condensation induces DNA aggregation and transcriptional repression

A protein undergoing LLPS can sequester binding partners as "clients", hence altering the biological function of these clients (Banani et al, 2016). Since TEAD4 is a transcription factor, we sought to determine whether DNA can serve as a client in VGLL4-mediated TEAD4 LLPS. Deploying gel mobility shift analysis, we first tested the changes in electrophoretic mobility of a plasmid including triple-tandem repeat muscle-CAT (M-CAT) DNA regulatory element (referred as M-CAT DNA hereafter) upon its binding TEAD4 (Fig. 4A). We found that presence of TEAD4 alone promoted the electrophoretic mobility of the DNA (Fig. 4A), and the migration of the DNA fragment was clearly held back in the presence of VGLL species with two TDUs (wild-type VGLL4, VGLL1$^{mut}$) but not in the presence of those with only one TDU (VGLL4$^{mut}$, wild-type VGLL1) or 1,6-Hex treatment (Fig. 4A,B). To further confirm that binding of TEAD4 to its cognate binding site is involved in VGLL4-mediated TEAD4 LLPS, we performed an EMSA experiment with the human TEAD4 S336A/K376A/V389A mutant, a construct unable to bind VGLL4. The results showed that TEAD4 or TEAD4$^{Mut}$ alone shifted the DNA (Fig. 4A, right). In contrast, a mixture of TEAD4 and VGLL4, but not a mixture of TEAD4$^{Mut}$ and VGLL4, held back the migration of the DNA segments (Fig. 4A, right). We also observed that VGLL4 markedly enhanced formation of droplets of the TEAD4-DNA complex (Fig. 4C), indicating that DNA was involved in the process of VGLL4-mediated TEAD4 LLPS.

LLPS is considered as principle for chromosome compartmentalization and condensation (Hildebrand and Dekker, 2020; Shin et al, 2018). Given the mounting evidence that nuclear condensation is a driving force behind chromatin organization and function

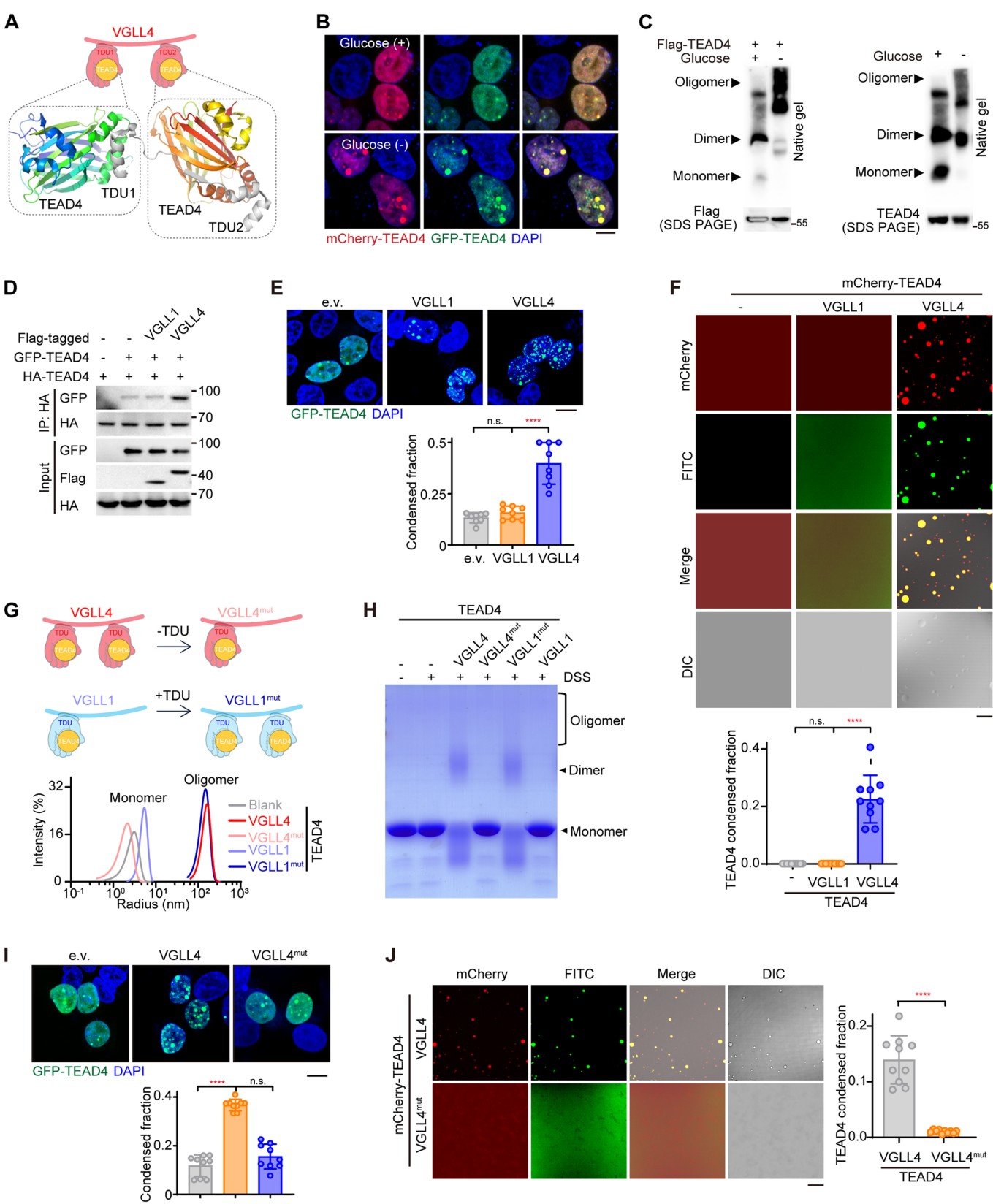

**Figure 3.  VGLL4 mediates TEAD4 LLPS by inducing its oligomerization.**

(A) Upper: cartoon illustration of the complex of VGLL4 (pink hand) and TEAD4 (yellow ball). Lower: crystal structure of the VGLL4 (TDU1/2, gray)-TEAD4 (YBD) complex. Ribbon depiction of the structure of two TEAD4 molecules in a single asymmetric unit. The two molecules are colored bluish-green and reddish-orange, respectively. (B) Colocalization of GFP-TEAD4 and mCherry-TEAD4 in HEK293FT cells with or deprived of glucose. Scale bar, 10 μm. (C) Immunoblots, using native gel or SDS-PAGE, of exogenous (left panel) or endogenous (right panel) TEAD4 in HEK293FT cells with or deprived of glucose. (D) Co-immunoprecipitation (co-IP) analysis to detect interactions between GFP-TEAD4 and HA-TEAD4 in HEK293FT cells transfected with VGLL1 or VGLL4. (E) Fluorescence images of TEAD4 condensates in HEK293FT cells overexpressing VGLL1 or VGLL4 (top) and quantification of TEAD4 condensed fraction (bottom). The quantification graph represents the data collected from 8 cells ($n = 8$). Data shown as means ± SD represent the representative results from two independent experiments. The data were analyzed using one-way ANOVA, followed by the Tukey's post-hoc test. n.s., no significance; ****$p < 0.0001$. Scale bar, 10 μm. (F) Images of droplet formation of purified mCherry-TEAD4 proteins in the presence of FITC-VGLL1 or FITC-VGLL4 (top) and quantification of TEAD4 condensed fraction (bottom). Data represent the representative results ($n = 10$ images) from two independent experiments. The data were analyzed using one-way ANOVA, followed by the Tukey's post-hoc test. n.s., no significance; ****$p < 0.0001$. Scale bar, 10 μm. (G) Dynamic light scattering (DLS) analysis of TEAD4 mixed with VGLL1, VGLL4, or their mutants. Cartoon illustrations of VGLL1, VGLL4, and their mutants (upper), and distributions of the hydrodynamic diameters of these molecules (lower) are shown. VGLL1$^{mut}$ is a construct in which TDU1 (amino acid residues 206–230) of VGLL4 was added to the N-terminus of wild-type VGLL1 to create a VGLL1 version with two TDUs. VGLL4$^{mut}$ is a construct in which TDU1 (amino acid residues 206–229) of VGLL4 was deleted to create a VGLL4 version with one TDU. (H) Cross-linking analysis of TEAD4 in the presence of VGLL1, VGLL4, or their mutants with Coomassie blue staining. DSS, a cross-linker reagent. (I) Fluorescence images of TEAD4 condensates in HEK293FT cells transfected with indicated plasmids (top), and quantification of TEAD4 condensed fraction (bottom). The quantification graph represents the data collected from 9 cells ($n = 9$). Data shown as means ± SD represent the representative results from two independent experiments. Significance was tested using one-way ANOVA, followed by the Tukey's post-hoc test. n.s., no significance; ****$p < 0.0001$. Scale bar, 10 μm. (J) Images and quantification showing mCherry-TEAD4 droplet formation in the presence of FITC-VGLL4 or FITC-VGLL4$^{mut}$. The quantification graph represents the data collected from 10 images ($n = 10$). Data shown as means ± SD represent the representative results from two independent experiments. Significance was tested using unpaired $t$ test, ****$p < 0.0001$. Scale bar, 10 μm. See also Appendix Fig. S3. Source data are available online for this figure.

(Lafontaine et al, 2021; Sabari et al, 2020), we speculated that aggregation of TEAD4 in the phase-separated droplets might disorganize the DNA/chromatin conformation and then inhibit its transcription. To test this hypothesis, we used confocal microscopy to observe the morphology of M-CAT DNA (Fig. 4D). DAPI staining showed M-CAT DNA chains appearing to be relaxed, with natural conformations (commonly known as a knob structure), and inclusion of TEAD4 proteins led to a slight tendency for M-CAT DNA to become aggregated (Fig. 4D). When VGLL4 protein was added into the mixture, the DNA chains appeared more aggregated and tangled (Fig. 4D). These results were consistent with a scanning electron microscopy analysis that also showed aggregated and tangled DNA upon addition of TEAD4 and VGLL4 (Fig. 4E).

Next, we used structure illumination microscopy (N-SIM) to examine deformation of DAPI-labeled chromatin in cells over-expressing VGLL4. DAPI staining of VGLL4-overexpressing cells showed chromatins that displayed an aggregated, condensed, and disorganized conformation (Fig. 4F). To further address this issue, we performed colloidal gold staining using TEAD4 antibody to observe the status of chromatin around TEAD4. As shown in Fig. 4G, we found TEAD4 (large black spots indicated by red arrows) having gathered nearby more chromatin (dispersed small spots) in VGLL4-overexpressed cells than in e.v.-overexpressed cells. The result also revealed a condensed chromatin around TEAD4. In comparison, nuclei of control cells appeared normal and homogeneous in shape and density in both DAPI staining and electron microscopic examinations (Fig. 4F,G), strengthening the idea that VGLL4-mediated TEAD4 LLPS induced formation of DNA aggregates and tangles, leading to disorganized chromatin conformation.

To further investigate the functional consequence of TEAD4 LLPS, we examined the signals of H3K27me3 and H3K27ac in VGLL4/RFXANK-generated TEAD4 condensates, keeping in mind that H3K27me3 and H3K27ac have been recognized as a "super-silencer" (Cai et al, 2021) and "super-enhancer" (Hnisz et al, 2013), respectively. To this end, we performed IF colocalization analysis in GFP-TEAD4-expressing cells, and found that H3K27me3, but not H3K27ac, colocalized with TEAD4 condensate mediated by

VGLL4/RFXANK-mediated (Fig. 4H; Appendix Figs. S4A–C), suggesting that TEAD4 condensates contribute to transcriptional repression. To further examine the potential effect of VGLL4-induced TEAD4 LLPS on the accessibility of chromatin to transcription machinery, we performed a chromatin immunoprecipitation (ChIP) assay using antibodies against H3K27ac, CTCF and RNA polymerase II (Pol II). ChIP-qPCR data at the *CTGF* gene showed weaker CTCF, Pol II and H3K27ac signals in HEK293FT cells transfected with VGLL4 than in the negative control— and, strikingly, showed these effects fully reversed upon treating the cells with 1,6-Hex (Fig. 4H, I), indicating a transcriptionally repressive effect dependent on TEAD4 LLPS.

We further performed a chromosome conformation capture (3C) assay to examine chromatin interactions across the *CYR61* locus. As expected, the quantitative results showed that the *CYR61* promoter region interacted frequently with a chromatin region of the *CYR61* enhancer (site 2 and site 3) in control cells, while the interactions of this chromatin loop were significantly weakened in VGLL4-overexpressing cells (Appendix Fig. S4D). Similar to the case for the *CYR61* locus, overexpression of VGLL4 also reduced the extent of the intrachromosomal looping at the *MYC* locus (Fig. 4J). Importantly, these interrupting or inhibitory effects of VGLL4 on chromatin looping were abrogated upon 1,6-Hex treatment (Appendix Fig. S4D and Fig. 4J), again indicating an effect of LLPS-dependent transcriptional repression. Together, we put forward a novel VGLL4-induced TEAD4 condensation inhibitory of TF functions.

## VGLL4-induced TEAD4 condensates against YAP-induced active ones

YAP/TAZ-induced TEAD4 condensation has been previously reported to promote TEAD4-dependent activation of transcription, in apparent contradiction to our newly proposed VGLL4-induced transcriptionally inhibitory condensation of TEAD4. To address this discrepancy, we dissected the domain architectures of YAP, VGLL4 and TEAD4. We noticed that the transcription factor TEAD4 contains a TEA DNA-binding domain and YBD domain

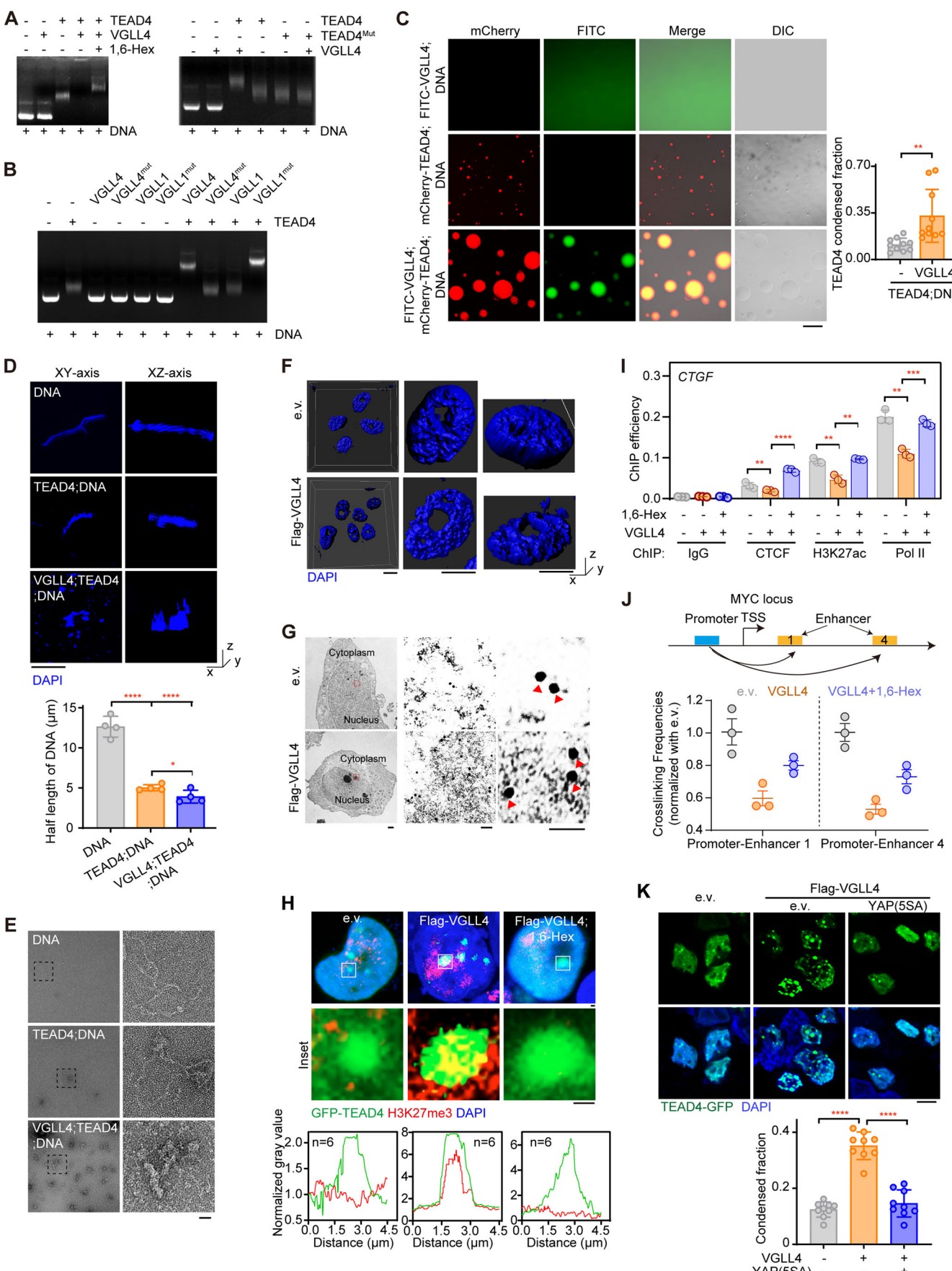

**Figure 4. TEAD4 LLPS induces aggregation of DNA to alter transcriptional status.**

(A) Electrophoretic mobility shift assay (EMSA) to detect TEAD4-DNA or TEAD4^Mut-DNA interaction in the presence or absence of VGLL4 with or without 1,6-Hex. The DNA referred to here was a triple-tandem repeat sequence of M-CAT (5′-TTGCATTCCTCTC-3′) inserted into a pUC-GW-Kan vector. (This same DNA was also used for subsequently described in vitro assays.) TEAD4^Mut: S336A/K376A/V389A of human TEAD4. (B) EMSA analysis to detect TEAD4-DNA interaction in the presence of VGLL1, VGLL4, or their mutants described in Fig. 3G. (C) Images showing droplet formation of the DNA with or without indicated proteins (left), and quantification of TEAD4 condensed fraction (right). The quantification graph represents the data collected from 10 images ($n = 10$). Data shown as means ± SD represent the representative results from two independent experiments. Significance was tested using unpaired $t$ test, **$p < 0.01$. Scale bar, 10 μm. (D) Representative images showing the DNA organization and appearance in the presence or absence of TEAD4 and VGLL4 proteins. The quantification graph represents the data collected from 4 images ($n = 4$). The DNA was stained with DAPI. Quantification of half-length of the DNA (bottom) is shown. Data shown as means ± SD represent the representative results from two independent experiments. Significance was tested using one-way ANOVA, followed by the Tukey's post-hoc test. *$p < 0.05$; ****$p < 0.0001$. Scale bar, 100 nm. (E) Scanning electron microscopy images of the DNA in the presence or absence of TEAD4 and VGLL4 proteins. The partially enlarged detailed view of the structure of the DNA is shown. Scale bar: 100 nm. (F) Representative images of chromatin with DAPI staining captured using N-SIM. Scale bar, 100 nm. (G) Immune electron microscopy (IEM) images showing TEAD4 particles in HEK293FT cells transfected with VGLL4. Red arrows denote TEAD4 particles containing colloidal gold-bound anti-TEAD4 antibody. Scale bars: 10 μm, 1 μm, and 10 nm for, respectively, the left, middle and right images. (H) Immunofluorescence staining (upper and middle, representative images with zoom-in; lower, quantification of fluorescence intensity of 6 cells with indicated color scheme) of GFP-TEAD4 and H3K27me3 in VGLL4-overexpressing cells treated with or without 1,6-Hex. Scale bar, 1 μm. (I) Chromatin immunoprecipitation-quantitative PCR (ChIP-qPCR) analysis for enrichment of CTCF, H3K27ac and polymerase II (Pol II) on *CTGF* promoter in VGLL4-overexpressing HEK293FT cells with or without 1,6-Hex (three biological replicates, $n = 3$). Data shown as means ± SD represent the representative results from two independent experiments. Significance was tested using one-way ANOVA, followed by the Tukey's post-hoc test. **$p < 0.01$; ***$p < 0.001$; ****$p < 0.0001$. (J) Schematic for and results of a chromosome conformation capture (3C)-based method used to assess the relative distances and potential interactions between the *MYC* promoter and its enhancers in VGLL4-overexpressing HEK293FT cells treated with or without 1,6-Hex (three biological replicates, $n = 3$). Data shown as means ± SD represent the representative results from two independent experiments. (K), Fluorescence images of TEAD4 condensates in VGLL4-expressing HEK293FT cells co-expressing e.v. or YAP(5SA) (top), and quantification of TEAD4 condensed fraction (bottom). The quantification graph represents the data collected from 9 cells ($n = 9$). Data shown as means ± SD represent the representative results from two independent experiments. Significance was tested using one-way ANOVA, followed by the Tukey's post-hoc test. ****$p < 0.0001$. YAP(5SA): S61A, S109A, S127A, S164A, S381A of YAP. Scale bar, 10 μm. See also Appendix Fig. S4. Source data are available online for this figure.

(cofactor-binding domain)—and YAP contains a TBD domain responsible for binding TEAD4 and hence able grab one TEAD4 molecule, and contains a transcription-activation domain (TAD) able to recruit polymerase II; in contrast, VGLL4 only contains TBD domains, hence lacking the ability to activate transcription (Appendix Fig. S4E, upper). Therefore, we reasoned that the transcription-activation domain of YAP/TAZ is vital for the YAP/TAZ-induced formation of transcriptionally active TEAD4 condensates. In contrast, VGLL4, which lacks the transcription-activation domain, induces the formation of TEAD4 inhibitory condensates (Appendix Fig. S4E, lower).

To verify this hypothesis, we went on to investigate whether the opposite functions of these two types of TEAD4 condensates are controlled by its cofactor YAP or VGLL4. First, we evaluated whether YAP could antagonize VGLL4-induced the formation of condensates. Here, transfection of a constitutively active YAP mutant (YAP 5SA) in VGLL4-expressed HEK293FT cells strongly decreased the number and area of TEAD4 condensates (Fig. 4K). Furthermore, we also performed colocalization analysis using IF in both YAP 5SA- and VGLL4-expressing cells, and found that H3K27me3 was significantly decreased in these TEAD4 condensates (Appendix Fig. S4F), indicating the ability of ectopic YAP expression to convert the TEAD4 repressive condensates into active ones. Also, transfection of YAP 5SA into glucose-deprived HEK293FT cells reduced the VGLL4-induced condensation of TEAD4 and induced an active TEAD4 LLPS (Appendix Figs. S4G–I). Re-supplementation of glucose into the glucose-deprived cells also yielded active TEAD4 LLPS (Appendix Fig. S4I), suggesting a reversible formation of TEAD4 condensates governed by cofactors.

In general, YAP having a transcriptional activation function can mediate TEAD4 active condensates to promote transcription and cell growth, whereas VGLL4 lacking a transcriptional activation function induced formation of repressive condensates, in which TEAD4 was shown to entangle DNA, triggering apoptosis (Appendix Fig. S4J).

## Engineering linker peptide GLUP to force repressive condensation of TEAD4

Considering the unique property of VGLL4-induced TEAD4 repressive condensation described above, we reasoned that a structural mimic of VGLL4 may also trigger TEAD4 condensation. To test this possibility, we synthesized a glue peptide (termed glup, intended to glue TEAD4 molecules together) with tandem-repeated core elements (HFHF) of the TEAD4-binding motif (Fig. 5A, left). Based on molecular docking, we predicted that glup could bind two molecules of TEAD4 via its tandem HF elements, forming a butterfly-shaped structure (Fig. 5A, right). Subsequently, we synthesized a fluorescein-labeled version of glup and confirmed, using a fluorescence polarization (FP) assay, its dose-dependent binding to TEAD4 protein (Appendix Fig. S5A). Further measurement with a microscale thermophoresis (MST) assay showed glup binding TEAD4 with a Kd value of 3.35 μM (Appendix Fig. S5B). In contrast, a mutant version of glup with H/F in the core elements substituted by alanine (termed as glup^mut) failed to directly interact with TEAD4 (Appendix Fig. S5B). To test whether glup is indeed able to "glue" TEAD4 molecules together, i.e., to induce oligomerization of TEAD4, we performed DLS using purified protein and the synthesized glup peptide, and here clearly found that TEAD4 formed dimers or oligomers of high molecular weight after 10–20 min of incubation with glup (Fig. 5B). Consistent with these observations, a DSS-cross-linking experiment showed that glup, but not glup^mut, induced TEAD4 oligomerization in a concentration-dependent manner (Fig. 5C; Appendix Fig. S5C).

To improve the stability and bioavailability of glup, we subjected it to disulfide-cyclization and C-terminal amidation to generate a macrocyclic version of the peptide and denoted it as GLUP

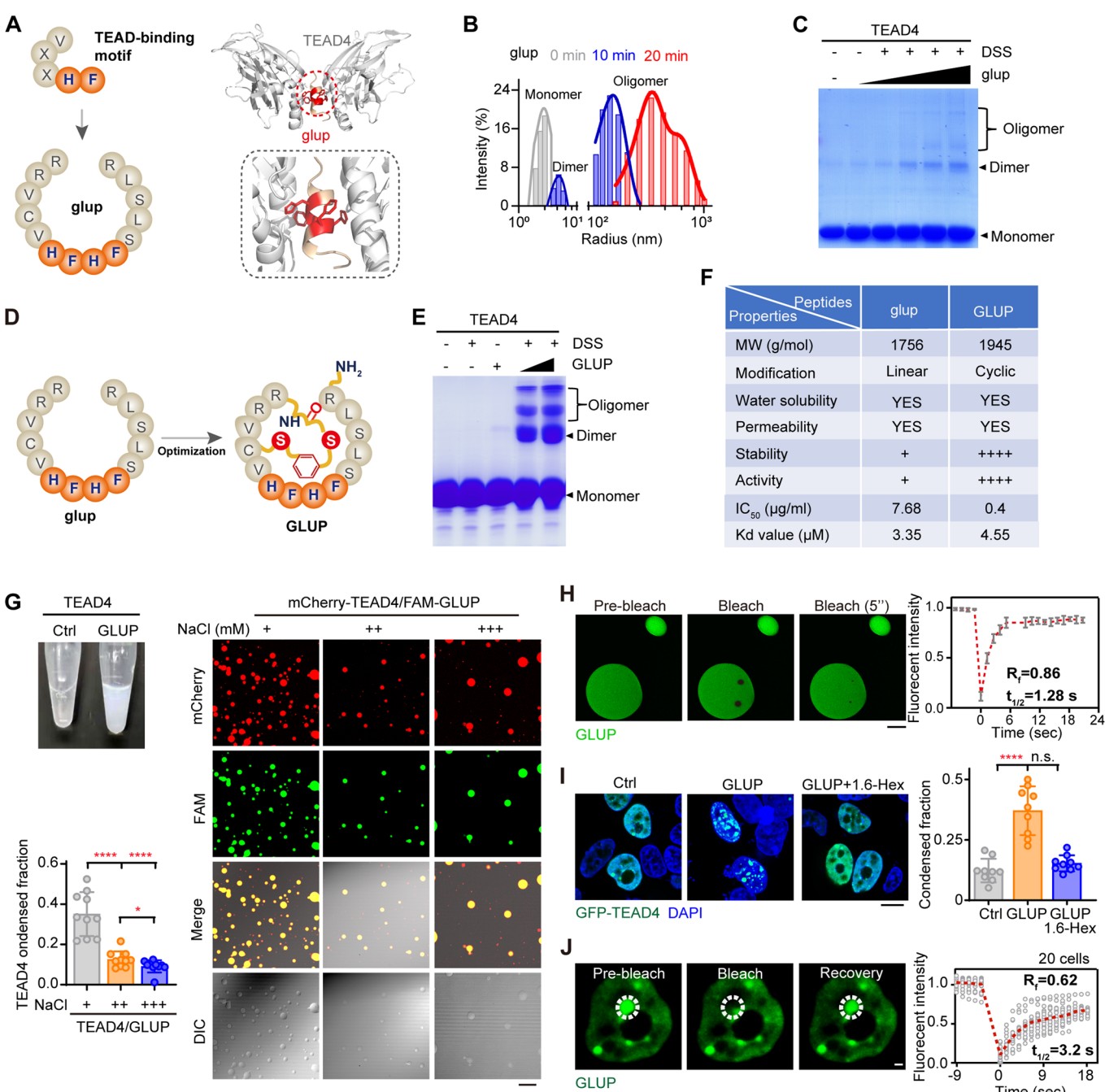

(Fig. 5D). The cross-linking assay showed that GLUP strongly induced TEAD4 oligomerization to occur, and did so in a concentration-dependent manner (Fig. 5E; Appendix Fig. S5D). As expected, GLUP displayed much higher stability than did glup: GLUP subjected to various proteases at 30 °C for 1 h remained intact while glup in these conditions was almost completely digested (Fig. 5F and Appendix Fig. S5E). However, there were no significant differences in solubility, permeability, and TEAD4-binding affinity between GLUP and glup (Fig. 5F and Appendix Figs. S5F–H). Notably, GLUP strongly promoted TEAD4 droplet formation (Appendix Fig. S5I), thus causing high turbidity in a

PEG solution (Fig. 5G, left). These droplets rapidly dissolved when exposed to increased concentrations of NaCl (Fig. 5G, right). Moreover, in vitro FRAP experiments on GLUP-induced formation of TEAD4 condensates yielded a τ value of about 1.28 s with a mobile fraction of about 86% (Fig. 5H). Similarly, treatment of HEK293FT cells with GLUP strongly induced formation of TEAD4 condensates, which was abrogated when the cells were also treated with 1,6-Hex (Fig. 5I). Moreover, FRAP experiments on GLUP-treated TEAD4 condensates yielded a τ value of about 3.2 s with a mobile fraction of about 62% (Fig. 5J). These results indicated a GLUP-induced formation of TEAD4 condensates via LLPS.

**Figure 5.  An engineered linker peptide to induce TEAD4 LLPS.**

(A) Schematic of the strategy of the initial version of a linker peptide termed "glup". Shown are a cartoon illustration of glup with core residues responsible for TEAD4-binding highlighted in orange (left), and a predicted structure of glup (orange) linking two TEAD4 molecules (right). The two TEAD4 molecules are colored cyan and blue, respectively. (B) DLS analysis of TEAD4 treated with glup for indicated periods of time (0, 10, 20 min). glup, 10 μg/ml. (C) Analysis of cross-linking of TEAD4 in the presence of increasing doses of glup. Monomer, dimer, and oligomer TEAD4 bands are labeled. (D) Schematic illustration of an optimized version of the linker peptide (referred to as GLUP). (E) Analysis of cross-linking of TEAD4 in the presence of increasing doses of GLUP. (F) Comparison of glup with GLUP. MW, molecular weight. Kd values, binding affinities with TEAD4. (G) GLUP-induced phase separation of TEAD4. Included are a photograph of vials containing TEAD4 proteins before and after they were treated with GLUP (top left), images showing droplet formations in TEAD4-GLUP mixtures exposed to different concentrations of NaCl (right), and quantification of TEAD4 condensed fraction (bottom left). The quantification graph represents the data collected from 10 images ($n = 10$). Data shown as means ± SD represent the representative results from two independent experiments. Significance was tested using one-way ANOVA, followed by the Tukey's post-hoc test. $*p < 0.05$; $****p < 0.0001$. NaCl (+), 20 mM; NaCl (++), 150 mM; NaCl (+++), 500 mM. Scale bar, 10 μm. (H) FRAP analysis of TEAD4 condensation with purified TEAD4 protein treated with FAM-GLUP. For each photobleached spot, the fluorescence recovery curve was traced. The graph represents the data collected from 5 droplets ($n = 5$). $t_{1/2}$: fluorescence recovery time; Rf: mobile fraction. Data shown as means ± SD represent the representative results from two independent experiments. Scale bar, 2 μm. (I) Representative fluorescence images of GFP-TEAD4 condensates in HEK293FT cells treated with GLUP for 12 h and/or 1,6-Hex for next continuous 2 h (left) and quantification of TEAD4 condensed fraction (right). The quantification graph represents the data collected from 9 cells ($n = 9$). GLUP, 10 μg/ml (5 μM). Data shown as means ± SD represent the representative results from two independent experiments. The data were analyzed using one-way ANOVA, followed by the Tukey's post-hoc test. n.s., no significance; $****p < 0.0001$. Scale bar, 10 μm. (J) FRAP analysis of TEAD4 condensates in GLUP-treated cells from (H). Three images were gradually taken during pre-bleach, bleaching pulses, followed by fluorescence recovery as indicated on the graph (left). The graph represents the data collected from 20 cells for the expression of GFP-TEAD4 (right, $n = 20$). The data represent the representative results from two independent experiments. White circles indicated the bleaching condensates. $t_{1/2}$: fluorescence recovery time; Rf: mobile fraction. GLUP, 10 μg/ml (5 μM). Scale bar, 10 μm. See also Appendix Fig. S5. Source data are available online for this figure.

---

Together, these results clearly indicated that a rationally engineered linker peptide, i.e., GLUP, can efficiently drive repressive condensation of TEAD4 through oligomerization.

## GLUP induces DNA aggregation to repress gene transcription

To confirm cellular entry of GLUP, we performed immunostaining of live cells treated with a fluorescein-labeled version of the peptide (FAM-GLUP). After 30 min of treatment, GLUP was found in the cytoplasm but outside of the nucleus. In contrast, it was found throughout in the cell including the nucleus when the treatment was for 80 min (Fig. 6A and Movies EV1–EV2), demonstrating the ability of GLUP to efficiently enter the cell and become distributed in the nucleus. We then examined the potential effect of GLUP on TEAD4-mediated perturbation of DNA conformation. Our in vitro assay using purified protein and synthesized peptide and M-CAT DNA clearly showed that GLUP strongly promoted TEAD4-mediated formation of DNA aggregations and tangles (Fig. 6B), as did VGLL4 protein treatment (Fig. 4D), indicating a gluing effect of GLUP on TEAD4-DNA. Confirming such an effect, DAPI staining of GLUP-treated cells also demonstrated that GLUP substantially disturbed chromatin distribution and organization (Fig. 6C). Consistently, scanning electron microscopy with colloidal gold staining revealed that GLUP induced extensive TEAD4 condensation, with concentrated chromatin around the condensates also observed (Appendix Fig. S6A). We also performed colocalization analysis, using IF, in GFP-TEAD4-expressing cells, and found strong signals of H3K27me3 but not H3K27ac in GLUP-mediated TEAD4 condensates (Fig. 6D), suggesting that TEAD4 condensates contribute to transcriptional repression.

Next, we conducted RNA-sequencing (RNA-seq) to evaluate the transcriptomics of HGC-27 cells treated with 10 μg/ml GLUP for 12 h. A total of 166 statistically significant DEGs (FDR < 0.05 and fold change > 2) were identified, of which 45 genes were found to be upregulated and 121 genes downregulated in GLUP-treated cells (Appendix Fig. S6B and Dataset EV3). As expected, gene set enrichment analysis (GSEA) revealed a negative enrichment of TEAD signature genes in GLUP-treated cells

(Appendix Fig. S6C). Remarkably, GLUP treatment induced an obvious downregulation in TEAD signature genes (for example, CTGF, CYR61, AXL, CCNA2 and CGB5) (Appendix Figs. S6D,E), again confirming the effect of GLUP-induced TEAD4 LLPS on cell-death-related transcription.

To further examine the perturbing effect of GLUP on chromatin accessibility and gene transcription, we performed a Cleavage Under Targets and Tagmentation (Cut&Tag) assay using antibodies specifically recognizing H3K27me3 (Cai et al, 2021). Notably, typical H3K27me3 peaks were found to be stronger for GLUP-treated cells than for control cells (Fig. 6E, upper). In contrast, tH3K27ac in GLUP-treated cells yielded weaker CTGF, CYR61 and CCNA2 signals than it did in control cells (Appendix Fig. 6E). Likewise, ChIP-qPCR results revealed that GLUP treatment led to decreased occupancies of H3K27ac and Pol II on the promoters of CYR61 and CCNA2 (Appendix Fig. S6F)—but to an increased occupancy of H3K27me3 on CTGF, with a smaller such increase when 1,6-Hex was also included (Appendix Fig. S6G).

To further address whether the transcription suppression is dependent on TEAD4 LLPS, we carried out an unbiased analysis of the Cut&Tag method using anti-TEAD4 antibody to confirm the validity of the ChIP-Seq assay. To this end, we performed triplicate experiments for both control-untreated (Ctrl) and GLUP-treated samples including a spike-in control to normalize the signals within each sample. We first calculated FRiP values for assessing data quality, and these values indicated the CUT&Tag data to be of high quality (Appendix Fig. S6H, FRiP value >0.03). We further analyzed the constituent peaks of TEAD4 in GLUP-treated cells: they showed weaker CTGF, TEAD4 and BCL2L1 signals than did those in control cells (Appendix Fig. S6I). Furthermore, to assess the specificity of GLUP, we divided the H3K27ac peaks into TEAD4-specific and non-specific groups. Interestingly, we found that GLUP treatment dramatically reduced the binding of H3K27ac onto the TEAD4-specific motifs, suggesting a targeting specificity of GLUP on TEAD4-binding gene promoters (Appendix Fig. 6J). At last, 1,6-Hex obviously abrogated GLUP-induced downregulation of CTGF (Appendix Fig. S6K). This result taken together with above-described results showed that GLUP could induce DNA aggregation to repress gene transcription.

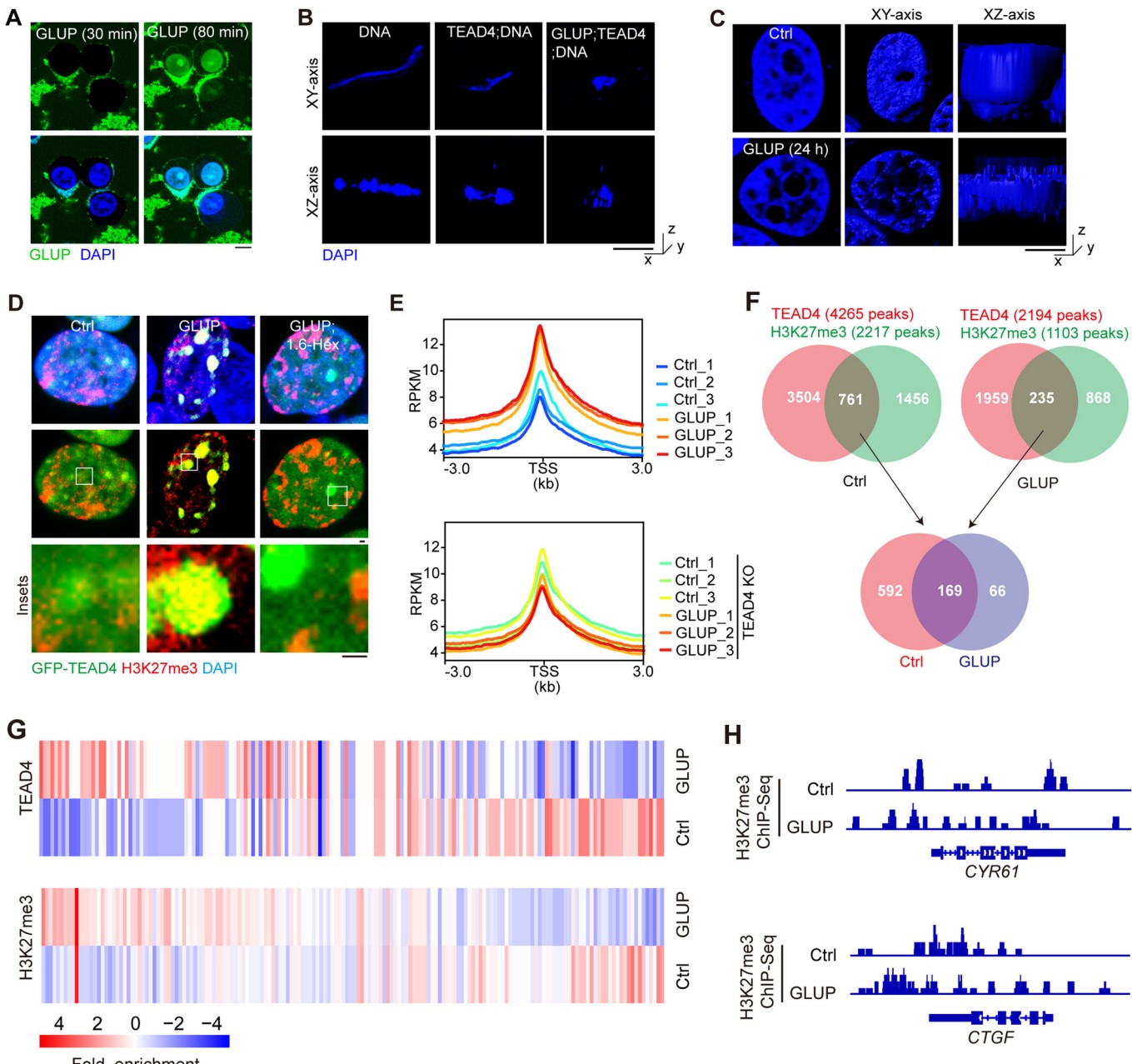

**Figure 6. GLUP treatment inhibits gene transcription mainly via TEAD4-induced repressive condensation.**

(A) Representative live-cell images of HGC-27 cells treated with FAM (a fluorescent moiety)-labeled GLUP for indicated periods of time. FAM-GLUP: 10 µg/ml (5 µM). Scale bar, 10 µm. (B) Representative DAPI staining images of DNA showing its organization and appearance in the presence or absence of TEAD4 and GLUP as indicated. GLUP: 1 µg/ml (0.5 µM). Scale bar, 100 nm. (C) Representative N-SIM images of chromatin in GLUP-treated cells stained with DAPI. GLUP, 1 µg/ml (0.5 µM). Scale bar, 100 nm. (D) Representative images showing co-localization of GFP-TEAD4 with H3K27me3 in HEK293FT cells treated with GLUP together with 1,6-Hex. Scale bar, 1 µm. (E) Average enrichment profiles of H3K27me3 in GLUP-treated WT or TEAD4KO HGC-27 cells, stratified by gene length (three biological replicates, $n = 3$). Normalization of coverage using RPKM was performed over the genes and flanking region 1 kb in length. GLUP: 10 µg/ml (5 µM). (F) Venn analysis of H3K27me3 and TEAD4 peaks revealed a group of co-existing peaks regardless of GLUP treatment. (G) Heatmap showing the chromatin association of H3K27me3 and TEAD4 onto the co-existing peaks with or without GLUP treatment. (H) Integrative Genomics Viewer (IGV) snapshot of H3K27me3 or TEAD4 Cut&Tag coverage upon indicated treatment. The interval scale was 10 in both cases. See also Appendix Fig. S6. Source data are available online for this figure.

To further elucidate the mechanism by which GLUP inhibits gene transcription, we performed Cut&Tag assay against H3K27me3 in both WT and TEAD4 KO cells. GLUP treatment here dramatically strengthened the signals of H3K27me3 at 3 kb regions flanking gene transcription start site (TSS) in WT HGC-27 cells, whereas no such effect was detected in TEAD4-depleted HGC-27 cells (Fig. 6E). Next, a Venn plot analysis we made to specify the peaks shared by both TEAD4 and H3K27me3 identified 761 and 235 co-existing peaks in control and GLUP groups, respectively (Fig. 6F). Among them, 169 peaks were found in both control and GLUP groups (Fig. 6F). These results allowed us to compare TEAD4 association with chromatin and H3K27me3 association with chromatin upon GLUP treatment. This analysis revealed a positive correlation between TEAD4 and H3K27me3 chromatin associations, i.e., the GLUP-induced TEAD4-binding peaks also showed elevated H3K27me3 signals (Fig. 6G), suggesting an involvement of H3K27me3 in the repressive TEAD4 condensates. These observations were further validated by the results of confocal imaging (Fig. 6D). Meanwhile, the constituent peaks of H3K27me3 in GLUP-treated cells showed weaker *CTGF* and *CYR61* signals than did those in control cells (Fig. 6H), suggesting that GLUP triggers repression of transcription mainly via TEAD4. Of note, we also observed GLUP treatment having reduced association of TEAD4 with chromatin in a subgroup of co-existing peaks (Appendix Fig. S6I), suggesting that GLUP-induced condensation of TEAD4 may sometimes strip TEAD4 off from chromatin. Nevertheless, the H3K27me3 signals were also weakened in these TEAD4-low peaks, consistent with the observed co-existence of TEAD4 and H3K27me3 signals.

## GLUP strongly reverses gastric cancer progression in mice

To evaluate the potential antitumor therapeutic effect of GLUP in vivo, we first assessed its bioavailability in mice. Consistent with its high stability in vitro (Fig. 5F; Appendix Fig. S5E), GLUP displayed a half-life of over 4 h in mouse peripheral blood (Appendix Fig. S7A). We then deployed a mouse model involving N-nitroso-N-methylurea (MNU)-induced GC, and treated such mice intraperitoneally with GLUP once tumors were palpable (Fig. 7A, upper). After a 5-week period of treatment, the average tumor area in mice treated with GLUP was much smaller than that in control mice (Fig. 7A,B). Such a dose-dependent inhibitory effect of GLUP was further confirmed by the observed sharply lower levels of Ki67 IHC staining in the GLUP-treated mice than in control littermates (Fig. 7C; Appendix Fig. S7B). Consistently and strikingly, we observed an almost normal gastric mucosa in mice receiving a high dose of GLUP (5 mg/kg), despite these mice having been exposed to MNU for weeks (Fig. 7C; Appendix Fig. S7B).

In addition, we also assessed the antitumor efficacy of GLUP using a xenograft GC mouse model. To this end, HGC-27 cells were inoculated into mice subcutaneously and tumors were allowed to grow until they were palpable, and the mice were then treated with GLUP (Fig. 7D). Similar to the observed therapeutic effect in the mouse model involving MNU-induced primary GC, treating the mice with GLUP inhibited growth of xenografted tumors in a dose-dependent manner as shown by the significantly reduced tumor volumes (Fig. 7D,E). Importantly, GLUP treatment also appeared to have induced formation of TEAD4 condensates in the tumor tissues (Fig. 7F). Moreover, upon GLUP treatment, expression of TEAD4 target gene *CTGF* was observed to be sharply decreased in a dose-dependent manner (Fig. 7G). Together, these results demonstrated a strong antitumor therapeutic effect of GLUP and further indicated a dependence of the effect on TEAD4.

## GLUP selectively kills human GC cells with elevated TEAD4 levels

To investigate the sensitivity of human GC cells to GLUP and select patients who might benefit from being treated with GLUP, we re-examined the inhibitory effect of GLUP on HGC-27 cells, which showed high levels of expression of TEAD4. As expected, treating HGC-27 cells with GLUP led to an obvious reduction in both the viability and colony formation of these cells (Fig. 8A,B). Subsequently, we measured the sensitivities of cells from 9 different human GC cell lines and one gastric epithelial cell line to GLUP (Fig. 8C). Also, we measured protein levels of TEAD4 in these cells and, based on this information, divided the cells into two groups: TEAD4$^{high}$ and TEAD4$^{low}$ (Fig. 8C). We also inquired the literature about the differentiation status of each of these cell lines. For example, HGC-27 is defined as undifferentiated (Leiherer et al, 2021), MKN-45 as poorly differentiated (Wang et al, 2017) and MGC-803 as showing low differentiation (Wu et al, 2011). For these GC cell lines, we found the grouping based on TEAD4 levels to be associated with the differentiation status. That is, GCs in the TEAD4$^{high}$ group were poorly differentiated, a phenotype shown to indicate that they are more aggressive (Fig. 8C). Importantly, we also noticed sensitivity of a given cell line to GLUP to be highly correlated with its expression level of TEAD4; i.e., the IC$_{50}$ value was found to be negatively correlated with TEAD4 protein level (Fig. 8C, lower).

To verify the dependence of GLUP sensitivity on TEAD4 expression in tumor cells, we chose AGS, a human GC cell line with a low level of TEAD4 expression (Fig. 8C, upper), for further testing. Indeed, the viability of AGS cells was only slightly or modestly inhibited by GLUP, showing the highest IC$_{50}$ of all examined cell lines (Fig. 8C, middle). However, AGS cells overexpressing TEAD4 but not an empty vector clearly became sensitive to GLUP as shown by their significantly reduced viability (Fig. 8D), results confirming TEAD4 as the target of GLUP. Also, we collected four lines of GC patient-derived cells (PDCs, #1–4) with relatively high (#1, #2) and low (#3, #4) copy numbers of TEAD4, respectively (Appendix Fig. S8A). Consistent with the above observations in GC cell lines, PDCs with high copy numbers of TEAD4 (#1, #2) were clearly more sensitive to GLUP than were PDCs with low copy numbers of TEAD4 (#3, #4) (Fig. 8E), again indicating the dependence of the GLUP therapeutic effect on TEAD4 expression of the tumor cell.

Furthermore, we also cultured 10 GC-patient-derived organoids (PDOs, #1–10), with #4 and #9 expressing the lowest levels of TEAD4 (Appendix Fig. S8B). Consistent with the results obtained from GC cell lines and PDCs, GLUP in a dose-dependent manner inhibited formation of PDOs (Fig. 8F). Of the 10 tested PDOs, however, #4 and #9 seemed to be most resistant to GLUP (Fig. 8F). We then picked PDOs #1 and #4 with moderate and low levels of TEAD4 expressions for further comparison and found that GLUP in a dose-dependent manner inhibited the growth of #1 but not #4 (Fig. 8G), confirming the TEAD4-dependent effect of GLUP.

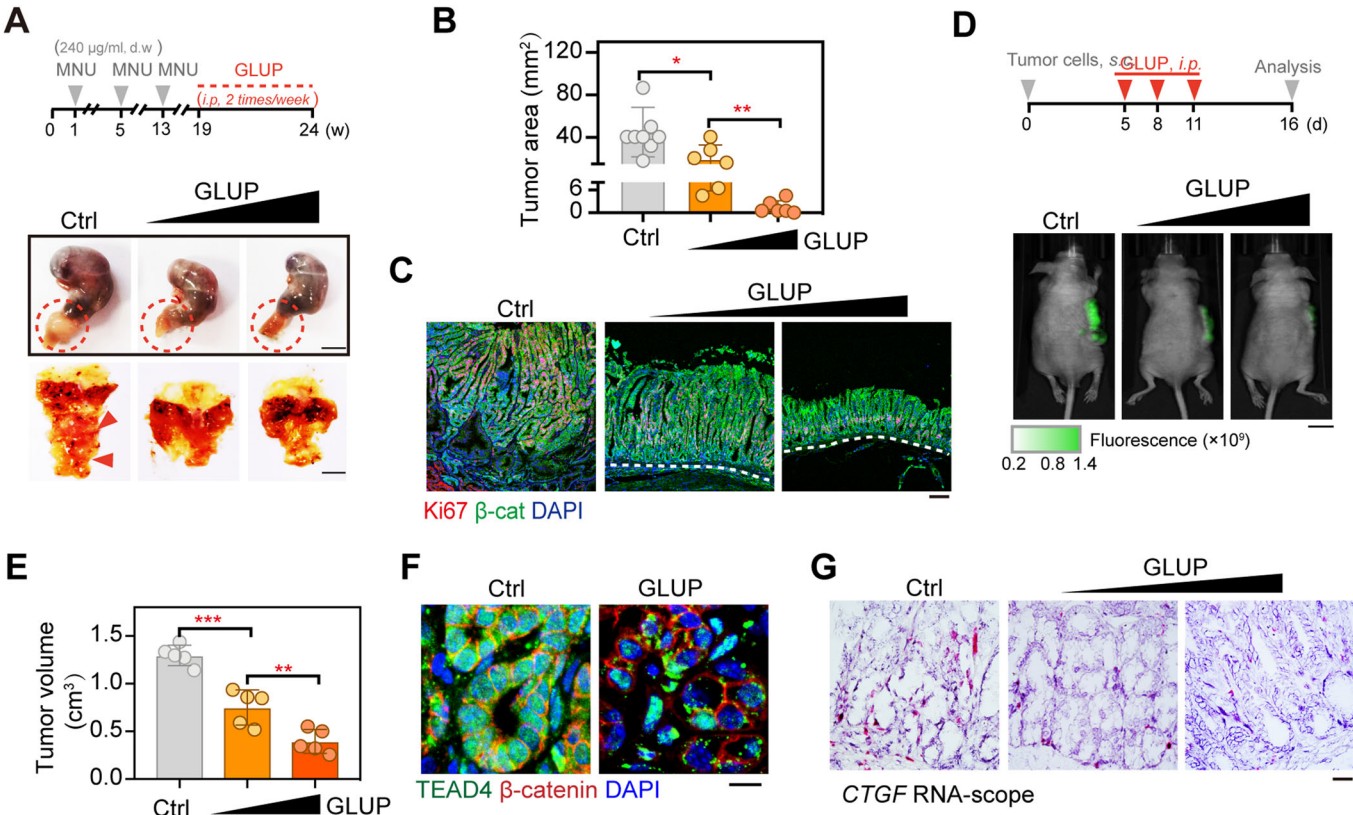

**Figure 7. GLUP displays a strong antitumor effect.**

(A) MNU-induced tumor formation in GLUP-treated mice. Experimental workflow of the treatment of MNU-administrated GC mice with 1 mg/kg or 5 mg/kg GLUP via subcutaneous injection (upper, same below for GLUP treatment). In the workflow, gray arrows represent, respectively, first, second, and third cycles of MNU treatment each with a 2-week duration, and orange dashed line represents GLUP treatment with a 5-week duration. *d.w.*, drinking water. Red circle indicated the tumor area. Macroscopic dissection view of the stomach (opened along the greater curvature) from each of MNU-induced mice with or without GLUP (lower). Red arrowhead indicated the tumor position. Scale bar, 1 cm. Data represent the representative results from two independent experiments. (B) Tumor areas of MNU-induced GC mice treated or not treated with GLUP ($n = 8$/group). Data are presented as means ± SD. Significance was tested using one-way ANOVA, followed by the Tukey's post-hoc test. *$p < 0.05$, **$p < 0.01$. (C) Dual immunofluorescence staining for Ki67 and β-catenin (β-cat) in the gastric mucosa of MNU-induced GC tumors in control mice and those treated with GLUP. Scale bar, 10 μm. Data represent the representative results from two independent experiments. (D) Xenograft tumor formation assay to assess the therapeutic efficacy of 1 mg/kg or 5 mg/kg GLUP treatment ($n = 5$/group). A schematic illustration of the experimental workflow for a subcutaneous tumor mouse model is shown (upper). GFP-expressing HGC-27 cells were subcutaneously injected into nude mice and allowed to grow for 5 days before the resulting tumors were treated with GLUP. Representative photographs of GLUP-treated tumors are also shown (lower). Scale bar, 1 cm. Data represent the representative results from two independent experiments. (E) Measured volumes of tumors in (D) ($n = 5$/group). Data are presented as means ± SD. Significance was tested using one-way ANOVA, followed by the Tukey's post-hoc test. **$p < 0.01$, ***$p < 0.001$. (F) Immunofluorescence of β-catenin and TEAD4 in tumor tissues of mice treated or not treated with GLUP. Scale bar, 10 μm. (G) RNA-scope of *CTGF* in the tumor tissues of mice treated or not treated with GLUP. Scale bar, 10 μm. See also Appendix Fig. S7. Source data are available online for this figure.

Moreover, after treating PDO #1 with GLUP for 12 h, significant levels of TEAD4 condensates were observed using immunofluorescence (Fig. 8H).

Finally, we used a patient-derived xenograft (PDX) model to assess the selectivity and therapeutic effect of GLUP. To this end, we chose one GC patient with high expression of TEAD4 (TEAD4[high]) and another with low TEAD4 expression (TEAD4[low]) for implantation. The xenografted tumors were allowed to grow for 1 week before treating the host mice with GLUP. After administration (*i.v.*) of GLUP for 2 weeks, tumors derived from the TEAD4[high] GC patient significantly regressed (Fig. 8I). In contrast, tumors derived from the TEAD4[low] GC patient did not respond to the GLUP treatment (Fig. 8I). That said, we did notice a range of blood biochemical abnormalities e.g., in numbers of platelets and neutrophils, upon GLUP treatment (Appendix Fig. S8C),

suggesting that its application needs to be spaced out considerably to avoid severe toxicity and side effects.

## Discussion

Cancer cells sense various stresses during tumorigenesis. Glucose limitation is one of the greatest challenges that cancer cells suffer (Koppenol et al, 2011; Schwartz et al, 2017). Notably, withdrawal of glucose can preferentially induce death of cancer cells, a process influenced by multiple signal transduction pathways (Buono and Longo, 2018; Wu et al, 2021). However, the current understanding remains fragmented on the signaling mechanisms underlying cell death induced by glucose deprivation. In this study, we discovered that glucose withdrawal rapidly induces condensation of a group of

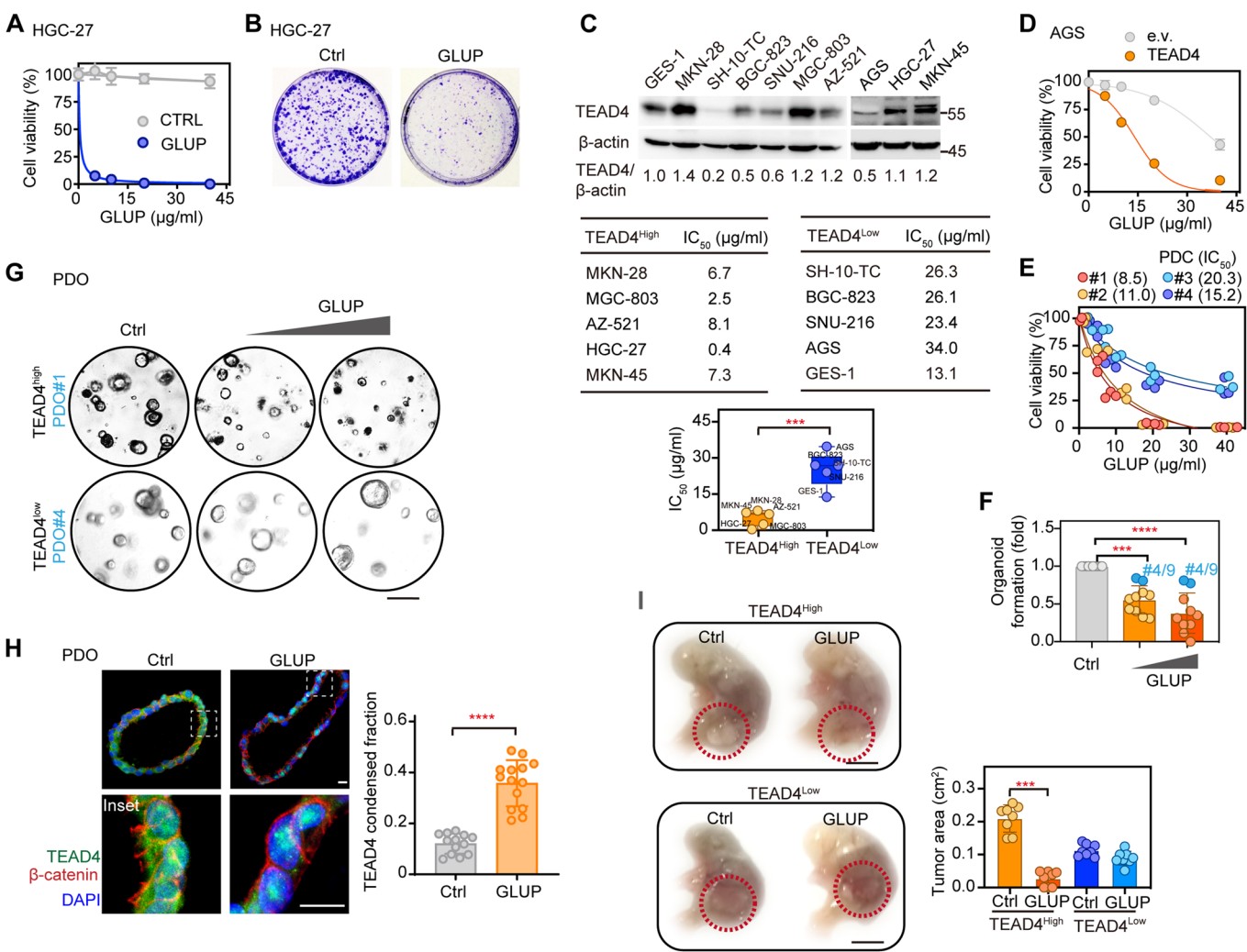

**Figure 8. TEAD4 expression levels dictate tumor sensitivity towards GLUP.**

(A) Cell viability levels of HGC-27 cells treated with indicated concentrations (0, 5, 10, 20, 40 µg/ml) of GLUP for 48 h (three biological replicates, $n = 3$). Data shown as means ± SD represent the representative results from two independent experiments. (B) Colony formation of HGC-27 cells treated with GLUP every 2 days for a 2-week duration (three biological replicates, $n = 3$). GLUP, 2 µg/ml (1 µM). (C) Cell viability levels of the indicated cells (from 10 different lines) treated with GLUP. Immunoblotting of TEAD4 in the cells (upper) (three biological replicates, $n = 3$). IC$_{50}$ values for the treatments of these cells with GLUP (lower). The expression of TEAD4 relative to that of β-actin was quantified according to the immunoblotting gray values and then used to group the indicated cells into TEAD4$^{Low}$ (0.2–1.0) and TEAD4$^{High}$ (1.1–1.4) groups. The box plots showing the minima, maxima, center, bounds of box and whiskers and percentiles (upper quartile, median and lower quartile). Data shown as means ± SD represent the representative results from two independent experiments. Significance was tested using the unpaired $t$ test, ***$p < 0.001$. (D) Cell viability of TEAD4-overexpressing AGS cells treated with GLUP for 48 h (three biological replicates, $n = 3$). Data shown as means ± SD represent the representative results from two independent experiments. (E) Cell viability levels of four lines of patient-derived cells (PDCs) treated with GLUP. The GLUP treatment IC$_{50}$ values are shown in parentheses (three biological replicates, $n = 3$). Copy numbers of TEAD4 were determined for these PDCs and are shown in Appendix Figs. S8A. (F) Patient-derived organoids (PDOs) treated with GLUP ($n = 10$). TEAD4 copy numbers determined for these PDOs are shown in Appendix Fig. S8B. Data shown as means ± SD represent the representative results from two independent experiments. Significance was tested using one-way ANOVA, followed by the Tukey's post-hoc test. ***$p < 0.001$; ****$p < 0.0001$. (G) DIC Images of representative above described (in (F)) TEAD4$^{high}$ and TEAD4$^{low}$ PDOs treated with 0.1 or 0.5 mg/ml GLUP. Scale bar, 10 µm. (H) Immunofluorescence of β-catenin and TEAD4 in PDOs treated with 0.1 mg/ml GLUP (left), and quantification of TEAD4 condensed fraction (right). The quantification graph represents the data collected from 14 cells ($n = 14$). Data shown as means ± SD represent the representative results from two independent experiments. Significance was tested using the unpaired $t$ test, ****$p < 0.0001$. Scale bar, 10 µm. (I) Patient-derived xenograft (PDX) model with representative TEAD4$^{high}$ or TEAD4$^{low}$ GC (left). Cells ($1 \times 10^6$) were injected into the gastric mucosa of immunodeficient mice. After cell transplantation for 5 d, mice were administered 5 mg/kg GLUP daily for another 15 d. Tumor areas were measured (right, $n = 8$/group) and indicated in red circles. Data shown as means ± SD represent the representative results from two independent experiments. Significance was tested using the unpaired $t$ test, ***$p < 0.001$. Scale bar, 1 cm. See also Appendix Fig S8. Source data are available online for this figure.

TFs, including TEAD4, EWSH, RFXDC1, GTF2A1L, C16orf5, ZNF800, and ELF1 (Fig. 1), a phenomenon hinting that TFs sense glucose deficiency by undergoing condensation with widespread and profound effects on cellular homeostasis. Importantly, we found that TEAD4 by itself could spontaneously form condensates in vitro. Instead, VGLL4/RFXANK was found to induce TEAD4 oligomerization and condensate assembly. Strikingly, we also demonstrated that the driving of TEAD4 into the condensate state

by VGLL4/RFXANK or an engineered peptide significantly limited cancer cell growth and increased apoptosis. In contrast to previous studies showing that glucose withdrawal induces cell death by multiple interconnected signaling pathways (Santagostino and Radaelli, 2021), our current findings revealed a new mechanism, i.e., an intranuclear mechanism related to condensation of TFs, for stress-induced cell death.

As LLPS is well-known for its role in promoting gene transcription activity, we believe one of the major innovative findings in our current work to be the discovery of a widespread phenomenon of condensation of TFs in response to a stressed condition such as glucose limitation. Therefore, we took TEAD4 as an example to showcase that TEAD4-mediated gene transcription can be switched on by YAP-triggered active condensation but switched off by VGLL4- or RFXANK-triggered repressive condensation. By understanding the underlying mechanism of VGLL4-triggered repressive LLPS condensation, we were able to manipulate or transform the TEAD4 condensates to a repressive state and thus induce cell death even for cells having a sufficient glucose supply. As such, we have provided a proof-of-concept for a new avenue to kill cancer cells, namely by realizing repressive TF condensation.

## Assembly and function of repressive TEAD4 condensates

The mammalian Hippo signaling pathway featuring MST1/2-LATS1/2-YAP/TAZ-TEADs as a core molecular axis has been extensively studied in development, tissue homeostasis, immune response, and tumorigenesis (Harvey et al, 2013; Jiao et al, 2018; Liu et al, 2016; Pan, 2010; Yu and Guan, 2013; Yu et al, 2015; Yu et al, 2015). Mounting evidence has demonstrated multiple regulatory roles for LLPS in this pathway. Some studies have highlighted that the bidirectional cytoplasm-nucleus shuttling of YAP/TAZ is much more dynamic than previously reported, and that YAP/TAZ are also regulated by LLPS (Cai et al, 2019; Franklin and Guan, 2020; Lu et al, 2020; Wei et al, 2021; Yu et al, 2021). For example, TAZ forms condensates via LATS-regulated LLPS to differ the function of coactivators BRD4 and MED1, the transcription elongation factor CDK9, and its DNA-binding cofactor TEAD4. YAP condensates can co-localize with super-enhancer markers such as Oct4, Sox2, H3K27ac and Nanog in mouse embryonic stem cells (Sun et al, 2020). In addition, LATS1 condensates have been reported to promote carcinogenic YAP signaling (Kastan et al, 2021). Most recently, AMOT or KIBRA were reported to together form condensates to activate Hippo signaling, and these condensates may coalesce with SLMAP condensates to inhibit STRIPAK function (Wang et al, 2022).

Here, we have focused on those nuclear condensates of TFs involved in gene expression and stress-induced cell death. Notably, most studies to date have documented a strong association of LLPS of TFs and their transactivators with transcriptional activation and tumorigenesis (Boija et al, 2018; Franklin and Guan, 2020; Kumar et al, 2012; Lu et al, 2020; Zanconato et al, 2016). In contrast to this transcription-promoting role, we found that condensates of TEAD4 can be formed in response to glucose deprivation to inhibit target gene transcription. Interestingly, purified TEAD4 protein did not form condensates in vitro. But VGLL4/RFXANK, a negative regulator of TEAD4, can drive TEAD4 LLPS droplet formation either in vitro or in vivo (Fig. 2). Mechanistically, VGLL4/

RFXANK-induced TEAD4 condensation was indicated to trigger significant DNA/chromatin aggregation and tangling, leading to decreased gene expression. Note that high concentrations of H3K27me3, a marker for transcriptional "super-silencer" (Cai et al, 2021) were found to be maintained in these TEAD4 condensates (Appendix Fig. S4). Note the consistency of our observation with a recent finding that MYC multimers accumulate on chromatin to generates a zone of transcription termination (Solvie et al, 2022). Therefore, we propose a transcriptionally repressive role for condensates of TFs like TEAD4 during stress-induced cell death. At this point, we do not know the specific components of TEAD4 condensates that define the exact mechanism of this transcription-repressive function. One possibility is that TEAD4 condensates facilitate the recruitment and assembly of a super-silencer in response to stress signals to repress transcription.

## TDU numbers dictate the ability of VGLL proteins to induce TEAD4 condensation

The VGLL family of proteins share the same TEAD-binding domain, i.e., the TDU domain, but some members of this family display opposite functions. This finding is puzzling because these VGLL proteins lack other functional domains and the only difference between them seems to be their number of TDUs. For example, VGLL1–3 proteins share one conserved TDU domain at their N-terminal regions; while VGLL4 has two partially conserved TDU domains at its C-terminus (Chen et al, 2004; Maeda et al, 2002; Pobbati et al, 2012; Vaudin et al, 1999). In the current work, we re-identified VGLL proteins as binding partners of TEAD4 with distinct valences.

We first clarified that TEAD4 exists as an ensemble of monomers, dimers, and high-order oligomers in the tested cells, but purified TEAD4 protein appears to predominately be quasi-stable monomers in vitro. Furthermore, we demonstrated that VGLL4, but not VGLL1, was able to promote TEAD4 phase separation, highlighting the inability of TEAD4 itself to undergo LLPS spontaneously but that its LLPS instead depends on VGLL4 as an inducer (Fig. 3). Our systematic mutational study clearly revealed that a minimum of two TDUs is required for VGLL proteins to initiate multi-valent heteromeric interactions with TEAD4 and therefore induce condensation, confirming a model whereby multi-valent (>2) protein-protein interactions serve as "responders" to physiological or stress conditions and can structurally alter molecular crowding (Boeynaems et al, 2018; Hyman et al, 2014; Jalihal et al, 2020; Sukenik et al, 2017). In the case of the VGLL family of proteins, the number of TDUs dictates the valence of the VGLL-TEAD interaction, explaining the opposing functions: having only one TDU makes VGLL1–3 serve as regular transcriptional co-activators to promote growth; while having two TDUs makes VGLL4 able to induce TEAD4 condensation and thus manifest as a transcriptional repressor to promote cell death.

Based on the principle of having two TDUs in the protein molecule be responsible for TEAD4 condensation, we further investigated the sequence of our newly identified RFXANK, which similarly induced TEAD4 condensation, to examine whether RFXANK harbored the TDU motif (V/IxxH/LF). We not surprisingly found two putative TDU motifs in RFXANK, at amino acid residues 39–43 and 246–250, suggesting that the number of TDUs

dictates the ability of VGLL proteins to induce TEAD4 condensation.

## Cofactors differentially mediate repressive or active condensation of TFs

Gene transcription is a tightly regulated and controlled the fate of a cell. The regulation of gene expression maintains homeostasis, whereas dysregulation of transcription leads to serious consequences such as tumorigenesis. Not only individual TFs but also complexes composed of TFs and cofactors (coactivators and corepressors) regulated gene transcription. Transcription cofactors can participate in gene organization and phase separation to control the process of transcription. In the last decade, multiple TFs are found to allow for multivalent weak interactions with other TFs and/or cofactors, subsequently leading to the formation of phase-separated condensates that promote transcription activity.

In contrast to the well-known condensation-induced activation of transcription, repressive TF condensation has been poorly studied. In this work, we dissected the molecular mechanism of YAP- or VGLL4-induced phase separation and proposed a machinery in which active and repressive condensations of TEAD4 play opposite roles for gene transcription (Fig. 4M). As previously reported (Jiao et al, 2014), the binding affinity between TEAD4 and YAP (Kd = 2.1 nM) is higher than that between TEAD4 and VGLL4 (Kd = 6.8 nM), and hence YAP can compete with VGLL4 and induce active condensation of TEAD4. Upon glucose starvation, YAP is translocated from the nucleus to the cytoplasm. This change in YAP localization allows VGLL4 to predominate in the nucleus. Thus cofactor-mediated differential condensation of TFs may act as a switch to control transcriptional activity. The point is that we can manipulate and change the state of condensates to control cell fate.

## Driving formation of repressive condensates as a new class of antitumor strategies

Cancer development has been closely linked with phase-separated macromolecular condensates, including stress granules, DNA repair condensates, PRC1 condensates, super-enhancer condensates, SPOP/DAXX bodies and PML puncta (Bergeron-Sandoval et al, 2016; Ong and Torres, 2020; Taniue and Akimitsu, 2022; Zbinden et al, 2020). Recent studies have shown that coactivators, mediators, and some TFs may be enriched on super-enhancers to form condensates generated as a result of phase separation (Boija et al, 2018; Cho et al, 2018). For example, β-catenin selectively occupies super-enhancer to form condensates via interacting with DNA-binding factors (Zamudio et al, 2019). During tumorigenesis, p53 mutants have the potential to form LLPS-like condensates (Lemos et al, 2020). As aforementioned, Hippo signaling molecules could also be activated through phase separation and thus play a key role in tumor progression (Kastan et al, 2021). Therefore, emerging studies have opened the possibility of targeting phase separation as an important way to treat cancer.

That said, most biomolecular condensates identified to date do not appear to limit cell growth or induce cell death. Perhaps, the very reason for this is that LLPS condensates can concentrate certain required factors within a limited space and time, while excluding other factors that suppress the related biological processes. We initially aimed to identify TFs that can undergo condensation in a context of glucose deprivation mimicking the tumor microenvironment (Fig. 1). TEAD4, one of the seven identified TF candidates, has been well studied as a downstream TF of the Hippo signaling pathway and has also been studied as an attractive target for cancer therapy. Here we discovered that TEAD4 can form repressive condensates in response to nutritional stress (glucose limitation) and that such TEAD4 condensates can trigger aggregation and tangling of DNA/chromatin, eventually causing cell death. Importantly, we were able to take this opportunity to forcefully drive the transformation of TEAD4 into repressive condensates by "gluing" TEAD4 molecules together with a rationally designed peptide, namely GLUP. And our study indeed led to the development of a GLUP antitumor strategy based on the finding of a repressive type of TF condensation.

Our study provided TEAD4 LLPS as an example of an artificial induction of the formation of cell-growth-limiting condensates, opening a new perspective to elicit a biomolecular-condensate-based antitumor effect for therapeutic intervention of cancer. As a molecular superglue for TEAD4, GLUP was shown to efficiently induce cancer cell death even in a cellular context with abundant glucose supply, yielding a strong antitumor effect in multiple mouse models including PDX. On top of the high selectivity of GLUP in targeting TEAD4, our chemical modifications further improved the potential of GLUP as a drug candidate. This peptide was found to be highly soluble in water and highly stable both in vitro and in vivo. Moreover, penetration of GLUP into cells was observed to be highly efficient. That said, further investigations are still required to corroborate the antitumor effect of GLUP in both preclinical and clinical settings with careful examination of possible side effects. Beyond the example of artificially inducing formation of TEAD4 repressive condensates, it is theoretically possible to identify more highly repressive condensates and dissect the related molecular mechanisms, which would offer further proof of concept for—and aid in the development of—this new class of therapeutic approaches.

## Limitations of this study

This study revealed a death-promoting role for TEAD4 condensation in glucose-limited cells. Yet other nutritional or non-nutritional stresses might also induce TEAD4 LLPS for currently unrecognized functions. Also, while we identified VGLL4 as a natural inducer of TEAD4 LLPS, the mechanisms for other potential inducers such as RFXANK remain to be fully dissected. Our results support the idea that a repressive complex can be assembled in TEAD4 condensates, but the specific machinery causing the transcriptional repression remains obscure. For example, it is not yet fully understood how TEAD4 in these condensates interacts and entangles DNA/chromatin to hinder the assembly of the transcription-related regulatory machinery. Alternatively, TEAD4 LLPS may possibly concentrate certain transcriptional repressors, while excluding other factors that enhance transcription. Notably, the observation of GLUP treatment decreasing chromatin binding of TEAD4 in a subset of genomic regions suggests that GLUP may suppress gene transcription either by effecting TEAD4-mediated DNA entangling or by stripping TEAD4 off chromatin. In this regard, further study is required to clarify the specific mechanism by which GLUP mediates suppression of gene transcription.

# Methods

## Plasmids

His-SUMO-tagged TEAD4 corresponding to amino acid residues 210–434, His-SUMO-tagged TEAD4 corresponding to amino acid residues 36–434 with Δ121–210, His-mCherry-tagged TEAD4, His-GFP-tagged TEAD4, His-tagged VGLL4 corresponding to amino acid residues 203–256, and His-tagged YAP were cloned into respective pET28a vector as described previously (An et al, 2020; Jiao et al, 2014). A TF library from Prof. Dan Ye including 759 Flag- or Gal4-tagged TFs was prepared as described previously (Xia et al, 2021). Flag-tagged full-length TEAD4, RFXANK, VGLL1 and VGLL4 were cloned into respective pcDNA3.1 vector. HA-TEAD4 was inserted into a pcDNA3.0 vector. Flag-TEAD4 was subcloned into a pCDH-EF1a-BirA lentivector for lentiviral expression. mCherry-TEAD4 and GFP-TEAD4 were introduced into a pEGFP-N1 vector and pEGFP-C1 vector, respectively.

VGLL1^mut was generated by replacing the coding region corresponding to amino acid residues 1–28 in wild-type VGLL1 with that corresponding to amino acid residues 206–230 of VGLL4. VGLL4^mut was generated by deleting the coding region corresponding to amino acid residues 206–229 in wild-type VGLL4. All DNA fragments encoding proteins of interest were amplified using PCR with 2 × Hieff® PCR Master Mix (Yeasen, Shanghai, China). Inserted sequences were validated using Sanger sequencing.

## Cells

HEK293FT, HEK293A and cancer cell lines including AGS, AZ-521, BGC-823, MGC-803, and HGC-27 cells were obtained from the cell library of the Chinese Academy of Sciences (Shanghai, China). MKN-45 and GES-1 cells were from the RIKEN BioResource Center (Tsukuba, Japan). SNU-216 cells and MKN-28 were from ATCC and the JCRB Cell Bank (NIBIOHN, Japan), respectively. HEK293A-YAPKO cells were kindly provided by Prof. Faxing Yu (Fudan University).

HEK293FT, HEK293A and HEK293A-YAPKO cells were cultured in DMEM medium (Invitrogen). Other cells were grown in RPMI 1640 medium (Invitrogen). Patient-derived cells (PDCs) were established as described previously (Tang et al, 2020). All cells were maintained in culture supplemented with 10% heat-inactivated FBS (Biological Industries) and 1% penicillin/streptomycin (Gibco) at 37 °C with 5% $CO_2$ in a humidified incubator (Thermo). The cells were regularly checked for mycoplasma contamination using MycoAlert kits (Lonza, ME, USA). The identities of all used cell lines were confirmed by carrying out fingerprinting (Shanghai Genening Biotechnologies Inc., Shanghai, China).

## Screening based on fluorescent puncta

TF plasmids were automatically added to black-walled 384-well plates at 80 ng per well to reach a final concentration of 1.6 μg/ml. Polyethylenimine (PEI, Sigma) was then added to the plasmid-loaded plates at 240 ng per well (4.8 μg/ml). Subsequently, HEK293FT cells were seeded into the plate at 2000 cells per well. After 60 h, cells were cultured in medium with or without glucose for 12 h, and then fixed with 4% paraformaldehyde (PFA) at room temperature for 30 min. After three washes with PBS, fixed cells

were permeabilized with 0.05% Triton X-100 for 15 min. Thereafter, cells were blocked with 5% BSA for 1 h, and then incubated overnight with anti-Flag (Sigma F3165, 1:200) and anti-GAL4 (Abbkine ABP57232, 1:200) primary antibodies at 4 °C. The cells were then gently washed with PBS, followed by being incubated with Alexa Fluor Plus 488 goat anti-mouse IgG secondary antibody (Thermo A32723, 1:400) or Alexa Fluor Plus 488 goat anti-rabbit IgG secondary antibody (Thermo A32731, 1:400) for 1 h. The plates were kept at 4 °C in the dark before being analyzed using an Opera System (Perkin Elmer). Fifty random fields per well were scanned and analyzed using the Columbus Plus Image Data Storage and Analysis System (Perkin Elmer).

## Immunofluorescence and FRAP assay

For immunofluorescence assays, cells were fixed with 4% PFA for 30 min. Images of cells incubated with primary and secondary fluorophore-conjugated antibodies were captured using a Zeiss LSM880 confocal microscope (Jena, Germany) with a 63× oil immersion lens. Twenty-five cells were scanned and analyzed using Fiji ImageJ software. The antibodies used for immunofluorescence assays included anti-H3K27ac (Cell Signaling Technology #8173, 1:100), anti-H3K27me3 (Cell Signaling Technology #9733S, 1:100), anti-Flag (Sigma F3165, 1:200), Alexa Fluor Plus 488 goat anti-mouse IgG (Thermo A32723, 1:400) and Alexa Fluor Plus 568 goat anti-rabbit IgG (Thermo A11011, 1:400). DAPI (Sigma D9542, 1:2000) was used to stain nuclear DNA.

For FRAP analysis, cells were seeded at a confluency of ~30% on a 35-mm glass-bottomed dish (Cellvis D35C4-20-1-N). Live cell imaging was carried out using a Leica SP8 confocal microscope (Leica, Germany) with a 63× oil immersion objective lens. A defined region with a diameter of ~1 μm was photobleached by the equipped laser at ~90% power intensity. Fluorescence intensities of the region were recorded by carrying out time-lapse live imaging with a 1.29 s interval and further quantified using Fiji ImageJ software.

## Protein expression and purification

The recombinant proteins were expressed in the *Escherichia coli* BL21(DE3) strain as described previously (Jiao et al, 2017; Jiao et al, 2014). In brief, protein expression in BL21 cells ($OD_{600} = 0.8$) was induced by 0.25 mM isopropyl-β-d-thiogalactopyranoside (IPTG) for 18 h at 18 °C. Collected bacteria were lysed with a high-pressure homogenizer, with the resulting material subjected to centrifugation at 18,000 rpm for 45 min at 4 °C. The supernatant was incubated with Ni-NTA agarose (GE Healthcare) for 2 h, and then washed with wash buffer (20 mM HEPES pH 7.5, 500 mM NaCl, 5% glycerol, 20 mM imidazole and 1 mM DTT). The proteins were finally purified using size-exclusion chromatography with a stocking buffer (20 mM HEPES pH 7.5, and 100 mM NaCl). Protein purity was examined by carrying out Coomassie blue staining, and protein concentrations were determined from the results of Bradford protein assays.

## In vitro LLPS assay

LLPS was induced at room temperature by diluting proteins and NaCl to specified concentrations (20, 150, 500 mM). After

subsequent mixing, 10 μl of the reaction solution were gently transferred onto a 20 mm glass-bottomed dish (NEST, 801001) for differential interference contrast (DIC) imaging performed using a Zeiss LSM 880 confocal microscope equipped with a 20× water immersion objective lens. Unless otherwise indicated (e.g., in the figure legend), the final concentration of purified protein used for in vitro LLPS assays was 40 μM.

## Immunoprecipitation and immunoblotting

For immunoprecipitation assays, cultured cells were harvested and lysed with an NETN lysis buffer (20 mM Tris·HCl pH 8.0, 100 mM NaCl, 0.5% NP-40, and 1 mM EDTA) supplemented with protease and phosphatase inhibitors for 30 min on ice. Cell lysate was then centrifuged at 12,000 rpm for 10 min at 4 °C to collect supernatant. The supernatant was incubated with anti-Flag (1:1000) or anti-HA (Cell Signaling Technology #3724, 1:1000) antibodies for 1 h at 4 °C, followed by being incubated with Protein A/G plus agarose beads (Santa Cruz, sc-2003) overnight at 4 °C. Thereafter, the beads were washed with the lysis buffer and then boiled in 1×SDS loading buffer (50 mM Tris-HCl pH 6.8, 6% SDS, 6% glycerol, 0.01% bromophenol blue, 2 mM DTT), after which they were used as samples for immunoblotting analysis.

For immunoblotting, proteins in samples were separated using SDS-PAGE, transferred onto 0.45 μm PVDF membranes (Bio-Rad, Hercules, CA), and subsequently blotted with the indicated antibodies. Immunoblotting images were acquired using the Mini Chemiluminescent Imaging and Analysis System (Beijing Sage Creation Science, China). The antibodies used for immunoblotting included anti-GFP (Santa Cruz sc-9996, 1:1000), anti-HA (1:2000), anti-Flag (1:5000), anti-YAP (H-9) (Santa Cruz sc-271134, 1:1000), anti-β-actin (Sigma 1:5000, A2228), anti-TEAD4 (Abcam ab58310, 1:1000), goat anti-rabbit (Thermo 31460, 1:4000), and goat anti-mouse (Thermo 31430, 1:4000).

## BioID

Cells expressing TEAD4-BirA were labeled with biotin (Sigma B4639, 500 μM) for 6 min at 37 °C, and then lysed on ice for 30 min with NETN buffer supplemented with micrococcal nuclease (NEB M0247S, 1:5000). Subsequently, cell lysate was centrifuged at 12,000 rpm for 10 min at 4 °C to collect supernatant. This supernatant was incubated with streptavidin beads (Smart-life-sciences SA021005) for 2 h at 4 °C. The beads were washed with NETN buffer 3 times and then used as samples for label-free quantitative mass spectrometry analysis (Majorbio, Shanghai, China).

## siRNA screening

siRNA reagents were obtained from the Dharmacon Human ON-TARGETplus siRNA Smartpool Library (Thermo) as described previously (Tang et al, 2020). For screening of TFs based on TEAD4 condensate, individual siRNAs targeting 192 BioID-identified genes were automatically added to black-walled 384-well plates to a final concentration of 25 nM. Lipofectamine RNAiMAX reagent (Thermo) at a volume of 0.05 μl per well was then added to the siRNA-loaded plates. Subsequently, cells expressing GFP-TEAD4 were seeded into the plate at 2000 cells per well. After

subsequent transfection for 48 h, the cells were fixed with 4% PFA at room temperature for 30 min. DAPI was used to identify individual nuclei. The plates were kept at 4 °C in the dark before being analyzed using the Opera system. Twenty focus fields per well were scanned and analyzed. Scrambled siRNA was used as an internal reference on each plate.

For screening based on cell growth, HGC-27 cells (2000 per well) were seeded into a 96-well plate overnight. The individual siRNAs targeting 192 BioID-identified genes were then added into each well to a final concentration of 25 nM. After subsequent incubation for 48 h, the cells were analyzed using an ATP-based CellCounting-Lite™ 2.0 reagent (Vazyme, Nanjing, China) according to the manufacturer's instructions. The intracellular ATP contents were measured using an EnSight™ Multimode Microplate Reader (Perkin Elmer).

## RT-PCR and RNA sequencing

For RT-PCR, total RNA was isolated from cells using RNA Isolator Total RNA Extraction Reagent (Vazyme) according to the manufacturer's instructions. The isolated RNA was reverse transcribed into cDNA using HiScpt II (Vazyme). Quantitative RT-PCR was performed using a CFX96™ Real-Time system (Bio-Rad) and SYBR Green PCR master mix (Yeasen). The fold change in the gene expression was calculated using the comparative Ct method, and three replicates were tested for each cDNA sample. *ACTB* was used as an internal reference. Transcript copies of TEAD4 were determined by generating standard curves using plasmid DNA (TEAD4). Each experiment was repeated at least three times. The sequences of the primers are listed in Appendix Table S1.

For RNA sequencing, HGC-27 cells in 6-well plates were treated with GLUP peptide at 10 μg/ml for 48 h. The stocking buffer (20 mM HEPES pH 7.5, and 100 mM NaCl) was used as a negative control. Total RNA was extracted from at least two biological replicates. Extracted RNA was monitored for its quality using a 2100 Expert Bioanalyzer (Agilent) before being sent for library preparation and RNA sequencing (Majorbio). The RNA sequencing data were analyzed on a free online Majorbio I-Sanger Cloud Platform (www.majorbio.com).

## Apoptosis detection

Apoptosis was detected using an Annexin V-PE Apoptosis Detection Kit (Beyotime, C1065L) and carrying out flow cytometry according to the manufacturer's protocol. In brief, cells were digested with 0.25% trypsin (without EDTA), and then centrifuged at $500 \times g$ for 5 min at 4 °C. The supernatant was discarded, and cell pellets were washed with PBS 3 times. Subsequently, the cells were resuspended in 500 μL binding buffer and then supplemented with 5 μl of annexin V labeled with PE. The supplemented resuspended cells were incubated at room temperature in the dark for 10 min, and then analyzed using an LSRFortessa™ flow cytometer (BD Bioscience).

## In vitro DSS-cross-linking assay

The deployed cross-linking reaction system contained 20 mM HEPES pH 7.5, 250 mM NaCl and 5 μM tested proteins with a total volume of 20 μl. This system was allowed to incubate for 30 min at 30 °C, and

then combined with disuccinimidyl suberate (DSS) at a final concentration of 3 mM, and the mixtures were further incubated at 4 °C for 20 min. The cross-linking reaction was stopped by adding 1 M Tris HCl (pH 8.5) to the incubated mixture at 4 °C, and allowing the mixture to sit for 10 min, and then boiling it in 1×SDS loading buffer to provide samples for SDS-PAGE analysis.

## Immunoelectron microscopy

Cells were fixed by adding to them a mixture of 4% PFA and 0.1% glutaraldehyde (GA) in 400 mM HEPES buffer (pH 7.4) at 4 °C overnight. The fixative was replaced by washing the cells with 0.15 M glycine in PBS 3 times for 5 min each to quench free aldehyde. Cells were then scraped and subjected to centrifugation in order to collect them in 1% gelatin. Pellets were embedded in 12% gelatin, solidified in ice and cut into 1 mm$^3$ blocks on a cold plate. The small blocks were immersed in 2.3 M sucrose and the immersion was rotated at 4 °C overnight and transferred to an aluminum specimen holder and frozen in liquid nitrogen. Sections (70 nm) of the transferred material were cut at −115 °C using an ultramicrotome (UC7, Leica) and mounted on 100-mesh formvar/carbon-coated nickel grids. These cryosections were immune-labeled with the primary antibody (anti-TEAD4 antibody, CST #13295, 1:50) and secondary antibody (protein A-conjugated 10-nm gold particles, Cell Microscopy Center, University Medical Center Utrecht). After being labeled, the sections were counter-stained with methylcellulose/uranyl acetate for 7 min, and imaged using an electron microscope (FEI, Tecnai G2 Spirit) at 120 kV.

## Chromatin conformation capture (3C) assay

The 3C assay was performed following a protocol described previously (Hagege et al, 2007). Briefly, cells were incubated with 0.05% trypsin for 10 min before detaching. Collected cells were cross-linked with 2% formaldehyde for 10 min, followed by treatment with 0.125 M glycine for 5 min to quench the cross-linking reaction. Cells were then pelleted and incubated with a lysis buffer (10 mM Tris-HCl pH 7.5, 10 mM NaCl, 5 mM MgCl$_2$ and 0.1 mM EGTA) at 4 °C for 15 min. Resulting pelleted nuclei were digested with Hind-III restriction enzyme (NEB) at 37 °C overnight and subsequently ligated with T4 DNA ligase (NEB) at 16 °C for 4 h. Thereafter, the samples were incubated at 65 °C overnight to de-crosslink the protein-DNA complexes and were subjected to phenol-chloroform extraction and ethanol precipitation to obtain the 3C templates. Primers flanking the Hind-III restriction sites located close to TEAD4-targeted gene *CYR61* (Xie et al, 2019) or *MYC* (Zanconato et al, 2015) promoter (anchor primers) and enhancers were used to quantitively measure the promoter-enhancer interactions. The primers used are listed in Appendix Table S1.

## ChIP-qPCR

ChIP was carried out following a procedure previously described (An et al, 2018). Briefly, cells were first cross-linked with freshly prepared formaldehyde (final concentration 1.42%) for 15 min, followed by addition of glycine (125 mM) for 5 min at room temperature. After a two-round wash with cold PBS, cells were scraped and collected by centrifuge. Resulting pelleted cells were resuspended in 400 μl of ChIP lysis buffer (50 mM HEPES/KOH, pH 7.5; 140 mM NaCl; 1 mM EDTA; 1% Triton X-100; 0.1% Na-deoxycholate and protease

inhibitors) and subjected to sonication with Bioruptor at high power for 15 cycles (30 s on and 30 s off for each cycle) to shear the chromatin. Samples were then further diluted twice with lysis buffer and the diluted samples were centrifuged to clear the supernatants. Meanwhile, eighty microliters of each of the supernatants were directly processed to extract total DNA as whole-cell input. The remaining respective portions of the supernatants were transferred to new Eppendorf tubes and were incubated with either IgG or antibodies including those against TEAD4 (13295, CST), histone H3 (ab176842, Abcam), H3K27me3 (9733S, CST), H3K27ac3 (8173S, CST) and CTCF (07-729, Millipore) at 4 °C overnight. Prewashed protein A/G beads (L2118; Santa Cuz) were added into these samples, which were then incubated for another 3 h. These incubated samples were washed five times with ChIP lysis buffer and then mixed with 100 μl of 10% chelex (1421253; Bio-Rad), followed by being boiled for 10 min and centrifuged at 4 °C for 1 min. Resulting supernatants were transferred to new tubes. Subsequently, another 120 μl of MilliQ water were added to each fraction of beads pellet, and the resulting mixtures were vortexed for 10 s, and then centrifuged again to spin down the beads. The supernatants were combined as templates for follow-up qPCR analysis.

The primers used in this study included
*CDK6* Forward: TACTCTGGCGCTTTGTTGTG,
*CDK6* Reverse: CGCTGTAGGTAGCAGAGGT;
*CCNA2* Forward: ACAGAAGGGGAGCGACTGG,
*CCNA2* Reverse: CCCACCGTTTTCACTTTTTC;
*CGB5* Forward: CGAGGGGTGAGCAATACTTCA,
*CGB5* Reverse: CCTCGAAGCAGGTGACAAAG;
*CGB7* Forward: CTCCTGCATTTCCAGGCGA,
*CGB7* Reverse: TCCCTAGGATGCGAAAGCTC;
*CTGF* Forward: TCTGTGAGCTGGAGTGTG, and
*CTGF* Reverse: GCCAATGAGCTGAATGGAGT.

## Electrophoretic mobility shift assay (EMSA)

The purified proteins and the triple-tandem repeat sequence of M-CAT DNA (5′-TTGCATTCCTCTC-3′) were incubated at 4 °C for 1 h with a molar ratio of 1:1 in a binding buffer (20 mM HEPES pH 7.5, 250 mM NaCl). Each reaction mixture contained 2.9 μg protein and DNA respectively. Samples were then loaded onto a 0.8% polyacrylamide gel, which was cast and run in a buffer of 25 mM Tris with 250 mM glycine at room temperature. The nucleic acids were detected using DNAGreen dye.

## Dynamic light scattering (DLS)

Purified TEAD4 protein was mixed with VGLL1/4 or their mutant proteins or glup at a molar ratio of 10:1 at 4 °C. In the case of glup, the mixtures were incubated at 4 °C for various periods of time as indicated in related figures. And then the mixtures were subjected to dynamic light scattering analysis at 25 °C. Scattered light was measured at a 90° angle at a wavelength of 658 nm.

## Chemical synthesis of the linear peptide

The linear peptide glup was synthesized by carrying out standard Fmoc-based solid-phase peptide synthesis using Rink Amide resin. The assembly of each amino acid included Fmoc deprotection, coupling of the amino acid, and capping of the unreacted amino group.

For Fmoc deprotection, the resin was treated with 20% piperidine in N,N-dimethylformamide (DMF) (5 min x2). For coupling of the amino acid, the resin was treated with a reaction mixture containing N,N'-diisopropylcarbodiimide (4.0 eq.), oxyma (4.0 eq.), and Fmoc-protected amino acid (4.0 eq.) at 55 °C for 40 min. For capping of the unreacted amino group, the resin was treated with Ac$_2$O/2,4-lutidine/DMF (5/6/89) at room temperature for 2 min. At the end of each step, the resin was thoroughly washed with DMF. After the completion of the assembly of glup, a freshly prepared trifluoroacetic acid (TFA) cocktail (TFA/water/m-cresol/TIS = 88/5/5/2, v/v) was added to the pre-dried resin. After 2 h, pre-chilled ether was added to the collected TFA mixture to precipitate the crude peptide. The peptide glup was confirmed from the results of MALDI-TOF-MS analysis.

### Head-to-side-chain cross-linking of peptide

To achieve head-to-side-chain cross-linking of glup, first freshly prepared 1,3-dibromomethylbenzene (1.4 mg, 5.43 μmol) in 0.2 ml dimethylsulfoxide (DMSO) was added to the reaction solution. The pH value of the resulting solution was quickly adjusted to 8.0 with 1 M NH$_4$HCO$_3$ (confirmed using pH test paper). After letting this solution sit for 1 h at room temperature, the head-to-side-chain cross-linking reaction was completed as confirmed from the results of an analytical high-performance liquid chromatography (HPLC). After subjecting the solution to centrifugation, the resulting supernatant was purified by subjecting it to semi-preparative HPLC to give the desired head-to-side-chain cross-linked peptide GLUP. The remaining precipitate (containing a large amount of GLUP) was redissolved into a small amount of TFA, and then this solution was combined with pre-chilled ether. The resulting mixture was subjected to centrifugation, and the resulting precipitate was collected and immediately dissolved in DMSO and purified using semi-preparative HPLC as soon as possible. The sequence of GLUP was confirmed from the results of MALDI-TOF-MS analysis, and lyophilized into a white powder.

### ChIP-Seq

ChIP-Seq was performed based on a previous protocol but with minor modifications (Meng et al, 2021; Wal and Pugh, 2012). Briefly, HGC-27 cells cultured with normal or glucose-deprived medium for 6 h were then resuspended in 400 μL of ChIP digestion buffer (20 mM Tris HCl, pH 7.5; 15 mM NaCl; 60 mM KCl; 1 mM CaCl$_2$ and protease inhibitors). To shear chromatin, cells were digested with the proper amount of micrococcal nuclease (MNase, M0247S, NEB) at 37 °C for 20 min to ensure that most of the chromatin was mono- and di-nucleosomes. The reaction was stopped with 2 × stop buffer (100 mM Tris HCl, pH 8.1; 20 mM EDTA; 200 mM NaCl; 2% Triton X-100; 0.2% Na-deoxycholate and protease inhibitors). Samples were further sonicated using Bioruptor at high power for 15 cycles (30 s on and 30 s off for each cycle). Sonicated samples were subjected to centrifugation, and soluble chromatin was immunoprecipitated with ChIP-grade antibodies against H3K9me3 (ab8898, Abcam), histone H3 (ab176842), H3K27me3 (9733S, CST), H3K27ac (8173S, CST) and CTCF (07-729, Millipore) at 4 °C overnight together with prewashed protein A/G beads. Subsequent samples were eluted and reverse cross-linked in an elution buffer (10 mM Tris HCl, pH 8.0; 10 mM EDTA; 150 mM NaCl; 5 mM DTT and 1% SDS) at 65 °C

overnight. After subjecting these eluted and reverse cross-linked samples to sequential digestion with DNase and Proteinase K, resulting DNA was purified using the PCR purification kit B518141 (Sangon Biotech). DNA samples from three immunoprecipitations were pooled to generate libraries using the Ovation Ultra-Low Library Prep kit (NuGEN) according to the manufacturer's instructions. Sequencing was performed using an Illumina HiSeq 2500 platform (Shbio, Shanghai).

### Cut&Tag

The Cut&Tag assay kit (TD-904, Vazyme) was used to decipher the TEAD4-chromatin binding profile in WT or TEAD4KO HGC-27 cells treated with or without GLUP. Briefly, about $1 \times 10^5$ cells were suspended in 100 μl of wash buffer and incubated with concanavalin A-coated magnetic beads for 1 h to tether the cells onto the beads. The bead-bound cells were then resuspended in 50 μl of antibody buffer with 1 μg anti-TEAD4 antibody (sc-101184, Santa Cruz) or IgG control antibody (ab171870, Abcam) at 4 °C with slow rotation overnight. Subsequently, the bead-bound cells were treated with 50 μl of secondary antibody diluted with the dig wash buffer for another hour at RT. After being washed with the dig wash buffer three times to remove unbound antibodies, the cells were further incubated with the hyperactive Pa-Th5 transposase adaptor complex (TD-903, Vazyme) to digest host genome and obtain the fragmented DNA. The fragmented DNA was purified using the phenol-chloroform-isoamyl alcohol extraction and ethanol precipitation. The above-described ChIP-Seq-derived libraries were amplified by mixing 15 μl of the DNA with 5 μl of a universal N5XX and uniquely barcoded N701 primer (TD-202, Vazyme) followed by incubating each mixture for 15 cycles of 98 °C for 10 s and 60 °C for 5 s, and then 1 cycle of 72 °C for 1 min, and finally hold at 4 °C. The size distribution of the libraries was examined by carrying out an Agilent 2100 analysis. Sequencing was performed using the Illumina HiSeq X-ten platform (Shanghai Biotechnology Co., China). Sequencing raw reads were preprocessed by filtering out sequencing adapters, short-fragment reads and other low-quality reads. Bowtie (version 0.12.8) was then used to map the clean reads to the human hg19 reference genome. Corresponding peak detection was performed using SEACR_1.3 (https://github.com/FredHutch/SEACR). The protein-binding motifs were identified using HOMER software (http://homer.ucsd.edu/homer/).

### Cell viability

For cell viability assays, cells were seeded into 96-well plates overnight to attach them to the plates, and then treated with GLUP for 48 h. Cells incubated for indicated periods of time were analyzed using an ATP-based CellTiter-Lumi™ Plus kit (Beyotime) according to the manufacturer's instructions. The intracellular ATP contents were measured using a BioTek Synergy™ NEO multi-detector microplate reader (Thermo). Cell viability was calculated using the equation cell viability (%) = [[value (test) − value (blank)] × [value (control) − value (blank)]] −1 × 100.

### RNA-scope

RNA-scope was performed according to manufacturer's protocols (RNAscope Red Detection Kit, ACDbio, 322360). Briefly, the tumor

sections (3 μm thickness) were obtained from MNU-generated gastric tumor tissues. The sections were deparaffinized, retrieved, and then hybridized with *Ctgf* probes following the manufacturer's instructions (ACD, Hayward, CA, USA). Subsequently, the signal of hybridized probes was further visualized using RED dye-labeled probes. Finally, the samples were redyed, this time with hematoxylin, and mounted with VectaMount Permanent Mounting Medium (Vector Laboratories, Burlingame, CA, USA), and photographed using a Nikon Upright Fluorescence Microscope (Nikon, Japan). Positive and negative control probes were used for each experiment to help derive conclusions from the data.

## GC patient specimens

The patients included in the study provided written informed consent for the use of specimens obtained from them. The studies were performed in accordance with the Declaration of Helsinki and approved by the Huashan Hospital Institutional Review Board (HIRB), Fudan University (Approval No. 2017-222).

## Patient-derived organoids (PDOs)

To establish PDOs of human GC, surgically fresh tumor samples resected from GC patients were immediately put into RPMI 1640 medium at 4 °C, and minced finely with scalpels, and then digested with collagenase II (sigma), Y-27632 (Sigma, 10 μM) and Primocin (Invivogen, 1:500) at 37 °C for 8 min. After three washes, tumor cells were resuspended in Matrigel (Coring, #354234) and plated in small drops on a 24-well culture plate. These drops were solidified at 37 °C for 20 min before overlaying them with organoid medium containing advanced DMEM/F12 (GIBCO) with 100 U/ml penicillin/streptomycin (GIBCO), Wnt-3a (Proteintech, 50 ng/ml), Noggin (Peprotech, 100 ng/ml), R-spondin1 (Peprotech, 500 ng/ml), 1× B27 supplement (GIBCO), 1× N2 supplement (GIBCO), 1 mM N-acetylcysteine (Invitrogen), 10 mM nicotinamide (Sigma-Aldrich), 50 ng/ml human recombinant EGF (Peprotech), 100 ng/ml FGF10 (Peprotech), 500 nM A83-01 (R&D Systems), 1 μM SB202190 (Sigma), 10 μM Y-27632, and 10 nM gastrin (Sigma). Until they were cryopreserved, organoids were cultured in an incubator at 37 °C under normoxic conditions with 5% $CO_2$ in the presence of 1:500 Primocin. Medium was refreshed every 3–4 days and organoids were passaged every 1–2 weeks at a 1:2 to 1:5 split ratio. For passaging, medium was removed, and then organoids were resuspended in 1 ml/well TrypLE (GIBCO) and incubated for 5 min at 37 °C.

## Patient-derived xenografts (PDXs)

To establish PDX models, cell suspensions from GC patients were prepared as they were for PDOs. Subcutaneous implantation was performed on one flank of 5-week-old SCID mice weighing from 16 to 18 g and that were anesthetized using a isoflurane/oxygen mixture. Tumor growth was monitored by taking biweekly measurements of the implantation sites. Serial engraftments of each tumor were performed when the tumor reached a volume of ~500 mm³. Subsequently, PDX tumors were enzymatically disassociated to form a single-cell suspension. To reduce murine cell populations, the cell suspension was then subjected to magnetic mouse cell depletion using a MACS mouse cell depletion kit (Miltenyi Biotech, Bergisch Gladbach, Germany). Following the cell depletion, all established PDX cell lines were maintained at steady rates of proliferation in RPMI 1640 medium. Additionally, a small piece of tumor tissue was obtained using a scalpel and fixed in 10% formalin-buffered saline for histological analysis.

## Pharmacokinetics assay

Blood samples were obtained at 0, 0.17, 0.33, 0.5, 0.67, 0.83, 1, 2, 4, 6, 8, 10, 12, and 24 h after intraperitoneally injected administration of FAM-GLUP. These blood samples were then allowed to coagulate at room temperature for 4 h, and then were subjected to centrifugation at $3000 \times g$ at 4 °C for 10 min in order to separate the plasma. The plasma samples were analyzed using an Ultra Performance Liquid Chromatography system (UPLC, Waters, Milford, MA). Pharmacokinetic parameters were calculated using Phoenix™ WinNonlin® software (version 6.1.0.173; Pharsight Corp., Mountain View, CA).

## Toxicity analysis

Healthy BALB/c mice (~8 weeks, $n = 5$) were injected intraperitoneally with GLUP (1, 10 mg/kg, respectively, every 2 days), and mice of the control group were injected with saline (10 ml/kg, $n = 5$). The mice were continuously observed for 14 days after the administration and then sacrificed for routine blood testing (of white blood cells, red blood cells, platelets, lymphocytes, monocytes, and neutrophils), examination of the blood biochemistry (hemoglobin, albumin, glucose, creatinine, urea, AST, and ALT) and pathology examination.

## Animals

All animals were housed under specific pathogen-free conditions in ventilated cages with automated watering and on a 12-h light/dark cycle and handled in accordance with the guidelines of the Institutional Animal Care and Use Committee of the Institute of Biochemistry and Cell Biology. The approval ID for the use of animals was SHDSYY-2023-P0011 issued by the Animal Core Facility of Shanghai Tenth People's Hospital.

Mice used in this study were from SLAC Laboratory Animal (Shanghai, China), and included 4-week-old BALB/c nude mice (female) and 8-week-old C57BL/6J mice (male and female). Tumors were allowed to grow to an average size of ~60 mm³ and then the mice were randomly allocated to groups of 5–6 animals. No blinding was used in the treatment schedules for these experiments since the different treatments were identified by different marks on the tail. Based on our previous experience, 5–6 animals per group would provide statistically significant data while keeping the number of animals used to a minimum. Tumor size was determined by taking caliper measurements of tumor length, width and depth; tumor volume was calculated as 0.5236 × length × width × depth (mm³).

## GC model

Healthy female BALB/c nude mice were maintained in pathogen-free conditions to which they were allowed to acclimate for 1 week before being used. For the tumor formation assay, tumor cells ($1 \times 10^6$) were injected into the respective flanks of the mice. Mice in which tumors were detected with an average size of ~60 mm³ were biweekly

administered 1 or 5 mg/kg GLUP via subcutaneous injection. These mice were sacrificed after 4 weeks, and tumor volumes were then measured.

For the model involving N-nitroso-N-methylurea (MNU)-induced GC, 8-week-old C57BL/6 mice received 3 cycles of MNU (TRC, Toronto, Canada) in drinking water. For each cycle, drinking water containing 240 µg/ml MNU was served to the mice for 1 week, and then normal drinking water the next week. After 19 weeks of MNU administration, mice were randomized to receive GLUP via intraperitoneal injection (*i.p.*) biweekly for the following 5 weeks (*n* = 8/group). After 24 weeks, mice were sacrificed for subsequent analysis. The tumor volume per stomach was measured.

## Data availability

The source data of this paper are collected in the following database record: BioImages accession number S-BIAD1333. All the sequencing data in this study have been deposited and available in the Gene Expression Omnibus database: RNA-Seq data: GSE244417, GSE255045 and GSE255046. ChIP-Seq data: GSE244418 and GSE244419. Cut&Tag data: GSE255047 and GSE274201.

The source data of this paper are collected in the following database record: biostudies:S-SCDT-10_1038-S44318-024-00257-4.

## Peer review information

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

## Acknowledgements

Our work was supported by the National Key R&D Program of China (2020YFA0803200 and 2023YFC2505903), National Natural Science Foundation of China (32270747, 32200567, 31930026, 82150112, 92168116, 22077002, 82222052, 32070710, 82372613, 82361168638, 32170706, 82002493), Science and Technology Commission of Shanghai Municipality (22ZR1448100, 22QA1407200, 22QA1407300, 23ZR1480400, 23YF1432900, 23ZR1448900), and Shanghai Super Postdoctoral Incentive Program. We thank the staff of the Chemical Biology Core Technology Facility of Center for Excellence in Molecular Cell Science for their assistance with the high-throughput screening assay.

## Author contributions

**Yang Tang**: Data curation; Software; Funding acquisition; Validation; Methodology; Writing—original draft; Writing—review and editing. **Fan Chen**: Data curation; Software; Validation; Investigation; Methodology; Project administration. **Gemin Fang**: Resources; Data curation; Validation; Investigation; Methodology. **Hui Zhang**: Validation; Visualization; Methodology. **Yanni Zhang**: Methodology. **Hanying Zhu**: Data curation; Methodology. **Xinru Zhang**: Data curation; Software; Methodology. **Yi Han**: Validation; Writing—review and editing. **Zhifa Cao**: Software; Methodology. **Fenghua Guo**: Resources; Software. **Wenjia Wang**: Resources; Data curation; Methodology. **Dan Ye**: Conceptualization; Resources. **Junyi Ju**: Methodology. **Lijie Tan**: Resources; Investigation. **Chuanchuan Li**: Resources; Software. **Yun Zhao**: Conceptualization; Investigation; Project administration. **Zhaocai Zhou**: Conceptualization; Resources; Funding acquisition; Methodology; Writing—original draft; Writing—review and editing. **Liwei An**: Resources; Data curation; Funding acquisition; Investigation; Methodology; Writing—original draft; Writing—review and editing. **Shi Jiao**: Conceptualization; Data curation; Supervision; Funding acquisition; Investigation; Visualization; Methodology; Writing—original draft; Project administration; Writing—review and editing.

Source data underlying figure panels in this paper may have individual authorship assigned. Where available, figure panel/source data authorship is listed in the following database record: biostudies:S-SCDT-10_1038-S44318-024-00257-4.

## Disclosure and competing interests statement

ZZ, SJ, YT, GF, WW, and YZ have filed a patent (202111647677.0) for the use of engineered linker peptide in treating gastric cancer.

