## [Peer Review File · The EMBO Journal]

A cofactor-induced repressive type of transcription factor condensation can be induced by synthetic peptides to suppress tumorigenesis

Zhaocai Zhou, Yang Tang, Fan Chen, Gemin Fang, Hui Zhang, Yanni Zhang, Hanying Zhu, Xinru Zhang, Yi Han, Zhifa Cao, Fenghua Guo, Wenjia Wang, Dan Ye, Junyi Ju, Lijie Tan, Chuanchuan Li, Yun Zhao, Liwei An, and Shi Jiao

Corresponding author(s): Zhaocai Zhou (zczhou@sibcb.ac.cn) , Shi Jiao (jiaoshi@fudan.edu.cn), Liwei An (lwan@tongji.edu.cn)

Review Timeline:

Submission Date:	2nd Sep 23
Editorial Decision:	3rd Nov 23
Revision Received:	7th Mar 24
Editorial Decision:	6th May 24
Revision Received:	7th Aug 24
Editorial Decision:	15th Aug 24
Revision Received:	23rd Aug 24
Accepted:	17th Sep 24

Editor: Ieva Gailite

Transaction Report:

Dear Zhaocai,

Thank you for providing a preliminary revision plan for your manuscript. Based on the overall interest expressed in the referee reports and your willingness to engage in a major revision as expressed during the pre-decision consultation, I would like to invite you to address the comments of all reviewers in a revised version of the manuscript.

We generally allow three months as standard revision time. As a matter of policy, competing manuscripts published during this period will not negatively impact on our assessment of the conceptual advance presented by your study. However, please contact me as soon as possible upon publication of any related work to discuss the appropriate course of action. Should you foresee a problem in meeting this three-month deadline, please let us know in advance in order to arrange an extension.

When preparing your letter of response to the referees' comments, please bear in mind that this will form part of the Review Process File and will therefore be available online to the community. For more details on our Transparent Editorial Process, please visit our website: <https://www.embopress.org/page/journal/14602075/authorguide#transparentprocess>. Please also see the attached instructions for further guidelines on preparation of the revised manuscript.

Please feel free to contact me if have any further questions regarding the revision. Thank you for the opportunity to consider your work for publication, and I look forward to receiving the revised manuscript.

With best wishes,

leva

leva Gailite, PhD
Senior Scientific Editor
The EMBO Journal
Meyerhofstrasse 1
D-69117 Heidelberg
Tel: +4962218891309
i.gailite@embojournal.org

Please remember: Digital image enhancement is acceptable practice, as long as it accurately represents the original data and

conforms to community standards. If a figure has been subjected to significant electronic manipulation, this must be noted in the figure legend or in the 'Materials and Methods' section. The editors reserve the right to request original versions of figures and the original images that were used to assemble the figure.

We realize that it is difficult to revise to a specific deadline. In the interest of protecting the conceptual advance provided by the work, we recommend a revision within 3 months (1st Feb 2024). Please discuss the revision progress ahead of this time with the editor if you require more time to complete the revisions.

Referee #1:

In this manuscript, Tang et al. identify the VGLL4 and TEAD4 interaction as a driver of repressive complexes in liquid-liquid phase separation (LLPS). Furthermore, they identify GLUP as a peptide that can glue together TEAD4 and VGLL4 to drive LLPS of repressive complexes. To this end, they use a combination of several methods, such as biophysical methods, functional genomics, as well as mouse and PDX models.

While the discovery of the GLUP compound, as well the discovery of LLPS-driven TEAD4/VGLL4 complexes, would certainly be of great interest to many researchers in the Hippo field, I am not yet convinced that this compound is really specific. In addition, sometimes the results of their experiments are overinterpreted, and several experiments lack essential controls and have issues with regard to statistics (low number of replicates, incorrect statistical test).

Specific points are:

Major:

1.)
Fig. 1B: The cutoffs seem arbitrary. The authors should perform a Z-score-based analysis for the analysis.

2.)
Fig. 2: The way the authors describe the BioID experiments, the experiment was not performed under Glucose starvation?! IF this is the case, wouldn't it be counterintuitive to identify LLPS-specific TEAD4 interaction partners? The authors should repeat the BioID in two conditions: + Glucose and - Glucose to bolster their conclusions.

3.)
Fig. 2B: Here, also please provide a Z-score test to avoid arbitrary cutoffs.
Why was the HGC-27 cell line used? Is the phenotype consistent in other gastric cancer cell lines?

4.)
Fig. 2D: Please also provide an IF showing the punctae formation after VGLL4 and TEAD4 depletion. Most of the IF pictures were performed using overexpression which can produce artefacts.

5.)
Supp. Fig. 2A/B: Are the data derived from independent experiments since the error bars seem tiny. Please use a proper statistical test to assess significance. Does EMSY/LDOC1/VGLL4 knockdown affect YAP phosphorylation/localization?

6.)
Is VGLL4-dependent LLPS and repression of YAP targets TEAD4-dependent? The authors should analyse VGLL4-mediated punctae formation in a TEAD4-deficient background.

7.)
Fig. 3C: I am not an expert on LLPS but would this finding not be against the concept of LLPS. Wouldn't one also expect to find many other proteins in condensates? A Hexanediol control would be needed to bolster the point that this is due to LLPS. The "smear" the authors refer to, I really cannot see.

8.) The CoIPs should also be performed with the VGLL4mut with just one TDU.

9.)
Fig. 4A,B: To demonstrate that binding of TEAD4 to its cognate binding site is involved in the process proposed by the authors.

10.)

Fig. 4I: Since the authors demonstrate a colocalization with H3K27me3, this modification should also be analysed by ChIP.

11.)

To demonstrate that this phenomenon is specific for TEAD binding sites, the authors should perform more unbiased analysis, e.g. CUT&RUN for H3K27me3/ac, TEAD, YAP, PolII +/- Glucose starvation. Especially to foster their conclusions which they depict in Fig. 4; Do they imply that TEAD always binds as a homodimer to DNA? That would be very surprising.

12.)

Fig. 5C,E: A negative control is missing to demonstrate that this is not simply aggregation due to GLUP's/glup's biophysical properties, ideally using a TEAD4 or a point mutant that (based on the authors' molecular docking analyses) does not bind to GLUP/glup anymore.

13.)

Fig. 6A: Why does GLUP not lead to punctae formation?

14.)

What is the concentration (in μM) of GLUP used for all the assays. Since I do not have (or overlooked) the MW of this compound, please always add this information in the respective figure legend. The duration of the treatment should also be indicated in the figure legend.

15.)

Having looked at the ChIP-Seq bigwig files, the TEAD4 ChIP-Seq data are of very poor (unpublishable) quality. I could not identify a single peak close to well established YAP targets: e.g. AMOTL2, CTGF, CYR61 or others. The authors should provide quality metrics for their ChIP-Seq data: FRiP, enrichment of TF motifs in peaks and should perform triplicates since they only performed a single ChIP-Seq per sample for TEAD4.

Fig. 6D:

In addition, they have to use spike-in controls which would allow them drawing quantitative conclusions from their ChIP-Seq. The effects of GLUP on H3K27me3 should be performed in a TEAD4 KO cell line. Furthermore, the effect on H3K27me3 should be stratified based on the presence/absence of TEAD4 binding to demonstrate a specific effect and rule out a pleiotropic mode of action of GLUP. Seeing a global impact on H3K27me3 is rather worrisome, as it implies to me that GLUP is not specific. Besides that, by simply analysing the TSS, the authors will miss most of the TEAD binding sites since several labs (e.g. Piccolo) demonstrated a preferred binding of TEAD to enhancers. In general, the ChIP-Seq analysis needs to be elaborated in much more detail.

16.)

RNA-Seq analysis (Supp. Fig. 6).

The authors only give the p-values for the RNA-Seq Experiment, but in a RNA-Seq experiment these are pretty meaningless due to the multiple testing issue. Please provide p-adjusted or FDR metrics. Besides that, the RNA-Seq was only performed in duplicates, please perform at least triplicates using a TEAD4 KO as a control for GLUP specificity.

17.)

I noticed that YAP1 as well as TEAD1/TEAD4 are among the most strongly downregulated genes after GLUP treatment in their RNA-Seq. This raises the possibility that the effect are rather mediated via YAP downregulation than GLUP acting on TEAD4. It is imperative that the authors address this issue, as it may also affect the conclusions of their mouse experiments.

18.)

According to Supp. Fig. 6K the plasma levels of GLUP in the mouse are around $0.15\mu\text{g/ml}$ max. The in vitro experiments, however, were mostly performed at $10\mu\text{g/ml}$ - meaning around 100x higher concentrations. This is worrisome regarding the interpretation of the in vitro experiments and how well one can transfer these findings to the in vivo mechanisms. Does GLUP also work at $0.1\mu\text{g/ml}$ in vitro? If not, I have my doubts that the in vivo data can be explained by their in vitro results.

19.)

Fig. 7A-F: Normally, YAP inhibition in vitro does not kill cancer cells. Why would a TEAD inhibitor do that?

Since GLUP should TEAD4 into a repressor, can the sensitivity be reduced by YAP (5SA) overexpression. Since mostly YAP is deregulated in human cancers, this also has therapeutic implications.

20.)

Fig. 7d: What about YAP target genes? The authors need to demonstrate that GLUP is on target here.

21.)

Fig. 7H: This figure requires an unbiased quantification.

22.)

Authors are encouraged to have the manuscript proofread for English.

Minor:

1.)

Fig. 1B: It would be easier for the reader to also put Spot area/number as a title.

2.)

Fig. 1B: What about the other TEAD proteins. Where do they score?

3.)

Fig. 1G: "these TEAD4 condensates turned out to be larger in YAP KO cells regardless of". Please provide a quantification to substantiate this claim. Also provide an experiment where YAP is added back in the KO cells since the size differences could also be due to clonal variation.

4.)

Sup. Fig. 3B,D: A +Hexandediol control is missing since this should disrupt/prevent the interaction.

5.)

The authors state

"By serendipity, we found this grouping were apparently associated with the differentiation status of these GC cell lines, i.e. e., GCs in the TEAD4 high group were poorly differentiated, a phenotype shown to indicate that they are more aggressive Fig. 7 C)."

How do the authors come to this conclusion?

Referee #2:

I have previously reviewed the same manuscript at another journal. It seems that this manuscript is the same as the one submitted previously to the another journal. None of the previous comments have been addressed. My comments and suggestions remain pretty much the same:

Transcriptional enhanced associate domain (TEAD) transcription factors are crucial for development, cell division, tissue homeostasis, and regeneration. Due to its strong association with clinicopathological features in human malignancies, deregulation of the TEAD transcriptional output plays significant roles in tumor progression. It also functions as a predictive biomarker. According to reports, TEADs integrate with transcription co-factors such YAP/TAZ and VGLLs to mediate gene transcriptions. The VGLL family, which also comprises VGLL1-4, contains the transcriptional cofactor known as Vestigial Like Family Member 4 (VGLL4). VGLL4 was described as a new tumor suppressor that contains two TDU motifs, unlike the other VGLL family members. Through interactions with transcription factors with TEA domains (TEAD), VGLL4 uses two TDU domains to execute its biological function. TEAD4 was identified as a glucose-deprivation-induced phase-phase separation transcription factor in the paper that Zhaocai Zhou et al. submitted. The formation of TEAD4 oligomers may be triggered by VGLL4, and the formation of the TEAD4-VGLL4 complex functions as a transcriptional repressor and prevents the expression of the target genes. Additionally, in order to strongly drive TEAD4's repressive condensation against YAP-induced transcriptional activation, the authors created a linker peptide that mimics VGLL4 to preferentially "glue" TEAD4 molecules together. In tumor models created using xenografts, the glue peptide demonstrated potent anti-tumor activity. The studies seems like to provide a new mechanistic model regarding the counteractive regulation of TEAD-VGLL4 against TEAD-YAP through modulating TEAD phase-phase separation. The roles of TEAD-VGLL4 in the regulation of the hippo pathway and cancer are well established, yet the majority of physiological and signaling pathway research are conventional. The authors should take into account a number of additional issues when revising the manuscript.

1, In Figure 1C, the authors used puncta area and puncta number to overlap the most significant proteins involved in the phase-phase separation, it would be more biologically sound to use replicated experiments for overlapping.

2, The authors overexpressed 255 transcription factors in 293FT cells and observed the formation of condensates, are these transcription factors endogenously expressed in 293FT cells or cancer cells to exhibit their transcriptional regulations?

3, In Figure 1G, the authors found enlargement of the TEAD4 condensates in YAP KO cells, how about TAZ, does it show

similar effects as TAZ could also undergo phase-phase separation?

4, The authors discovered that YAP reduces TEAD4 condensates whereas VGLL4 promotes their formation. Since it is widely known that YAP and VGLL4 compete for TEAD4 binding, the authors should talk about what makes the current study innovative in this regard.

5, In Figure 3C, looks like glucose deprivation increased the formation of both TEAD4 monomer and dimer in Native-Page, while the total proteins were not changed in denatured SDS-PAGE, what causes the discrepancy?

6, In Supplemental Figure 3D, it doesn't sound like VGLL1 mutant binds more to TEAD4, given the input of VGLL1 mutant is much higher than wild-type? Maybe VGLL1 mutant is more stable? The authors can repeat this experiment.

7, In the studies, the authors mostly used HEK293 cell line to perform the experiments. Because the HEK293TF cell line is an embryonic kidney cell, which cannot phenotypically represent cancer cell lines, the authors' use of HEK293TF cells under glucose restriction to simulate the cancer microenvironment does not seem credible. Several important investigations may be replicated by the author utilizing tested cancer cell lines.

Referee #3:

In the manuscript "Identification of a repressive type of TF condensation and engineering of a glue peptide therapy against cancer" the authors design a synthetic peptide that convincingly multimerizes TEAD4 transcription factor and show that this peptide can inhibit gastric cancer growth in a mouse model, xenograft model and patient-derived organoids. I have reviewed this identical manuscript for a different journal previously and maintain my opinion that, while this is an interesting finding and the data supporting these findings is solid, the first part of the manuscript describing two classes of TEAD4 condensates is lacking. I have listed set of issues with the condensate part of this manuscript below. While the findings of this manuscript are timely and very interesting, I can only support publication when the issues with the condensate part are sufficiently addressed.

Specific comments:

- Introduction suggests a single role of phase separation during tumorigenesis, while there are probably many different roles of a process as general as phase separation
- "liquidity and dynamics" - please change to material properties.
- Fig 1 What is being tested by using glucose deprivation? How does it represent both the tumor-micro-environment and cell death-related stress? Is glucose deprivation necessary, since the authors focus on TEAD4 which forms condensates upon over-expression even in the absence of glucose deprivation. The authors need to explain the choices for the conditions of their screen more clearly.
- Fig 1 over-expression of TF will be sufficient to induce puncta (evidence in 1D:CDX1), therefore the condensates that are found here need to be confirmed under physiological conditions. Physiological confirmation of TEAD4 (or other) condensates is completely absent from the manuscript, leaving the possibility that all observations pertain over-expression artifacts. This can be achieved by endogenous fluorescent labeling of TEAD4 or immunofluorescence on fixed cells.
- Fig1B/C - please quantify condensed fraction instead of size and number.
- Fig1 D Hexanediol disruption is not proof of LLPS, many other parameters need to be evaluated to be able to support LLPS (Alberti, Gladfelter, Mittag Cell 2019). Live cell imaging of cells with endogenously labeled TEAD4 (previous comment) would allow for extensive FRAP, droplet fusion and dissolution (in response to glucose restoration?). These experiments are needed to support endogenous condensate formation of TEAD4.
- Fig1 D Lack of puncta in Hexanediol treated cells says nothing about material properties. What is the conclusion "gel-like transition" based on? Material properties can be evaluated by comparing FRAP kinetics and immobile fraction across conditions. These data are necessary to claim "gel-like transition".
- Fig 1E Why focus on TEAD4 out of the screen? Can the authors justify? Alternatively, the authors can keep the screen from confounding the message of this manuscript and just focus on TEAD4 because of its importance in cancer and the previously reported observation that YAP/TAZ in the Hippo pathways form condensates.
- Fig 1E YAP/TAZ/TEAD4 form condensates under osmotic stress. Is glucose deprivation a different type of stress that induces Hippo condensates?
- Fig 1F "glucose deprivation induces fluid-to-gel transition" To support this claim +glucose transition FRAP data displaying faster recovery kinetics is needed.
- Fig 1F 20 sec recovery is slow and on average there is a 40% immobile fraction. How liquid are these compared to YAP/TAZ condensates (Cai et al Nature Cell Biol 2019)?
- Fig S1B what is the time frame of the reduction after glucose supplementation?
- Fig1G and S1C-D Data shown here shows that there is absolutely no difference in TEAD4 puncta area or number between WT

and YAPKO cells. The difference is purely caused by glucose withdrawal. This is evident from both the displayed images and the quantification and is diametrically opposed to the conclusion in the text.

- Fig S2AB please provide a statistical test.
- Fig 2D shows large TEAD4 puncta while glucose present, why is this? If glucose withdrawal is not necessary to observe the condensates, why is the screen in Fig 1 performed under glucose withdrawal?
- Fig 2F please provide quantification
- Fig S2F please provide quantification
- Fig 2 apoptosis data, purified droplet assays are missing for RFXANK. To support the claims made by the authors these data are also necessary.
- What do the authors mean with "homo-oligomers" while there are at least two different proteins in there? Please explain.

- Figure 3 why do the authors start using glucose deprivation again? TEAD4 puncta are observed with over-expression alone. Please justify this decision.
- Fig 3C oligomerization combined with slow recovery kinetics point to the direction of gel formation, not LLPS. However LLPS is suggested as the mechanism.
- Fig 3C authors note a smear that is visible in the figure.
- Fig3C authors describe enhanced TEAD4 aggregates of high molecular mass, which is hard to appreciate without quantification. Furthermore, the increase intensity of the TEAD4 monomer and dimer bands indicate that the general level of TEAD4 is increased, not specifically the high molecular mass aggregates. This means the increased protein levels of TEAD4 are likely driving the increased condensate formation upon glucose withdrawal, not TEAD4 oligomerization. Can the authors eliminate the effect of TEAD4 over-expression as a cause for oligomerization?
- Fig 3H clearly shows dimers, but not oligomers? Is the observed effect just explained by two binding sites on VGLLx?

- Figure 4. Short linear DNA may be incorporated as a client, but long chromosomal DNA cannot permeate condensates. The in vitro experiment with short DNA segments is therefore not representative of the in vivo situation. Longer DNA can better mimic the endogenous conditions, for example lamda-DNA.
- Fig 4A this assay cannot distinguish between binding and client retention
- Fig 4C why does TEAD4:DNA form droplets? Multiple TEAD4 binding sites on the DNA? If it can be nucleated, maybe TEAD4 has a high threshold concentration? Can the authors perform titration of TEAD4 in a droplet formation assay?
- Fig 4D its unclear what is imaged here. Are the used DNA molecules 39bp long?
- Fig 4F There appears to be no difference between the conditions.
- Fig 4G unclear what is being shown here.
- Fig 4H Contrary to the data in Fig 2D TEAD4 now can form condensates while treated with Hexanediol. Can the authors explain?
- Fig 4H,L and S4A-C are all individual examples, the authors need to quantify this colocalization over multiple cells and TEAD4 condensates.

- Fig 5 there must be additional interactions between GLUP-mediated TEAD4 dimers in order to have oligomers. How can the authors explain the oligomerization? What parts of TEAD4 are responsible for this?

- Fig 6 these results can be explained by wholesale TEAD4 aggregation in the nucleus and complete genomic rearranging, leading to a toxic effect. Unrelated to "repressive TEAD4 condensates".
- Fig 6C supports gross chromatin deformation as a mechanism.

Referee #4:

Tang et al. screened the condensation formation of over 700 transcription factors in cells with glucose deprivation and found that TEAD4, among other transcription factors, can form condensates. They further identified association factors of TEAD4 by Bio-ID and found that VGLL4 is an inducer of condensate formation of TEAD4. They further found a peptide that can induce TEAD4 oligomerization to form condensates. The peptide has a strong antitumor effect via inhibition of TEAD4 related gene transcription. These results are interesting and exciting. However, whether these condensates are formed in physiological conditions are questionable. Additionally, the insufficient description of methods and figures prevents the reviewer from assessing the quality of data. I would recommend the manuscript for publication if concerns were addressed.

Essential revisions

The authors should provide evidence that endogenous VGLL4 and TEAD4 form condensates and colocalize with each other. Crucially, depletion of VGLL4 in cells should abolish the condensation formation of TEAD4.

Major points

1. 1,6-Hex is not a reagent that can test whether proteins undergo LLPS.
2. The authors state that "Notably, no obvious puncta were observed in 1,6 Hex treated cells, suggesting that the TF condensates may undergo a gel like transition in glucose deprived cells." This statement is incorrect.
3. Previous studies have shown that YAP can condense TEAD4 condensates; however, the author's studies did not support this. What are the discrepancies among these experiments?
4. TEAD4 can be partitioned into YAP-condensates or VGLL4 condensates. What is the potential mechanism for this selectivity?
5. The authors state that "Transitions from liquid to gel like states in proteins are often facilitated by weak multivalent interactions between proteins". There is no such data to support this transition.
6. The authors used N-STORM to image DNA in cells overexpressing VGLL4 and then claimed there are differences in these cells. Based on my examination, these images are similar. The authors should develop a metric to quantify these differences if they believe these images are different.
7. The quality of images of co-immunostaining of H3K27me3 and H3K27Ac with GFP-TEAD4 is low. High-quality images should be helpful.
8. The length of DNA sequences is less than 40 bp, which is under the resolution of optic microscopy (confocal). Could the authors explain how they measure the morphology of DNA in vitro?
9. I would recommend that the authors use condensed fraction to describe the condensation capacity of proteins, which allows a systematic comparison among transcription factors and between different conditions.
10. Figure 1F, why are the FRAP results normalized to 240 rather than 1?
11. Figure 1H, I would suggest that the authors label YAP and TEAD4 with fluorescence protein or fluorescence dye, which facilitates the quantification. The authors should quantify the phase separation ability.
12. Figure 2F, the authors should label VGLL4 and TEAD4 with respective fluorescence dyes. I would suggest that the authors vary the concentrations of TEAD4 and test how TEAD4 impacts the condensation capacity of VGLL4.
13. Figure 2E, again, why are the FRAP results normalized to 240 rather than 1?
14. Figure 3B, the authors should quantify the colocalization.
15. Figure 3, the current in vitro cross-linking data suggests that TEAD4 forms dimer rather than oligomer. Other alternative mechanisms may exist for the formation of TEAD4 condensates by VGLL4. For instance, an alternative mechanism is scaffold-client one.
16. Figure 3J and 4C, again, fluorescent images should be given.
17. Figure 4D, a quantitative description of compaction should be shown. Three-color images should be shown with proteins labelled with different fluorescent proteins or dyes.

Minor point

1. Sanger Cloud Platform (www.sanger.ac.uk) is not accessible.

A Point-by-point Response to Reviewers' Comments

We thank the reviewers for the constructive comments, which we found very helpful in improving and strengthening our work. Accordingly, we performed additional experiments and analyses to address these major issues: 1) Quantification for LLPS and detailed information for methods (Figure R5, R18, R25, R26, R34, R45 and R51); 2) ChIP-Seq and RNA-Seq (Figure R12-15, R19-20); 3) Endogenous TEAD4 condensation analysis with its specific antibody (Figure R30); 4) Fluorescent-labelled droplet formation assay (Figure R51).

Page 02-22, Reviewer #1

Page 22-27, Reviewer #2

Page 28-44, Reviewer #3

Page 45-55, Reviewer #4

Referee #1 (Report for Author)

In this manuscript, Tang et al. identify the VGLL4 and TEAD4 interaction as a driver of repressive complexes in liquid-liquid phase separation (LLPS). Furthermore, they identify GLUP as a peptide that can glue together TEAD4 and VGLL4 to drive LLPS of repressive complexes. To this end, they use a combination of several methods, such as biophysical methods, functional genomics, as well as mouse and PDX models.

While the discovery of the GLUP compound, as well the discovery of LLPS-driven TEAD4/VGLL4 complexes, would certainly be of great interest to many researchers in the Hippo field, I am not yet convinced that this compound is really specific. In addition, sometimes the results of their experiments are overinterpreted, and several experiments lack essential controls and have issues with regard to statistics (low number of replicates, incorrect statistical test).

We thank this reviewer for the constructive comments. Following the suggestions, we have provided more supporting data to show the specificity of the GLUP peptide, as well as the essential controls to improve the statistical analysis in our revised manuscript. Meanwhile, we also double-checked our wording throughout the manuscript to avoid overinterpretation.

Specific points are:

Major:

1) Fig. 1B: The cutoffs seem arbitrary. The authors should perform a Z-score-based analysis for the analysis.

Following the reviewer's suggestion, we performed Z-score-based analysis of the condensed fraction instead of spot size and number to compare fluorescent spot formation following the methodology previously described (Brown et al., 2023). Typically, the activity cut-off value is chosen by 3-fold of the standard deviation of the normalized activity of all samples (Chen, 2010). Therefore, the results showed that seven transcriptional factors including TEAD4, EWSH, RFXDC1, GTF2A1L, C16orf5, ZNF800 and ELF1 were identified as top hits forming increased condensates in response to glucose withdrawal (**Figure R1**).

We have put this data in the revised manuscript (Figure 1B).

Figure R1. Z-score calculation of condensed fraction for each TF in HEK293FT cells upon glucose starvation. The cutoff value was set as 3-fold of standard deviation.

2) Fig. 2: The way the authors describe the BioID experiments, the experiment was not performed under Glucose starvation?! IF this is the case, wouldn't it be counterintuitive to identify LLPS-specific TEAD4 interaction partners? The authors should repeat the BioID in two conditions: + Glucose and - Glucose to bolster their conclusions.

Thanks for pointing this out. We believe that a repeated BioID assay under glucose starvation is unnecessary under this condition. The rationale is this: we first screened for transcription factors that may form condensates upon glucose starvation, a condition mimicking tumor microenvironment. After we found that TEAD4 is one of the transcription factors that can form condensation upon glucose starvation, we then performed BioID, a method allowing for identification of transient protein-protein interactions, to search for potential factors that can directly trigger enhanced TEAD4 condensation. Moreover, we subsequently performed a siRNA-based screening assay to examine the potential effect of each candidate upon glucose starvation-induced TEAD4 condensation. Combing these two strategies, we thus identified VGLL4, as well as RFXANK, as such regulatory partners.

To further support this notion, we have alternatively performed Co-immunoprecipitation (Co-IP) assay to examine the interaction of VGLL4 or RFXANK with TEAD4 with or without glucose deprivation. As shown below, we reproducibly detected the interaction of VGLL4 or RFXANK with TEAD4 (**Figure R2**). Moreover, these interactions were dramatically enhanced upon glucose deprivation, which are consistent with their abilities to promote TEAD4 condensation (**Figure R2**).

Figure R2. Co-IP analysis of the interaction between TEAD4 and RFXANK or VGLL4 in HEK293FT cells treated with or without glucose limitation.

3) Fig. 2B: Here, also please provide a Z-score test to avoid arbitrary cutoffs.

Agree! Using the same methodology as described above, we also performed Z-score calculation to compare the condensed fraction of TFs (**Figure R3A**) and cell growth (**Figure R3B**) between samples. Accordingly, we identified 20 hits whose depletion via siRNAs resulted in decreased TEAD4 condensates in glucose-deprived cells (**Figure R3A**). Meanwhile, there are 7 hits whose knockdown increased the viability of HGC-27 cells (**Figure R3B**). Taken together, we found that top 7 hits responsible for both TEAD4 condensation and cell viability were VGLL4, ARID3B, RFXANK, YY1, CTCF, EMSY, and LDOC1 (**Figure R3C**).

We have put these data into the revised manuscript (Figures 2B-C).

Figure R3. Z score analysis of siRNA screening results in TEAD4 condensed fraction and cell growth. (A) Z score analysis of siRNA screening results in condensed fraction. The cutoff value was <-3 fold and highlighted in red dots. (B) Z score analysis of siRNA screening results in cell growth. The cutoff value was >3 fold and highlighted in red dots. (C) Venn diagram of combined analysis revealed 7 candidates whose depletion impair TEAD4 condensates formation but enhance cell viability.

Why was the HGC-27 cell line used? Is the phenotype consistent in other gastric cancer cell lines?

We understand the reviewer is concerned with the representativity and the utility of the HGC-27 cell line as a model system. The HGC-27 cell line is originally derived from undifferentiated gastric cancer tissue and has endured as a representative GC cell line. Meanwhile, our experimental evidence also showed that the expression of YAP and TEADs in this cell line is relative higher (Jiao et al., 2014; Tang et al., 2020) (**Figure R4**). Therefore, we chose the HGC-27 cell line to explore the related phenotype in this study. In addition to HGC-27, we also compared the GLUP sensitivity in a panel of GC cell lines such as AZ-521 and MKN-45 (original Figure 7C) and PDO tissues (original Figure 7G). Taken together, these data demonstrated the repressive role of TEAD4 LLPS as well as the robustness of our targeting strategy.

Figure R4. Reported protein levels of YAP, TEAD4 and VGLL4 in different gastric cancer cell lines. (A) Protein levels of YAP and VGLL4 were determined by western blotting against special antibody. (B) Protein expression levels of STRN3 and YAP in cells from 13 GC cell lines and 1 noncancerous gastric epithelial cell line. (C) Protein levels of TEAD4 in 10 cell lines.

4) Fig. 2D: Please also provide an IF showing the punctae formation after VGLL4 and TEAD4 depletion. Most of the IF pictures were performed using overexpression which can produce artefacts.

Following this advice, we have conducted the IFA assay to observe the exogenously expressed GFP-TEAD4 condensate formation in HEK293FT cells pre-treated with indicated siRNAs under glucose starvation condition. In line with the notion that both

VGLL4 and RFXANK promotes TEAD4 condensation, we found that depletion of VGLL4 or RFXANK markedly reduced the abilities of TEAD4 condensates formation in response to glucose starvation (**Figure R5**). We have updated this data in the revised manuscript (Supplementary Figure 2D).

Figure R5. Condensate formation in HEK293FT cells transfected with GFP-TEAD4 were treated with individual siRNAs under glucose deprivation. (A) Realtime-PCR (RT-PCR) analysis of knockdown efficiencies of indicated 2 siRNAs in HEK293FT cells using *GAPDH* as an internal control (n = 3/group). The data were analyzed using one-way ANOVA, followed by the Tukey’s post-hoc test. ****, p < 0.0001. (B) HEK293FT cells transfected with GFP-TEAD4 were treated with individual siRNAs for 48 hr and then treated with glucose deprivation for another 12 hr. After treatment, the condensates of cells were imaged and analyzed. Quantification of TEAD4 condensed fraction is showed in the right panel. The data were analyzed using one-way ANOVA, followed by the Tukey’s post-hoc test. ****, p < 0.0001. Scale bar, 10 μm.

5) Supp. Fig. 2A/B: Are the data derived from independent experiments since the error bars seem tiny. Please use a proper statistical test to assess significance.

Many thanks! We now thoroughly went through the manuscript to ensure that all statistical tests were performed properly. The data of supplementary figures 2A/B were derived from three independent experiments. And the statistical test was added to the figures using one-way ANOVA, followed by the Tukey’s post-hoc test (**Figure R6**). We have updated the data in the revised manuscript (Supplementary Figures 2A-B).

Figure R6. Knockdown efficiencies of 7 siRNAs and mRNA levels of *CTGF* and *CYR61* in HGC-27 cells transfected with the indicated siRNAs. (A) Realtime-PCR (RT-PCR) analysis of knockdown efficiencies of indicated 7 siRNAs in HGC-27 cells using *ACTB* as an internal control (n = 3/group). The data were analyzed using one-way ANOVA, followed by the Tukey’s post-hoc test. ***, p < 0.001, ****, p < 0.0001. (B) mRNA levels of *CTGF* and *CYR61* in HGC-27 cells transfected with

the indicated siRNAs (n = 3/group). The cutoff value was >2 fold in mRNA change. The data were analyzed using one-way ANOVA, followed by the Tukey's post-hoc test. n.s., no significance; ***, p < 0.001, ****, p < 0.0001.

Does EMSY/LDOC1/VGLL4 knockdown affect YAP phosphorylation/localization?

As suggested by this reviewer, we have performed immunofluorescent and immunoblotting assays to examine the YAP phosphorylation and localization in HGC-27 cells pre-treated with individual siRNAs under glucose deprivation (**Figure R7A**). Results showed that knockdown of these factors neither affect YAP phosphorylation level (**Figure R7B**) nor its nuclear localization (**Figure R7C**) in HGC-27 cells regardless of glucose deprivation. In addition, we also examined the subcellular localization of VGLL4, EMSY and LDOC1 in [GeneCards - Human Genes | Gene Database | Gene Search](https://www.genecards.org/) which show that these proteins are all strictly nuclear-localized proteins (**Figure R7D**). Thus, considering that VGLL4 functions as a competitor of nuclear YAP for TEAD4-binding, it is highly possible that EMSY and LDOC1 function similarly as VGLL4.

Figure R7. Knockdown of EMSY/LDOC1/VGLL4 do not affect YAP phosphorylation and its nuclear localization in HGC-27 cells under glucose deprivation. (A) Realtime-PCR (RT-PCR) analysis of knockdown efficiencies of indicated 3 siRNAs in HGC-27 cells using *GAPDH* as an internal control (n = 3/group). The data were analyzed using one-way ANOVA, followed by the Tukey's post-hoc test. ****, p < 0.0001. (B) Western blot to analyze the phosphorylation level of YAP1 with indicated siRNAs treatment. (C) Representative images of YAP1 nuclear localization with indicated siRNAs treatment with or without glucose starvation. Scale bar, 10 μm. (D) Subcellular localization analysis of VGLL4, EMSY and LDOC1 in Genecards website (<https://www.genecards.org/>).

6) Is VGLL4-dependent LLPS and repression of YAP targets TEAD4-dependent? The authors should analyse VGLL4-mediated punctae formation in a TEAD4-deficient background.

Thanks for pointing this out. As reported previously, the YAP/TAZ-TEAD4 LLPS complex formation is required for downstream gene transcription and cell growth (Cai et al., 2019; Lu et al., 2020). Here we further revealed a VGLL4-mediated TEAD4 LLPS that is

repressive for gene transcription and cell death. Accordingly, we concluded that VGLL4 and YAP act in a competition manner for TEAD4-binding and subsequent formation of distinct types of condensates (original Figure 4M).

To address this reviewer's concern, we applied *in vitro* droplet formation assay to assess the VGLL4 LLPS, alone or co-incubation with TEAD4, using purified recombinant proteins. Of note, resembling the TEAD4 protein (**Figure R8**, column 1), VGLL4 protein alone does not form LLPS (**Figure R8**, column 2). However, LLPS droplets rapidly assemble upon mixing VGLL4 and TEAD4 proteins together (**Figure R8**, column 3), suggesting VGLL4 forms LLPS in a TEAD4-dependent manner.

We have updated this data in the revised manuscript (Figure 3F).

Figure R8. Droplet formation of mCherry-TEAD4 and FITC-VGLL4 proteins, alone or combined *in vitro*. Scale bar, 10 μ m.

7) Fig. 3C: I am not an expert on LLPS but would this finding not be against the concept of LLPS. Wouldn't one also expect to find many other proteins in condensates? A Hexanediol control would be needed to bolster the point that this is due to LLPS. The "smear" the authors refer to, I really cannot see.

Thanks for pointing this out. The reviewer #2 also expressed similar concern of the "smear" issue (Page 31-32). We re-performed the western blotting assay under native condition to gain insight into the assembly behavior of TEAD4 with or without glucose deprivation. First, we transfected the HEK293FT cells with Flag-TEAD4 for 24 h before subjected to 6 h of glucose deprivation, a time-period window relative longer than the primary manuscript setting, and thus observed much stronger "smear" signals of oligomerization of TEAD4 (**Figure R9A**). Moreover, we further examined the endogenous TEAD4 aggregates in HGC-27 cells under the same condition and reproducibly observed such "smear" signals upon glucose starvation (**Figure R9B**).

We have put this part of data into the revised manuscript (Figure 3C).

Figure R9. Immunoblots of endogenous TEAD4 or Flag-tagged TEAD4 in cells with or without glucose deprivation using native gel or SDS-PAGE. (A) Immunoblots of Flag-tagged TEAD4 in HEK293FT cells with or without glucose deprivation using native gel or SDS-PAGE. (B) Immunoblots of endogenous TEAD4 in HGC-27 cells with or without glucose deprivation using native gel or SDS-PAGE.

8) The CoIPs should also be performed with the VGLL4mut with just one TDU.

Agree! We now evaluated the requirement of TDU domains for VGLL4-mediated TEAD4 LLPS. As shown in Supplementary Figure 3D (**Figure R10**), our co-IP assay showed that VGLL4^{mut}, a construct in which the TDU1 (amino acid 206-229) of VGLL4 was deleted, failed to promote TEAD4 homo-association compared to the wildtype VGLL4 (right panel). More importantly, forced addition of a TDU domain to VGLL1 (referred as VGLL1^{mut}) endowed VGLL1 the ability to promote TEAD4 LLPS (left panel). Collectively, these findings confirm the necessity of two TDUs for VGLL4 in promoting TEAD4 LLPS.

Figure R10. Co-IP analysis of TEAD4 oligomerization in HEK293FT cells transfected with VGLL1, VGLL4 or their mutants. VGLL1^{mut}, a construct in which TDU1 (amino acid 206-230) of VGLL4 was added to the N-terminal of wildtype VGLL1 to create a VGLL1 version with two TDUs. VGLL4^{mut}, a construct in which TDU1 (amino acid 206-229) of VGLL4 was deleted to create VGLL4 version with one TDU.

9) Fig. 4A, B: To demonstrate that binding of TEAD4 to its cognate binding site is involved in the process proposed by the authors.

According to the reviewer's comment, we performed gel mobility shift experiment with human TEAD4 (S336A/K376A/V389A) mutant, a construct which is unable to bind VGLL4.

As showed below, TEAD4 or TEAD4^{Mut} alone could shift the DNA (**Figure R11**, line 4-5). Whereas, the migration of the DNA segments was significantly retarded in the presence of TEAD4 and VGLL4 (**Figure R11**, line 3), but not in the presence of TEAD4^{Mut} and VGLL4 (**Figure R11**, line 6). These results demonstrated that binding of TEAD4 to its cognate binding site is involved in the process of VGLL4-mediated TEAD4 LLPS.

We have put this part of data into the revised manuscript (Figure 4A).

Figure R11. EMSA analysis to detect TEAD4-DNA or TEAD4^{Mut}-DNA interaction in the presence of VGLL4. TEAD4^{Mut}: S336A/K376A/V389A of TEAD4.

10) Fig. 4I: Since the authors demonstrate a colocalization with H3K27me3, this modification should also be analysed by ChIP.

Agree! Based on our scenario, TEAD4 can form either transcriptional active LLPS (previous reported, H3K27ac-enriched and YAP-included) or transcriptional repressive LLPS (the current work, H3K27me3-enriched and VGLL4-include). As suggested by this reviewer, we performed ChIP-Seq assay using TEAD4 and H3K27ac antibodies under untreated or glucose starvation conditions. Specifically, we individually extracted 7599 and 12717 peaks from TEAD4 and H3K27ac in control untreated cells, respectively, and found that 30.3 % peaks were overlapped between TEAD4 with H3K27ac in untreated cells (**Figure R12A**), and the shared peaks between TEAD4 with H3K27ac were dramatically decreased to 5.4% in glucose-starved cells, implying a repressive role for TEAD4 LLPS in glucose-deprived cells.

To validate the ChIP-Seq results, we performed ChIP-qPCR analysis in HEK293FT cells in the presence or absence of glucose treatment using H3K27me3 antibodies. In sharp contrast to the transcriptional active marker, we observed that glucose starvation dramatically enhanced H3K27me3-binding onto the *CTGF* promoter (**Figure R12B**). Similarly, overexpression of VGLL4 phenocopied the enrichment of H3K27me3 on *CTGF* promoter, whereas further co-treatment with 1,6-Hex dramatically reduced such binding (**Figure R12C**). Taken together, these data confirmed the transcriptional repressive role for TEAD4 LLPS upon glucose deprivation or VGLL4 ectopic expression, which eventually triggers the cell death. We have put the part of data into the revised manuscript (Supplementary Figure 4D).

Figure R12. Combined analysis of TEAD4/H3K27ac ChIP-Seq under untreated or glucose starvation condition and ChIP-qPCR analysis of H3K27me3-binding on *CTGF* promoter in 293FT cells with indicated conditions. (A) Overlapping analysis of TEAD4-specific peaks in ChIP-Seq using H3K27ac antibody under untreated or glucose starvation condition. (B) ChIP-qPCR analysis of H3K27me3-binding on *CTGF* promoter in 293FT cells subjecting to glucose deprivation, or (C) VGLL4 overexpression with or without 1,6-Hex. Data represents three replicates from one experiment.

11) To demonstrate that this phenomenon is specific for TEAD binding sites, the authors should perform more unbiased analysis, e.g. CUT&RUN for H3K27me3/ac, TEAD, YAP, PolIII +/- Glucose starvation. Especially to foster their conclusions which they depict in Fig. 4;

We understand this reviewer is concerned about the contribution of transcriptional repressive TEAD4 LLPS to glucose deprivation-induced cell death, as well as the targeting specificity of our GLUP for anti-tumor therapy. First, our primary nuclear condensates screening indeed identified a panel of TFs including TEAD4 which can form condensations upon glucose starvation, indicating that such TFs-DNA LLPS may represent a general mechanism to shut-down gene transcription under stressed conditions. Based on this scenario, we further chose TEAD4 to showcase the molecular mechanism (s) and thereby revealed a competition between YAP-mediated transcriptional active TEAD4 LLPS (previous reported, H3K27ac-enriched) and VGLL4-induced transcriptional repressive TEAD4 LLPS (H3K27me3-enriched).

As suggested by this reviewer, we re-analyzed our previous data including TEAD4 and H3K27ac ChIP-Seq results, and individually extracted 7599 and 12717 peaks, and found that 30.3 % peaks were overlapped between TEAD4 with H3K27ac in control untreated cells (**Figure R13**); however, the shared peaks between TEAD4 with H3K27ac were dramatically decreased from 30.3% to 5.4% upon glucose starvation (**Figure R13**), suggesting that TEAD4 acts as the major TF in mediating transcriptional repression in glucose-deprived cells.

Figure R13. Overlapping between TEAD4 and H3K27ac in control/glucose-starved cells.

Furthermore, we also applied ChIP-Seq assay to decipher the binding signatures of TEAD4 and H3K27ac in control and GLUP-treated HGC-27 cells. To better discriminate the specificity of GLUP, we divided the peaks into TEAD4-overlapped (Specific) or not (Non-specific) for those identified from H3K27ac ChIP-seq results. Interestingly, we found that GLUP treatment dramatically reduced the H3K27ac-binding onto the TEAD4-specific target gene motifs from 22.1% (2810/9907) to 1.4% (202/14699) (**Figure R14**).

Figure R14. Peaks overlapping between TEAD4 and H3K27ac in control/GLUP-treated cells.

Moreover, we also utilized the ChIP-qPCR analysis to validate these observations in TEAD4-specific motifs such as *CTGF*, *CCNA2* and *CGB5*. Consistent with the ChIP-Seq, either glucose deprivation (**Figure R15A**) or ectopic expression of *VGLL4*/*RFXANK* (**Figure R15B**) efficiently attenuated the transcriptional active markers (CTCF, H3K27ac and RNA-Pol II), but increased the transcriptional repressive marker H3K27me3, on these genes' promoter regions.

Taken together, we concluded that the repressive type of TEAD4 LLPS could mediate glucose starvation-triggered suppression of gene transcription, and our GLUP peptide may specifically induce such type of repressive TEAD4 LLPS for cancer therapy.

Figure R15. Chromatin immunoprecipitation-quantitative PCR (ChIP-qPCR) analysis of TEAD4-specific motifs such as *CTGF*, *CCNA2* and *CGB5* under glucose deprivation or in VGLL4/RFXANK-overexpressing HEK293FT cells. (A) ChIP-qPCR analysis of TEAD4-specific motifs such as *CTGF*, *CCNA2* and *CGB5* under glucose deprivation of HEK293FT cells. (B) ChIP-qPCR analysis of TEAD4-specific motifs such as *CTGF*, *CCNA2* and *CGB5* in VGLL4/RFXANK-overexpressing HEK293FT cells. (C) Western blotting to confirm the expression of VGLL4 or RFXANK in HEK293FT cells. Data represents three replicates from one experiment.

Do they imply that TEAD always binds as a homodimer to DNA? That would be very surprising.

Not really. The DNA sequence length of the pUC-GW-Kan vector contains the inserted M-CAT triple tandem repeat sequence which can gather more than one TEAD4 molecules together. The M-CAT triple tandem repeat sequence of DNA provided multiple TEAD4 binding sites, making it possible for TEAD4:DNA to form droplets. Also, we and others has previously revealed that TEAD4 normally binds to DNA as a monomer (Shi et al., 2017) to activate gene transcription. Here, as we proposed in this work, TEAD4 can form dimers and/or oligomers in the presence of transcriptional repressor VGLL4 (Jiao et al., 2014) or glucose starvation.

12) Fig. 5C,E: A negative control is missing to demonstrate that this is bot simply aggregation due to GLUP's/glup's biophysical properties, ideally using a TEAD4 or a point mutant that (based on the authors' molecular docking analyses) does not bind to GLUP/glup anymore.

Agree! We understand this reviewer is concerned with the binding specificity of GLUP peptide. As shown in Supplementary figure 5B and 5C (**Figure R16**), the MST assay and DSS-crosslinking experiment both showed that *glup^{mut}* peptide, an interaction-dead control peptide with a sequence of 'RRVCVAAAASLSLR', could not bind TEAD4 (**Figure R16A**) and therefore failed to induce TEAD4 oligomerization (**Figure R16B**).

Figure R16. The specificity of *glup* peptide towards TEAD4. (A) Binding curves of purified TEAD4 protein with *glup* or *glup^{mut}* performed by microscale thermophoresis (MST) assay. K_d values are shown. *glup^{mut}*, an interaction-dead control peptide with a sequence of 'RRVCVAAAASLSLR'. (B). Crosslinking assay of TEAD4 oligomerization in the presence of *glup* or *glup^{mut}* after DSS treatment.

13) Fig. 6A: Why does GLUP not lead to punctae formation?

Thanks. In original figure 6A, we mainly focused on tracking the time-dependent penetration of GLUP into cancer cell. Thus, our observation time was chosen at early stage (30 min and 80 min) post GLUP treatment. Indeed, we observed that the FAM (a fluorescent moiety)-labeled GLUP could efficiently accumulate in the nucleus at 80 min post treatment (**Figure R17A**). Moreover, once the incubation time was extended to 12 hr, we could readily detect an increased TEAD4 puncta formation ability compared with

control untreated cells (**Figure R17B**).

Figure R17. Differences of condensate formation between GLUP-treated cells with different incubation times. (A) Live cell images of HGC-27 cells treated with FAM (a fluorescent moiety)-labeled GLUP for indicated periods of time. Scale bar, 10 μm . (B) Representative fluorescent images of GFP-TEAD4 condensates in HEK293FT cells treated with GLUP for 12 hr. Scale bar, 10 μm .

14) What is the concentration (in μM) of GLUP used for all the assays. Since I do not have (or overlooked) the MW of this compound, please always add this information in the respective figure legend. The duration of the treatment should also be indicated in the figure legend.

We are sorry for missing this important information. As suggested, we have provided the related information in the related figure legends and listed the concentration and treatment time of GLUP used for all cellular experiments below (**Figure R18**).

Figures	GLUP Conc.	GLUP treatment time	Cell lines
Figure 5H	5 μM	24 hr	HEK293FT
Figure 5I	5 μM	24 hr	HEK293FT
Figure 6A	5 μM	30min; 80min	HEK293FT
Figure 6C	5 μM	24 hr	HEK293FT
Figure 6D	5 μM	12 hr	HEK293FT
Figure 6E	1mg/kg; 5mg/kg	i.p. 2 times per weeks	B6 mice
Figure 6H	1mg/kg; 5mg/kg	s.c. 2times per weeks	Nude mice
Figure 6J	1mg/kg	i.p. 2 times, 15 days after GLUP treatment	stomach
Figure 6K	1mg/kg; 5mg/kg	i.p. 2 times per weeks	B6 mice
Figure 7B	1 μM	every two days for a two-week duration	HGC-27
Figure 7G	50 μM ; 250 μM	1 weeks	PDO
Figure 7H	50 μM	12 hr	PDO
Figure 7I	5mg/kg	15 days	PDC

Figure R18. List for the concentration and treatment time of GLUP used for all cellular experiments.

15) Having looked at the ChIP-Seq bigwig files, the TEAD4 ChIP-Seq data are of very poor (unpublishable) quality. I could not identify a single peak close to well established YAP targets: e.g. AMOTL2, CTGF, CYR61 or others. The authors should provide quality metrics for their ChIP-Seq data: FRiP, enrichment of TF motifs in peaks and should perform triplicates since they only performed a single ChIP-Seq per sample for TEAD4. The effects of GLUP on H3K27me3 should be performed in a TEAD4 KO cell line. Fig. 6D: In addition, they have to use spike-in controls which would allow them drawing quantitative conclusions from their ChIP-Seq.

Thanks for pointing this out. On one hand, we provided more in-depth analysis of our ChIP-Seq results. First, TF motif analysis revealed the TEAD motifs in TEAD4 ChIP-Seq results in both control and glucose-deprived cells, partly validating the feasibility of our ChIP data (**Figure R19A**). Moreover, as described above, we also analyzed the overlapped peaks between TEAD4 and H3K27ac in control and glucose-deprived cells. As shown in **Figure R12**, the shared peaks between TEAD4 and H3K27ac are highly overlapping in control untreated cells, suggesting an active state of transcription. In line with our notion that GLUP triggers repressive TEAD4 LLPS, we observed that the shared motifs were reduced upon GLUP treatment, especially for *CTGF* and *CYR61* (supplementary figure 6G) (**Figure R19B**).

Meanwhile, as suggested by this reviewer, we re-performed the TEAD4-related assay using Cut&Tag strategy in triplicates for Ctrl and GLUP treatment including a spike-in control to normalize the signals within each sample. We firstly calculated FRiP value for data quality, which showed a good quality of Cut&Tag ChIP-seq data (**Figure R19C**, FRiP value >0.03). Similar to our ChIP-Seq results, we observed the enrichment of TEAD motifs in all 6 samples (**Figure R19D**). We further analyzed the constituent peaks of TEAD4 in GLUP-treated cells, which had lower *CTGF*, *TEAD4* and *BCL2L1* signals than those in control cells (**Figure R19E**).

Regarding the specificity of GLUP, we divided the H3K27ac peaks into TEAD4-specific and non-specific groups (**Figure R19F**). Interestingly, we found that GLUP treatment dramatically reduced the H3K27ac-binding onto the TEAD4-specific motifs, but did not affect the H3K27ac-binding onto the non-specific motifs (**Figure R19G**), indicating a specific role of GLUP on TEAD4-binding gene promoters.

Figure R19. Multiple analysis of TEAD4 ChIP-Seq data. (A) Motif analysis for TEAD4 ChIP-Seq assay in control and glucose-deprived cells. (B) Integrative Genomics Viewer (IGV) snapshot depicting ChIP-Seq signal of the indicated genes in GLUP-treated cells. Signals are plotted on a normalized read per million (RPM) bases. (C) FRiP value for TEAD4 ChIP-Seq assay using CUT&TAG strategy for triplicates with control or GLUP treatment. (D) Motif analysis for TEAD4 ChIP-Seq assay using CUT&TAG strategy for triplicates with control or GLUP treatment. (E) IGV showing the single peak for *CTGF*, *TEAD4* and *BCL2L1*. (F) Venn analysis of H3K27ac-binding onto the TEAD4-specific motifs upon GLUP treatment. (G) Heatmap for H3K27ac-binding onto the TEAD4-specific or non-specific motifs upon GLUP treatment. (H) Heatmap for H3K27me3-binding

onto the TEAD4-specific or non-specific motifs upon GLUP treatment.

Furthermore, the effect on H3K27me3 should be stratified based on the presence/absence of TEAD4 binding to demonstrate a specific effect and rule out a pleiotropic mode of action of GLUP. Seeing a global impact on H3K27me3 is rather worrisome, as it implies to me that GLUP is non-specific. Besides that, by simply analysing the TSS, the authors will miss most of the TEAD binding sites since several labs (e.g. Piccolo) demonstrated a preferred binding of TEAD to enhancers. In general, the ChIP-Seq analysis needs to be elaborated in much more detail.

Agree! Following this reviewer's suggestion, we divided the H3K27me3 peaks into TEAD4-specific and non-specific groups. Interestingly, we found that GLUP treatment dramatically reduced the H3K27ac-binding onto the TEAD4-specific motifs, but barely affected those non-specific motifs (**Figures R19F-G**). Similarly, the H3K27me3 signals were only increased in TEAD4-specific motifs upon GLUP treatment but not in non-specific motifs (**Figure R19H**), again suggesting a specific role of GLUP on TEAD4-binding gene promoters.

For the detailed analysis of ChIP-Seq, we now provided TF motif analysis and Peaks overlap analysis as described above (**Figures R19C-G**). Overall, these analyses support the notion that the repressive type of TEAD4 LLPS could mediate glucose starvation-triggered suppression of gene transcription, and our GLUP peptide can specifically induce such type of repressive TEAD4 LLPS for cancer therapy.

16) RNA-Seq analysis (Supp. Fig. 6). The authors only give the p-values for the RNA-Seq Experiment, but in a RNA-Seq experiment these are pretty meaningless due to the multiple testing issue. Please provide p-adjusted or FDR metrics. Besides that, the RNA-Seq was only performed in duplicates, please perform at least triplicates using a TEAD4 KO as a control for GLUP specificity.

Thanks. As suggested by this reviewer, we re-analyzed the RNA-Seq data and provide p-adjusted volcano plot (**Figure R20A**). Next, to address GLUP specificity, we performed RNA-Seq experiment in triplicates in WT and TEAD4-depleted HGC-27 cells treated with or without GLUP. The PCA plot shows apparent difference between GLUP-treated and Ctrl groups when WT cells were used; whereas no dramatic difference was observed between Ctrl and GLUP-treated groups when TEAD4-deficient cells were used (**Figure R20B**). Meanwhile, the heatmap showed that GLUP treatment reduced a greater number of gene transcription in WT cells than that of TEAD4-depleted cells (**Figure R20C**). Specifically, there were more TEAD4-specific genes were reduced upon GLUP treatment in WT (232 genes) than those in TEAD4-depleted cells (175 genes, **Figure R20D**). Also, we chose AGS, a human GC cell line with a low level of TEAD4 expression for further confirmation. AGS cells overexpressing TEAD4 but not an empty vector clearly became sensitive to GLUP as shown by their significantly reduced viability (**Figure R20E**), results confirming the target specificity of GLUP on TEAD4.

Figure R20. Analysis of RNA-Seq data with TEAD4-depleted HGC-27 cells treated with or without GLUP. (A) Volcano plot of altered genes in HGC-27 cells treated with GLUP. Red dots represent up-regulated genes while blue ones represent down-regulated genes. (B) PCA analysis of WT and TEAD4-depleted HGC-27 cells treated with or without GLUP in HGC-27 cells. (C) Heatmap of WT and TEAD4-depleted HGC-27 cells treated with or without GLUP in HGC-27 cells. (D) Overlapping analysis of indicated groups. (E) Cell viability of TEAD4-overexpressing AGS cells treated with GLUP for 48 h.

17) I noticed that YAP1 as well as TEAD1/TEAD4 are among the most strongly downregulated genes after GLUP treatment in their RNA-Seq. This raises the possibility that the effect are rather mediated via YAP downregulation than GLUP acting on TEAD4. It is imperative that the authors address this issue, as it may also affect the conclusions of their mouse experiments.

Thanks for pointing this out. We re-analyzed the RNA-Seq data and found the expression of *YAP1* and *TEADs* had no statistically significant changes (**Figure R21A**), but sample reproducibility was relatively poor due to the large variation which may mislead this reviewer to such impression. We re-performed the RNA-Seq experiment in triplicates in HGC-27 cells treated with or without GLUP (**Figure R20B**). We thus re-analyzed the tpm value for *YAP1* and *TEADs* and reproducibly found no significant changes of these genes upon GLUP treatment (**Figure R21B**). Collectively, these results verified the specificity and efficiency of GLUP on targeting TEAD4 LLPS but not protein levels.

Figure R21. Relative expression levels of *YAP1*, *TEAD1*, *TEAD2*, *TEAD3*, *TEAD4* in two RNA-Seq experiments with GLUP treatment. (A) Relative expression levels of *YAP1*, *TEAD1*, *TEAD2*, *TEAD3*, *TEAD4* of the RNA-Seq data in our original manuscript. (B) Relative expression levels of *YAP1*, *TEAD1*, *TEAD2*, *TEAD3*, *TEAD4* of our newly-performed RNA-Seq. n.s., no significance.

18) According to Supp. Fig. 6K the plasma levels of GLUP in the mouse are around 0.15µg/ml max. The in vitro experiments, however, were mostly performed at 10 µg/ml - meaning around 100x higher concentrations. This is worrisome regarding the interpretation of the in vitro experiments and how well one can transfer these findings to the in vivo mechanisms.

We tried to address the reviewer's concern from three aspects. Firstly, due to the different experimental metrics, different doses of GLUP were administered. For pharmacokinetic assay, healthy mice were administrated with 10 mg/kg (~75 µg/ml) GLUP to monitor the plasma concentration of GLUP at continuous times. For cellular assays, the concentration of GLUP in cultured medium is 10 µg/ml (**Figure R22A**). Moreover, initial administrated concentration of GLUP is about 500-fold higher than the initial free plasma concentration of GLUP. Secondly, the GLUP peptide must enter the nucleus to induce TEAD4 condensates in cells. In the live imaging of figure 6A (**Figure R22B**), we can see that the majority of GLUP adhere to cell membrane. Due to the low efficiency of cell permeability of peptide and two membrane penetrations (cell membrane and nuclear membrane) of GLUP, the concentration of GLUP in the nucleus is much less than the treatment dose in cultured medium. Thirdly, even a low concentration of GLUP (0.1 µg/ml) can also inhibit GC cell growth (**Figure R22C**).

Does GLUP also work at 0.1µg/ml in vitro? If not, I have my doubts that the in vivo data can be explained by their in vitro results.

Indeed, cell viability assay showed nearly 60% cell death after 48 h of treatment with 0.1 µg/ml GLUP (**Figure R22C**).

Figure R22. Experiments to illustrate same mechanism for *in vitro* and *in vivo* assays. (A) List for concentration of GLUP in *in vitro* and *in vivo* assays. (B) Representative live cell images of HGC-27 cells treated with FAM (a fluorescent moiety)-labeled GLUP for indicated periods of time. FAM-GLUP: 10 µg/ml (5 µM). Scale bar, 10 µm. (C) Cell viability of HGC-27 cells treated with indicated concentrations of GLUP for 48 hr.

19) Fig. 7A-F: Normally, YAP inhibition *in vitro* does not kill cancer cells. Why would a TEAD inhibitor do that?

Generally, as YAP is not the only co-factor to drive TEAD-dependent transcriptional activity, YAP inhibition could not fully phenocopy TEAD inhibition in terms of target gene expression and suppression of cancer cell growth. As for GLUP, targeting TEAD4 itself, causes aggregation of TEAD4 and shut down gene transcription, thereby inducing cell death.

Since GLUP should TEAD4 into a repressor, can the sensitivity be reduced by YAP (5SA) overexpression. Since mostly YAP is deregulated in human cancers, this also has therapeutic implications.

Thanks for pointing this out! We previously demonstrated that the active YAP (5SA) mutant can attenuate VGLL4-mediated TEAD4 LLPS and its colocalization with H3K27me3 (original Figure 4K-4L), suggesting a competition between YAP and VGLL4 in inducing different types of TEAD4 LLPS. Nevertheless, as our GLUP design was based on the binding region within TEAD4 to cause TEAD4 aggregation in a manner independent of YAP, thus it is highly possible that YAP could not affect such GLUP-induced TEAD4 LLPS. To validate this, we performed cell viability assay in GLUP-treated HGC-27 cells with or without YAP (5SA) overexpression. Results showed that GLUP can efficiently suppress HGC-27 cancer cell growth regardless of YAP (5SA) expression (**Figure R23**). Thus, we concluded that YAP activity should not affect the therapeutic role of the GLUP.

Figure R23. Cell viability assay in 0.1/1 µg/ml GLUP-treated HGC-27 cells with or without YAP (5SA) overexpression. Data are presented as means ± SD. The data were analyzed using one-way ANOVA, followed by the Tukey’s post-hoc test. n.s., no significance; ***, $p < 0.001$, ****, $p < 0.0001$.

20) Fig. 7d: What about YAP target genes? The authors need to demonstrate that GLUP is on target here.

We have performed the QPCR assay to detect the YAP target genes in GLUP-treated AGS cells transfected with empty vector (e.v.) or TEAD4 plasmid. When treated with GLUP,

TEAD4-transfected cells showed much lower levels of transcription of target genes *CTGF* and *CYR61*, compared to that of e.v.-transfected cells (**Figure R24**). In another word, GLUP targets TEAD4 to inhibit downstream gene expression.

Figure R24. mRNA levels of *CTGF* and *CYR61* in TEAD4-overexpressing AGS cells treated with GLUP for 48 h. Data are presented as means \pm SD. Significance was tested using unpaired t test, **, $p < 0.01$; ***, $p < 0.001$.

21) Fig. 7H: This figure requires an unbiased quantification.

We now provided the quantification for the figure (**Figure R25**) and updated the figure in revised manuscript (Figure 7H).

Figure R25. Immunofluorescence of β -catenin and TEAD4 in PDOs treated with 0.1 mg/ml GLUP. Scale bar, 10 μ m. The quantification is showed in the right panel. Data are presented as means \pm SD. Significance was tested using unpaired t test, ****, $p < 0.0001$.

22) Authors are encouraged to have the manuscript proofread for English.

We now proofread the manuscript carefully and corrected the wrong spelling.

Minor:

1) Fig. 1B: It would be easier for the reader to also put Spot area/number as a title.

Agree! As the condensed fraction can be calculated to evaluate the phase separation potential of candidate proteins, we put condensed fraction instead of spot area/number as title.

2) Fig. 1B: What about the other TEAD proteins. Where do they score?

We re-analyzed our screening data to obtain Z-score for TEAD proteins. The Z score for TEAD1-4 is 0.08, 2.62, 0.38 and 3.26, respectively. We now provided the score data in the supplementary table 1.

3) Fig. 1G: "these TEAD4 condensates turned out to be larger in YAP KO cells regardless of". Please provide a quantification to substantiate this claim. Also provide an experiment where YAP is added back in the KO cells since the size differences could also be due to clonal variation.

Following reviewer's suggestion, we have performed the rescue assay in the YAP KO cells and added the quantification for the images. The TEAD4 condensates is the same in YAP KO cells as WT cells under glucose deprivation condition. When we overexpressed YAP in YAP KO cells, the TEAD4 condensates were not increased by YAP overexpression (**Figure R26**). So, we concluded that YAP could not affect TEAD4 condensates in glucose deprivation condition. We have put this data into the revised manuscript (Figure 1F).

Figure R26. Fluorescent images of GFP-TEAD4 condensates in wild type (WT) or YAP knockout (YAP KO) cells rescued by YAP overexpression with or without glucose limitation for 12 h. Scale bar, 10 μ m. The quantification is showed in the right panel. Data are presented as means \pm SD. The data were analyzed using one-way ANOVA, followed by the Tukey's post-hoc test. n.s., no significance; ****, $p < 0.0001$.

4.) Sup. Fig. 3B,D: A +Hexandediol control is missing since this should disrupt/prevent the interaction.

We performed IP assay to determine TEAD4 oligomerization using 1,6-hexandediol as a control. The results showed that the oligomerization of TEAD4 in glucose starvation (**Figure R27A**) or in the presence of VGLL4/VGLL1^{mut} was abrogated by 1,6-Hex treatment (**Figure R27B**). We have put these data into the revised manuscript (Supplementary Figures 3B and 3D).

Figure R27. Glucose deprivation or VGLL4 promotes TEAD4 oligomerization. (A) TEAD4 oligomerization between HA-TEAD4 and Flag-TEAD4 in cells of glucose limitation with or without 1,6-Hex at indicated time points measured by Co-IP assay. (B) Co-IP analysis of TEAD4 oligomerization in HEK293FT cells transfected with VGLL1, VGLL4 or their mutants with or without 1,6-Hex. VGLL1^{mut}, a construct in which TDU1 (amino acid 206-230) of VGLL4 was added to the N-terminal of wildtype VGLL1 to create a VGLL1 version with two TDUs. VGLL4^{mut}, a construct in which TDU1 (amino acid 206-229) of VGLL4 was deleted to create VGLL4 version with one TDU.

5) The authors state "By serendipity, we found this grouping were apparently associated with the differentiation status of these GC cell lines, i.e. e., GC s in the TEAD4 high group were poorly differentiated, a phenotype shown to indicate that they are more aggressive Fig. 7 C)." How do the authors come to this conclusion?

This conclusion was drawn from the related background information for each cell line used in our studies. For example, HGC-27 has been defined as undifferentiated (Leiharer et al., 2021); and MKN-45 has been defined as poorly differentiated (Wang et al., 2017) and MGC-803 as low differentiated (Wu et al., 2011). Therefore, we concluded that GCs in the TEAD4 high group such as HGC-27 were poorly differentiated. We now incorporated this information in the revised manuscript. Thanks!

Referee #2 (Report for Author)

I have previously reviewed the same manuscript at another journal. It seems that this manuscript is the same as the one submitted previously to another journal. None of the previous comments have been addressed. My comments and suggestions remain pretty much the same:

Transcriptional enhanced associate domain (TEAD) transcription factors are crucial for development, cell division, tissue homeostasis, and regeneration. Due to its strong association with clinicopathological features in human malignancies, deregulation of the TEAD transcriptional output plays significant roles in tumor progression. It also functions as a predictive biomarker. According to reports, TEADs integrate with transcription co-factors such YAP/TAZ and VGLLs to mediate gene transcriptions. The VGLL family, which also comprises VGLL1-4, contains the transcriptional cofactor known as Vestigial Like Family Member 4 (VGLL4). VGLL4 was described as a new tumor suppressor that contains two TDU motifs, unlike the other VGLL family members. Through interactions with transcription factors with TEA domains (TEAD), VGLL4 uses two TDU domains to execute its biological function. TEAD4 was identified as a glucose-deprivation-induced phase-phase separation transcription factor in the paper that Zhaocai Zhou et al. submitted. The formation of TEAD4 oligomers may be triggered by VGLL4, and the formation of the TEAD4-VGLL4 complex functions as a transcriptional repressor and prevents the expression of the target genes. Additionally, in order to strongly drive TEAD4's repressive condensation against YAP-induced transcriptional activation, the authors created a linker peptide that mimics VGLL4 to preferentially "glue" TEAD4 molecules together. In tumor models created using xenografts, the glue peptide demonstrated potent anti-tumor activity. The studies seems like to provide a new mechanistic model regarding the counteractive regulation of TEAD-VGLL4 against TEAD-YAP through modulating TEAD phase-phase separation. The roles of TEAD-VGLL4 in the regulation of the hippo pathway and cancer are well established, yet the majority of physiological and signaling pathway research are conventional. The authors should take into account a number of additional issues when revising the manuscript.

We thank the reviewer for the constructive comments. Following this reviewer's suggestions, we have provided additional evidence to strengthen our work.

1, In Figure 1C, the authors used puncta area and puncta number to overlap the most significant proteins involved in the phase-phase separation, it would be more biologically sound to use replicated experiments for overlapping.

Agree! Actually, the reviewer #1 also suggested us to perform an unbiased analysis (Page 2-3). To address this issue, we performed Z-score calculation of condensed fraction instead of spot size and number to compare fluorescent spot formation between samples as previously described (Brown et al., 2023). In this way, transcriptional factors TEAD4, EWSH, RFXDC1, GTF2A1L, C16orf5, ZNF800 and ELF1 were identified as 7 top hits able to form increased spots in response to glucose withdrawal (**Figure R1**). We have put this data into the revised manuscript (Figure 1B).

Figure R1. Z-score calculation of condensed fraction for each TF in HEK293FT cells upon glucose starvation. The cutoff value was set as 3-fold of standard deviation.

2, The authors overexpressed 255 transcription factors in 293FT cells and observed the formation of condensates, are these transcription factors endogenously expressed in 293FT cells or cancer cells to exhibit their transcriptional regulations?

Following the reviewer's suggestion, we re-analyzed our RNA-Seq data in untreated HEK293FT and HGC-27 cells and provided information about the endogenous expression levels of these 255 indicated TFs. The results showed that about 65% or 80% of TFs are endogenously expressed in HEK293FT or HGC-27 cells, respectively. And the expression levels of the 255 TFs are much higher in HGC-27 cells than that of HEK293FT cells (**Figure R28**). Also, the relative expression of the 7 indicated TFs (TEAD4, EWSH, RFXDCA, GTF2A1L, C16orf5, ZNF800 and ELF1) identified to form increased condensates in response to glucose starvation were showed in the heatmap (**Figure R28**).

Figure R28. Heatmap for the relative gene expression levels of indicated 255 TFs with RNA-Seq

data in HEK293FT and HGC-27 cells. The color key represents the relative expression fpkm value.

3, In Figure 1G, the authors found enlargement of the TEAD4 condensates in YAP KO cells, how about TAZ, does it show similar effects as TAZ could also undergo phase-phase separation?

Following the reviewer's suggestion, we performed IF assay to detect TEAD4 condensates in TAZ knockout (TAZ KO) cells. We compared the TEAD4 condensates in both wildtype (WT) and TAZ KO cell line with or without glucose deprivation. The results are similar to that of YAP KO experiments. Specifically, TEAD4 formed similar condensates in both WT and TAZ KO cells at basal state. Moreover, TAZ KO did not affect TEAD4 condensates formation under glucose starvation condition either (**Figure R29**). We have put this data in the revised manuscript (Supplementary Figure 1E).

Figure R29. Fluorescent images of GFP-TEAD4 spots in wild type (WT) or TAZ knockout (TAZKO) cells with or without glucose limitation for 12 h. Scale bar, 10 μ m. The quantification is showed in the right panel. Data are presented as means \pm SD. Significance was tested using unpaired t test, n.s., no significance; ****, $p < 0.0001$.

4, The authors discovered that YAP reduces TEAD4 condensates whereas VGLL4 promotes their formation. Since it is widely known that YAP and VGLL4 compete for TEAD4 binding, the authors should talk about what makes the current study innovative in this regard.

Thanks for pointing this out. As LLPS is well-known for its role in promoting gene transcription activity, we believe that one of the major innovative findings in our current work is the discovery of a widespread phenomenon, *i.e.*, condensation of TFs in response to stressed condition such as glucose limitation. Based on that, we took TEAD4 as an example to showcase that TEAD4-mediated gene transcription can be switched on YAP-triggered active condensation but switched off by VGLL4- or RFXANK- triggered repressive condensation. By understanding the behind mechanism of VGLL4-triggered repressive condensation of LLPS, we were able to manipulate or transit the TEAD4 condensates to a repressive state and thus induce cell death even these cells have sufficient glucose supply. As such, we provide a proof-of-concept for a new avenue to kill cancer cell via creating repressive TF condensation. We now provided more elaboration in our revised manuscript.

5, In Figure 3C, looks like glucose deprivation increased the formation of both TEAD4 monomer and dimer in Native-Page, while the total proteins were not changed in denatured SDS-PAGE, what causes the discrepancy?

Thanks for pointing this out. The reviewer #1 also raised this "smear" issue (Page 9). We re-performed the western blotting assay under native condition to gain insight into the assembly behavior of TEAD4 with or without glucose deprivation. First, we transfected the HEK293FT cells with Flag-TEAD4 for 24 h before subjected to 6 h of glucose deprivation, a time-period window relative longer than the primary manuscript setting, and thus observed much stronger "smear" signals of oligomerization of TEAD4 (**Figure R9A**). Moreover, we further examined the endogenous TEAD4 aggregates in HGC-27 cells under the same condition and reproducibly observed such "smear" signals upon glucose starvation (**Figure R9B**). We have put this part of data into the revised manuscript (Figure 3C).

Figure R9. Immunoblots of endogenous TEAD4 or Flag-tagged TEAD4 in cells with or without glucose deprivation using native gel or SDS-PAGE. (A) Immunoblots of endogenous TEAD4 in HEK293FT cells with or without glucose deprivation using native gel or SDS-PAGE. (B) Immunoblots of Flag-tagged TEAD4 in HGC-27 cells with or without glucose deprivation using native gel or SDS-PAGE.

6, In Supplemental Figure 3D, it doesn't sound like VGLL1 mutant binds more to TEAD4, given the input of VGLL1 mutant is much higher than wild-type? Maybe VGLL1 mutant is more stable? The authors can repeat this experiment.

The reviewer #1 also suggested us to add the 1,6-Hex control (Page 26-27). We re-performed this experiment. Notably, our co-IP assay showed the VGLL1^{mut} with two TDU domains significantly increased TEAD4 homo-association compared to the wildtype VGLL1. Also, the oligomerization of TEAD4 in the presence of VGLL1^{mut} was abrogated by 1,6-Hex treatment (**Figure R27B**, left panel).

Figure R27. Glucose deprivation or VGLL4 promotes TEAD4 oligomerization. (A) TEAD4 oligomerization between HA-TEAD4 and Flag-TEAD4 in cells of glucose limitation with or without

1,6-Hex at indicated time points measured by Co-IP assay. (B) Co-IP analysis of TEAD4 oligomerization in HEK293FT cells transfected with VGLL1, VGLL4 or their mutants with or without 1,6-Hex. VGLL1^{mut}, a construct in which TDU1 (amino acid 206-230) of VGLL4 was added to the N-terminal of wildtype VGLL1 to create a VGLL1 version with two TDUs. VGLL4^{mut}, a construct in which TDU1 (amino acid 206-229) of VGLL4 was deleted to create VGLL4 version with one TDU.

7, In the studies, the authors mostly used HEK293 cell line to perform the experiments. Because the HEK293TF cell line is an embryonic kidney cell, which cannot phenotypically represent cancer cell lines, the authors' use of HEK293TF cells under glucose restriction to simulate the cancer microenvironment does not seem credible. Several important investigations may be replicated by the author utilizing tested cancer cell lines.

Agree! Following this reviewer's suggestion, we first examined the endogenous TEAD4 condensates formation in HGC-27 cells under glucose starvation. In resemble to those observed in HEK293FT cells, we also observed that glucose deprivation clearly induced bigger TEAD4 condensates in HGC-27 cells (**Figure R30**). Meanwhile, as suggested by the reviewer #1, we have also re-performed the TEAD4 ChIP and RNA-Seq assays in HGC-27 cells treated with or without GLUP (**as described above in Figure 20**). Overall, these results demonstrated a transcriptional repressive TEAD4 LLPS in cancer cells. We have put this data into the revised manuscript (Supplementary Figure 1C).

Figure R30. Fluorescent images of endogenous TEAD4 condensates formation in HGC-27 cells under glucose starvation. Scare bar: 1 μ m.

Referee #3 (Report for Author)

In the manuscript "Identification of a repressive type of TF condensation and engineering of a glue peptide therapy against cancer" the authors design a synthetic peptide that convincingly multimerizes TEAD4 transcription factor and show that this peptide can inhibit gastric cancer growth in a mouse model, xenograft model and patient-derived organoids. I have reviewed this identical manuscript for a different journal previously and maintain my opinion that, while this is an interesting finding and the data supporting these findings is solid, the first part of the manuscript describing two classes of TEAD4 condensates is lacking. I have listed set of issues with the condensate part of this manuscript below. While the findings of this manuscript are timely and very interesting, I can only support publication when the issues with the condensate part are sufficiently addressed.

We thank the reviewer for the constructive comments. We have performed additional experiments and analyses to address all the raised issues.

Specific comments:

- Introduction suggests a single role of phase separation during tumorigenesis, while there are probably many different roles of a process as general as phase separation

We added the relevant description on the different roles of phase separation during tumorigenesis in the introduction part.

- "liquidity and dynamics" - please change to material properties.

Thanks! We modified our wording in the revised manuscript as suggested.

- Fig 1 What is being tested by using glucose deprivation? How does it represent both the tumor-micro-environment and cell death-related stress? Is glucose deprivation necessary, since the authors focus on TEAD4 which forms condensates upon over-expression even in the absence of glucose deprivation. The authors need to explain the choices for the conditions of their screen more clearly.

Thanks for pointing this out. It is well established that cancer cells heavily rely on glucose to overly proliferate and the tumor microenvironment is constantly short of glucose supply. In fact, the glucose starvation therapy has been emerging as a method to suppress tumor growth. In this regard, we wanted to figure out what happened to TFs in cells facing glucose deprivation. Thus, we performed the screen in a context of glucose deprivation. We have added the relevant description in our revised manuscript (page 5). Thanks!

- Fig 1 over-expression of TF will be sufficient to induce puncta (evidence in 1D:CDX1), therefore the condensates that are found here need to be confirmed under physiological conditions. Physiological confirmation of TEAD4 (or other) condensates is completely absent from the manuscript, leaving the possibility that all observations pertain over-expression artifacts. This can be achieved by endogenous fluorescent labeling of TEAD4 or immunofluorescence on fixed cells.

As suggested by this reviewer, we now used ChIP-grade TEAD4 antibody to detect endogenous TEAD4 condensates on fixed cells and consistently observed TEAD4 condensation in HGC-27 cells under physiological state. Nevertheless, we also observed that glucose deprivation clearly induced bigger TEAD4 condensates in HGC-27 cells (**Figure R30**). Collectively, these data revealed that TEAD4 condensates represent a general phenomenon in cancer cells, whereas glucose starvation may dramatically

enhance such TEAD4 condensation. We have put this data into the revised manuscript (Supplementary Figure 1C).

Figure R30. Fluorescent images of endogenous TEAD4 condensates formation in HGC-27 cells under glucose starvation. Scale bar: 1µm.

- Fig1B/C - please quantify condensed fraction instead of size and number.

Thanks. The other two reviewers (#1 and #2) also raised this issue (Page 2-3 and 28-29). Following the reviewer’s suggestion, we re-quantified condensed fraction for Figure 1B, C and performed Z-score calculation of condensed fraction to compare fluorescent spot formation between samples. Typically, the activity cut-off value was chosen by three times of the standard deviation of the normalized activity of all samples. Therefore, the results indicated 7 transcriptional factors (TEAD4, EWSH, RFXDC1, GTF2A1L, C16orf5, ZNF800 and ELF1) as candidates able to form increased spots in response to glucose withdrawal (**Figure R1**).

Figure R1. Z-score calculation of condensed fraction for each TF in HEK293FT cells upon glucose starvation. The cutoff value was set as 3-fold of standard deviation.

- Fig1 D Hexanediol disruption is not proof of LLPS, many other parameters need to be evaluated to be able to support LLPS (Alberti, Gladfelter, Mittag Cell 2019). Live cell imaging of cells with endogenously labeled TEAD4 (previous comment) would allow for extensive FRAP, droplet fusion and

dissolution (in response to glucose restoration?). These experiments are needed to support endogenous condensate formation of TEAD4.

We used CRISPER knock-in system to insert GFP at the C-terminal of endogenous *TEAD4* locus. We performed live-cell imaging with fluorescent-labeled TEAD4 to examine endogenous condensate formation of TEAD4. Briefly, GFP-labelled TEAD4-containing cell lines were constructed and labelled with Hoechst for 30 min prior to confocal imaging (**Figure R31**). Indeed, the result indicated endogenous condensate formation of TEAD4.

Figure R31. FRAP analysis of endogenous TEAD4 condensates in HEK293FT cells upon glucose deprivation. Scale bar: 10 μ m.

- Fig1 D Lack of puncta in Hexanediol treated cells says nothing about material properties. What is the conclusion "gel-like transition" based on? Material properties can be evaluated by comparing FRAP kinetics and immobile fraction across conditions. These data are necessary to claim "gel-like transition".

Figure R32. FRAP analysis of TEAD4 condensate in HEK293FT cells upon glucose deprivation. The graph represents the data collected from 25 cells expressing GFP-TEAD4. $t_{1/2}$: fluorescence recovery time; R_f : mobile fraction. Scale bar, 10 μ m.

Following the reviewer's suggestion, we evaluated FRAP kinetics and immobile fractions upon glucose starvation. The half-time ($t_{1/2}$) of TEAD4 condensation upon glucose

starvation was 5.9s and the mobile fraction (Rf) of TEAD4 condensates upon glucose starvation was 0.71 (**Figure R32**). These data suggest that TEAD4 condensation is highly dynamic, with rapid diffusion of molecules within the condensates and that the TEAD4 condensates represent a separate liquid phase that is formed through LLPS. In this regard, we changed our wording "gel-like transition" to LLPS in the revised manuscript.

- Fig 1E Why focus on TEAD4 out of the screen? Can the authors justify? Alternatively, the authors can keep the screen from confounding the message of this manuscript and just focus on TEAD4 because of it's importance in cancer and the previously reported observation that YAP/TAZ in the Hippo pathways form condensates.

Thanks. We initially aimed to identify TFs that can undergo condensation in a context of glucose deprivation mimicking tumor microenvironment. This screening identified 7 candidates (**Figure R1**). Among these 7 TFs, TEAD4 is well studied as a downstream TF of the Hippo signaling pathway and has also been studied as an attractive target for cancer therapy (Holden and Cunningham, 2018). In addition, YAP/TAZ in the Hippo pathways has been previously reported to form condensates together with TEAD4 to promote gene transcription. Thus, it would be intriguing to investigate the stressed-induced TEAD4 condensation, which may help to eventually define the machinery that shut down gene transcription in cells facing glucose starvation. As a matter of fact, our study indeed led to the development of a GLUP anti-tumor strategy based on the finding of a repressive type of TF condensation.

Figure R1. Z-score calculation of condensed fraction for each TF in HEK293FT cells upon glucose starvation. The cutoff value was set as 3-fold of standard deviation.

- Fig 1E YAP/TAZ/TEAD4 form condensates under osmotic stress. Is glucose deprivation a different type of stress that induces Hippo condensates?

As shown in Figure 1E, obvious condensates were observed in cells stably expressing TEAD4 fused with a C-terminal enhanced GFP upon glucose deprivation. We believe glucose deprivation is a type of stress different from osmotic stress. Since glucose withdrawal induces cell death, our current findings indicated that glucose deprivation induces repressive condensates of TEAD4 in a VGLL4-dependent manner, contrasting to the active condensates of TEAD4 induced by YAP/TAZ.

- Fig 1F "glucose deprivation induces fluid-to-gel transition" To support this claim +glucose transition FRAP data displaying faster recovery kinetics is needed.

Agree. Following the reviewer's comment, we determined the half-time ($t_{1/2}$) of TEAD4 condensation upon glucose starvation to be 5.9s and the mobile fraction (R_f) to be 0.71 (**Figure R32**). These data suggest that TEAD4 condensation is highly dynamic, with rapid diffusion of molecules within the condensates and that the TEAD4 condensates represent a separate liquid phase that was formed through LLPS. Thus, we changed our wording "gel-like transition" to LLPS.

- Fig 1F 20 sec recovery is slow and on average there is a 40% immobile fraction. How liquid are these compared to YAP/TAZ condensates (Cai et al Nature Cell Biol 2019)?

We re-quantified our FRAP data and compared the liquidity of TEAD4 condensates with YAP/TAZ condensates previously reported (Cai et al., 2019; Lu et al., 2020). The half-time ($t_{1/2}$) of YAP or TAZ condensation were 1.3s or 3.2s (**Figure R33B-C**), respectively; while the half-time ($t_{1/2}$) of TEAD4 condensation upon glucose deprivation was 5.9s (**Figure R33A**). The mobile fraction (R_f) of TEAD4 condensation under glucose deprivation was 0.71 (**Figure R33A**), which was comparable to that of TAZ condensate ($R_f=0.83$) (**Figure R33C**). These results indicated rapid diffusion of molecules within the TEAD4 condensates which was comparable to that of YAP/TAZ condensates.

Figure R33. FRAP analysis of TEAD4/YAP/TAZ condensate in indicated conditions. (A) FRAP

analysis of TEAD4 condensate in HEK293FT cells upon glucose deprivation. (B) FRAP recovery images of nuclear and cytoplasmic EGFP-YAP condensates. (C) Typical FRAP curves with $\times 40$ objectives in organelles larger than the laser beam.

- Fig S1B what is the time frame of the reduction after glucose supplementation?

Figure S1B shows the fluorescent images of GFP-TEAD4 puncta in glucose starvation-induced HEK293FT cells without glucose supplement for 12 h.

- Fig1G and S1C-D Data shown here shows that there is absolutely no difference in TEAD4 puncta area or number between WT and YAPKO cells. The difference is purely caused by glucose withdrawal. This is evident from both the displayed images and the quantification and is diametrically opposed to the conclusion in the text.

Agree! The reviewer #1 raised the same issue (Page 26). We repeated the YAP KO experiment and performed rescue assay in the YAP KO cells and added quantification for the images. The TEAD4 condensates were the same in YAP KO cells as WT cells. When we overexpressed YAP in YAP KO cells, the TEAD4 condensates were not increased by YAP overexpression (**Figure R26**). So, we concluded that YAP could not affect TEAD4 condensates under glucose deprivation condition. Accordingly, we modified our description in revised manuscript.

Figure R26. Fluorescent images of GFP-TEAD4 condensates in wild type (WT) or YAP knockout (YAPKO) cells rescued by YAP overexpression with or without glucose limitation for 12 h. Scale bar, 10 μm . The quantification is showed in the right panel. Data are presented as means \pm SD. The data were analyzed using one-way ANOVA, followed by the Tukey's post-hoc test. n.s., no significance; ****, $p < 0.0001$.

- Fig S2AB please provide a statistical test.

The reviewer #1 raised similar issue (Page 6). We now re-analyzed the data and added proper statistics to the figures using one-way ANOVA, followed by the Tukey's post-hoc test (**Figure R6**).

Figure R6. Knockdown efficiencies of 7 siRNAs and mRNA levels of *CTGF* and *CYR61* in HGC-27 cells transfected with the indicated siRNAs. (A) Realtime-PCR (RT-PCR) analysis of knockdown efficiencies of indicated 7 siRNAs in HGC-27 cells using *ACTB* as an internal control ($n = 3/\text{group}$). The data were analyzed using one-way ANOVA, followed by the Tukey's post-hoc test. ***, $p < 0.001$, ****, $p < 0.0001$. (B) mRNA levels of *CTGF* and *CYR61* in HGC-27 cells transfected with the indicated siRNAs ($n = 3/\text{group}$). The cutoff value was >2 fold in mRNA change. The data were analyzed using one-way ANOVA, followed by the Tukey's post-hoc test. n.s., no significance; ***, $p < 0.001$, ****, $p < 0.0001$.

• Fig 2D shows large TEAD4 puncta while glucose present, why is this? If glucose withdrawal is not necessary to observe the condensates, why is the screen in Fig 1 performed under glucose withdrawal?

We initially aimed to identify TFs that can undergo condensation in a context of glucose deprivation mimicking tumor microenvironment. That's why we used glucose deprivation condition for the screening. In the manuscript, we intended to show smaller TEAD4 condensates in physiological state (glucose present or e.v.-transfected) (**Figure R34A**, left panel and **Figure R34B**, left panel) and larger TEAD4 condensates under glucose starvation or RFXANK/VGLL4-overexpression (**Figure R34A**, right panel and **Figure R34B**, middle and right panels). And the condensed fractions of TEAD4 in glucose present or e.v.-transfected condition had no difference. Thus, we emphasized that TEAD4 condensates could transit from small size to large size (**Figure R34**). Such transition might reflect the switch from active condensates to suppressive ones.

Figure R34. Comparison of TEAD4 condensates in different conditions. (A) Fluorescent images of GFP-TEAD4 spots in glucose starvation-induced HEK293FT cells. Quantification of TEAD4 condensed fraction were shown. Scale bar, 10 μ m. (B) Fluorescent images and quantification of TEAD4 spot in VGLL4- or RFXANK-overexpressing HEK293FT cells after treatment. Quantification of TEAD4 condensed fraction were shown. Scale bar, 10 μ m.

• Fig 2F please provide quantification

In this regard, we purified mCherry-tagged TEAD4 and labelled VGLL4 proteins with fluorescent dye FITC and re-performed this assay. Also, we added the quantification of TEAD4/VGLL4 condensed fraction (**Figure R35**). We have put this data into the revised manuscript (Figure 2F).

Figure R35. *In vitro* droplet formation assay by VGLL4-TEAD4 mixtures in the presence of different concentrations of NaCl. NaCl (+), 100 mM; NaCl (++) , 250 mM; NaCl (+++), 500 mM. Scale bar, 10 μ m.

• Fig S2F please provide quantification

We suppose the reviewer meant for the quantification of figure S2E, instead of S2F. Here, we added the quantification for TEAD4 condensed fraction (**Figure R36**). We have updated this data in the revised manuscript (Supplementary Figure 2D).

Figure R36. knockdown of VGLL4 or RFXANK HEK293FT cells of glucose starvation results in decreased TEAD4 condensates. HEK293FT cells were treated with individual siRNAs for 60 hr and then treated with glucose deprivation for another 12 hr. After treatment, the condensates of cells were imaged and analyzed. Quantification of TEAD4 condensed fraction is showed in the right panel.

- Fig 2 apoptosis data, purified droplet assays are missing for RFXANK. To support the claims made by the authors these data are also necessary.

Following the reviewer’s comment, we now performed both apoptosis assay and *in vitro* droplet formation assay for RFXANK. The results showed that overexpression of RFXANK induced apoptosis and 1,6-Hex treatment abrogated the regulatory effects of RFXANK on apoptosis (**Figure R37A**). We also purified mCherry-tagged TEAD4 and labelled RFXANK proteins with fluorescent dye FITC and re-performed the droplet formation assay. Like VGLL4, RFXANK protein could also induce TEAD4 condensates *in vitro*, and higher NaCl concentrations inhibited the droplet formation (**Figure R37B**). We have put this data into the revised manuscript (Figure 2G and supplementary figure 2G).

Figure R37. Apoptosis assay and droplet assay of RFXANK. (A) Annexin V staining of RFXANK-overexpressing cells with or without 1,6-Hex. e.v., empty vector. (B) *In vitro* droplet formation assay by RFXANK-TEAD4 mixtures in the presence of different concentrations of NaCl. NaCl (+), 100 mM; NaCl (++) , 250 mM; NaCl (+++) , 500 mM. Scale bar, 10 μ m.

- What do the authors mean with "homo-oligomers" while there are at least two different proteins in there? Please explain.

Indeed, "homo-oligomers" was misleading. What we meant was "VGLL4-induced TEAD4 oligomerization" in our manuscript. We now corrected this statement to oligomers for VGLL4-induced TEAD4 oligomerization.

- Figure 3 why do the authors start using glucose deprivation again? TEAD4 puncta are observed with over-expression alone. Please justify this decision.

We intended to demonstrate the either glucose deprivation or VGLL4 overexpression can induce TEAD4 oligomerization.

- Fig 3C oligomerization combined with slow recovery kinetics point to the direction of gel formation, not LLPS. However LLPS is suggested as the mechanism.

As oligomerization is thought as a general driving force for protein LLPS, we calculated the mobile fraction (R_f) and recovery time ($t_{1/2}$) in VGLL4-overexpressed cells, and found the half-time of TEAD4 condensation was 4.19s. The mobile fraction of TEAD4 condensates in VGLL4-overexpressed cells was 0.77, which was comparable to that of TAZ condensates ($R_f=0.83$). These results indicated rapid diffusion of molecules within the TEAD4 condensates comparable to that of TAZ condensates.

- Fig 3C authors note a smear that is visible in the figure.
- Fig3C authors describe enhanced TEAD4 aggregates of high molecular mass, which is hard to appreciate without quantification. Furthermore, the increase intensity of the TEAD4 monomer and dimer bands indicate that the general level of TEAD4 is increased, not specifically the high molecular mass aggregates. This means the increased protein levels of TEAD4 are likely driving the increased condensate formation upon glucose withdrawal, not TEAD4 oligomerization. Can the authors eliminate the effect of TEAD4 over-expression as a cause for oligomerization?

The other two reviewers also raised the same issue (Page 9 and 32). We now re-performed the western blotting assay under native condition to gain better insight into the oligomeric behavior of TEAD4 with or without glucose deprivation. First, we transfected the HEK293FT cells with Flag-TEAD4 for 24 h before subjected to 6 h of glucose deprivation, a time-period window relative longer than previously described in the manuscript. We observed much stronger "smear" signals of oligomerization of TEAD4 (**Figure R9A**). Moreover, we further examined the endogenous TEAD4 aggregates in HGC-27 cells under the same condition and reproducibly observed such "smear" signals upon glucose starvation (**Figure R9B**).

We have put this part of data into the revised manuscript (Figure 3C).

Figure R9. Immunoblots of endogenous TEAD4 or Flag-tagged TEAD4 in cells with or without glucose deprivation using native gel or SDS-PAGE. (A) Immunoblots of endogenous TEAD4 in HEK293FT cells with or without glucose deprivation using native gel or SDS-PAGE. (B) Immunoblots of Flag-tagged TEAD4 in HGC-27 cells with or without glucose deprivation using native gel or SDS-PAGE.

- Fig 3H clearly shows dimers, but not oligomers? Is the observed effect just explained by two binding sites on VGLLx?

This was because the upper half of the SDS-PAGE gel image was cropped. We now show the uncropped image of this assay. As shown below, we clearly detected dimers and oligomers of TEAD4 in the presence of VGLL4 (**Figure R38**). We have updated this part of data into the revised manuscript (Figure 3H).

Figure R38. Crosslinking analysis of TEAD4 in the presence of VGLL1, VGLL4, or their mutants with Coomassie blue staining. DSS, a cross-linker reagent.

- Figure 4. Short linear DNA may be incorporated as a client, but long chromosomal DNA cannot permeate condensates. The in vitro experiment with short DNA segments is therefore not representative of the in vivo situation. Longer DNA can better mimic the endogenous conditions, for example lamda-DNA.

We are sorry for not clearly describing this part. Actually, the DNA sequence length of the pUC-GW-Kan vector containing the inserted M-CAT triple tandem repeat sequence used in our work was 3 to 4 Kbp.

- Fig 4A this assay cannot distinguish between binding and client retention

Indeed, this assay cannot distinguish between binding and retention. We intended to say that TEAD4 but not VGLL4 was able to retard the migration of DNA; TEAD4 in combination with VGLL4 significantly increased such effect. Meanwhile, 1,6-Hex largely abrogated this promoting effect of VGLL4, suggesting a process related to LLPS (Figure 4A).

- Fig 4C why does TEAD4:DNA form droplets? Multiple TEAD4 binding sites on the DNA? If it can be nucleated, maybe TEAD4 has a high threshold concentration? Can the authors perform titration of TEAD4 in a droplet formation assay?

We are sorry for not clearly describing the information of DNA used in this experiment. The DNA here contained three M-CAT motifs in tandem which can bind more than one TEAD4 molecule. The M-CAT triple tandem repeat sequence of DNA provided multiple TEAD4 binding sites, making it possible for TEAD4:DNA to form droplets. We also re-performed the droplet formation assay using fluorescent proteins (**Figure R39**). We now updated this data in the revised manuscript (Figure 4C).

Figure R39. Images showing the droplet formation of DNA with or without indicated proteins. Scale bar, 10 μm .

- Fig 4D its unclear what is imaged here. Are the used DNA molecules 39bp long?

As mentioned above, the DNA sequence length of the pUC-GW-Kan vector containing the inserted M-CAT triple tandem repeat sequence used in our work was 3 to 4 Kbp. Figure 4D showed the compaction level of the DNA in a context of VGLL4-induced TEAD4 oligomerization. For better clarity, we quantified the DNA compaction level by measuring the half-length of DNA. The staining showed that TEAD4 itself could only slightly lead to DNA aggregation, whereas, VGLL4-TEAD4 complex caused severe aggregation of the DNA (**Figure R40**). We now updated this data in the revised manuscript (Figure 4D).

Figure R40. Representative images of DNA organization and appearance in the presence or absence of TEAD4 and VGLL4. DNA was stained with DAPI. Half-length of DNA was determined. Scale bar, 100 nm.

- Fig 4F There appears to be no difference between the conditions.
- Fig 4G unclear what is being shown here.

These two figures were performed to address the compaction of DNA upon VGLL4-overexpression. In figure 4F, DAPI staining of VGLL4-overexpressing cells showed that chromatin displayed aggregated and compacted structure. As shown in Figure 4G, VGLL4-induced extensive TEAD4 condensation were found to be gathered nearby chromatin. Red arrow denotes TEAD4 particles with colloidal gold-conjunct TEAD4 antibody. The black dots nearby the TEAD4 particles represented the DNA. The more number of black dots around the TEAD4 particles, the more aggregation and compaction of the DNA.

- Fig 4H Contrary to the data in Fig 2D TEAD4 now can form condensates while treated with Hexanediol. Can the authors explain?

Thanks. Such discrepancy may be caused by the transcriptional active versus repressive condensates. In VGLL4-transfected cells treated with 1,6-hexanediol, TEAD4 may form small and transcriptional active condensates in which TEAD4 and HEK27me3 were not colocalized (**Figure R41A**, panel 3 and **Figure R42B**, panel 3); while the VGLL4-induced bigger condensates of TEAD4 were repressive for gene transcription (**Figure R41A**, panel 2 and **Figure R42B**, panel 2).

Figure R41. Fluorescent images of TEAD4 condensates in VGLL4- or RFXANK-overexpressing HEK293FT cells after treatment with or without 1,6-Hex. (A) Immunofluorescence staining (upper and lower, representative images with zoom-in) of GFP-TEAD4 and H3K27me3 in VGLL4-overexpressing cells treated with or without 1,6-Hex. Scale bar, 10 μ m. (B) Fluorescent images of TEAD4 condensates in VGLL4- or RFXANK-overexpressing HEK293FT cells after treatment with or without 1,6-Hex. Scale bar, 10 μ m.

- Fig 4H,L and S4A-C are all individual examples, the authors need to quantify this colocalization over multiple cells and TEAD4 condensates.

Thanks for pointing this out. We now provided high-quality images and quantified the colocalization over multiple cells, as well as the TEAD4 condensed fraction.

Figure R42. Colocalization over multiple cells and the TEAD4 condensed fraction of these fluorescent images. (A) Immunofluorescence staining (upper and middle, representative images with zoom-in; lower, quantification of fluorescent intensity with indicated color scheme) of GFP-TEAD4 and H3K27me3 in VGLL4-overexpressing cells treated with or without 1,6-Hex (n=6). Scale bar, 1 μm. (B) Immunofluorescence staining (upper and middle, representative images with zoom-in; lower, quantification of fluorescent intensity with indicated color scheme) of GFP-TEAD4 and H3K27ac in VGLL4-overexpressing cells treated with or without 1,6-Hex (n=6). Scale bar, 1 μm. (C) Immunofluorescence staining (upper and middle, representative images with zoom-in; lower, quantification of fluorescent intensity with indicated color scheme) of GFP-TEAD4 and H3K27me3 in RFXANK-overexpressing cells treated with or without 1,6-Hex (n=6). Scale bar, 1 μm. (D) Immunofluorescence staining (upper and middle, representative images with zoom-in; lower, quantification of fluorescent intensity with indicated color scheme) of GFP-TEAD4 and H3K27ac in RFXANK-overexpressing cells treated with or without 1,6-Hex (n=6). Scale bar, 1 μm. (E) Immunofluorescence staining (upper and middle, representative images with zoom-in; lower, quantification of fluorescent intensity with indicated color scheme) of GFP-TEAD4 and H3K27me3 in VGLL4-expressed HEK293FT cells with or without YAP (5SA) co-transfection (n=6). Scale bar, 1 μm. (F) Quantification of TEAD4 condensed fraction of above images. The data were analyzed using one-way ANOVA, followed by the Tukey's post-hoc test. ****, p < 0.0001.

• Fig 5 there must be additional interactions between GLUP-mediated TEAD4 dimers in order to have oligomers. How can the authors explain the oligomerization? What parts of TEAD4 are responsible for this?

As shown in figure 5A, we used TEAD4 YBD (YAP-binding domain) domain, not the full-length TEAD4, to dock with the GLUP peptide. In fact, previous structural studies have revealed multiple mode of homo-association of the YBD domain in the crystal packing, suggesting a tendency to form multiple types of oligomers under certain conditions. Here,

VGLL4 or GLUP may act as a trigger to create such kind of conditions---initial dimerization of TEAD4 triggered by VGLL4 or GLUP may cause further oligomerization through "mismatch", *i.e.*, dimerization via different interfaces. In addition to the YBD domain, TEAD4 contains a TEA domain responsible for DNA binding. It is also possible that the TEA domain may contribute to GLUP-induced high-order oligomerization.

- Fig 6 these results can be explained by wholesale TEAD4 aggregation in the nucleus and complete genomic rearranging, leading to a toxic effect. Unrelated to "repressive TEAD4 condensates".

Except for the results in figure 6, we also performed a series of experiments in supplementary figure 6 to address the "repressive TEAD4 condensates". Firstly, we conducted RNA-sequencing to evaluate the transcriptomics of HGC-27 cells treated with GLUP. GLUP treatment induced an obvious downregulation in TEADs signature genes (for example, *CTGF*, *CYR61*, *AXL*, *CCNA2* and *CGB5*) (**Figure R43A**). Secondly, we performed ChIP-Seq assay by using an antibody specifically recognizing H3K27me3. Typical H3K27me3 peaks were found to be upregulated, while TEAD4-associated peaks were found to be downregulated in GLUP-treated cells compared to control cells (**Figure R43B**). Thirdly, we performed colocalization analysis by IF in GFP-TEAD4-expressing cells, and found that only H3K27me3 but not H3K27ac, colocalized in GLUP-treated TEAD4 condensates (**Figure R43C**), suggesting that TEAD4 condensates contribute to transcriptional repression.

Figure R43. GLUP-induced repressive TEAD4 condensates. (A) Heatmap for down-regulated genes in GLUP-treated HGC-27 cells. (B) Heatmap representing the distribution of H3K27me3-binding relative to the gene transcription start site (TSS) in HGC-27 treated with or without GLUP. (C) Representative fluorescence images of mCherry-TEAD4 and H3K27ac in GLUP-treated HEK293FT cells treated with or without 1,6-Hex. Quantification of fluorescent intensity with indicated color scheme is showed in the lower panel. Scale bar, 10 μm.

- Fig 6C supports gross chromatin deformation as a mechanism. Firstly, we detected the expression of TEAD4 downstream genes decreased in HGC-27 cells treated with GLUP. Then, to test the specificity of GLUP, we chose AGS, a human GC cell line with a low level of TEAD4 expression and detected the viability of AGS cells was only slightly or modestly inhibited by GLUP treatment. However, AGS cells overexpressing TEAD4 but not an empty vector clearly became sensitive to GLUP. In addition, we performed ChIP-Seq and divided the H3K27ac peaks into TEAD4-specific and non-specific groups. We found that GLUP treatment dramatically reduced the H3K27ac-binding onto the TEAD4-specific motifs. Also, 1,6-Hex obviously abrogated GLUP-induced *CTGF* downregulation, showing that GLUP could induce DNA aggregation to repress gene transcription. Therefore, these data indeed indicate chromatin deformation as a mechanism for GLUP-induced repressive TEAD4 condensation; and

such chromatin deformation appears to specially repress TEAD4 target gene transcription. We now revised our description and discussion in the manuscript.

Tang et al. screened the condensation formation of over 700 transcription factors in cells with glucose deprivation and found that TEAD4, among other transcription factors, can form condensates. They further identified association factors of TEAD4 by Bio-ID and found that VGLL4 is an inducer of condensate formation of TEAD4. They further found a peptide that can induce TEAD4 oligomerization to form condensates. The peptide has a strong antitumor effect via inhibition of TEAD4 related gene transcription. These results are interesting and exciting. However, whether these condensates are formed in physiological conditions are questionable. Additionally, the insufficient description of methods and figures prevents the reviewer from assessing the quality of data. I would recommend the manuscript for publication if concerns were addressed.

We thank the reviewer for the constructive comments. In our revised manuscript, we now provided quantifications and added detailed descriptions.

Essential revisions

The authors should provide evidence that endogenous VGLL4 and TEAD4 form condensates and colocalize with each other. Crucially, depletion of VGLL4 in cells should abolish the condensation formation of TEAD4.

Following the reviewer's suggestion, we performed immunofluorescent assay to detect endogenous TEAD4 condensates formation and observed endogenous TEAD4 condensates in HEK293FT cells under physiological state, which was increased upon glucose starvation (**Figure R44A**). Indeed, knockdown of VGLL4 significantly inhibited TEAD4 condensates formation (**Figure R44B**).

Figure R44. Endogenous TEAD4 condensates in different conditions. (A) Fluorescent images of endogenous TEAD4 condensates in glucose starvation-induced HEK293FT cells. Scale bar, 10 μ m. (B) Knockdown of VGLL4 in HEK293FT cells of glucose starvation results in decreased TEAD4 condensates. Quantification of TEAD4 condensed fraction is showed in the right panel. Scale bar, 10 μ m.

Major points

1. 1,6-Hex is not a reagent that can test whether proteins undergo LLPS.

In addition to 1,6-Hex, we also used FRAP assay to test whether TEAD4 proteins undergo LLPS. As the half-time ($t_{1/2}$) of TEAD4 condensate with GLUP treatment was 1.28 s and the mobile fraction (R_f) was 0.86 (**Figure R45**). These data suggest that TEAD4 condensation under GLUP treatment was highly dynamic, with rapid diffusion of molecules within the condensates and that the TEAD4 condensates represent a separate liquid phase that is formed through LLPS.

Figure R45. FRAP analysis of TEAD4 condensate with purified TEAD4 protein with FAM-GLUP treatment. For each photobleached spot, the fluorescence recovery curve was traced. The graph represents the data collected from 5 droplets. $t_{1/2}$: fluorescence recovery time; R_f : mobile fraction. Scale bar, 2 μm .

2. The authors state that "Notably, no obvious puncta were observed in 1,6 Hex treated cells, suggesting that the TF condensates may undergo a gel like transition in glucose deprived cells." This statement is incorrect.

Thanks for pointing this out. We have modified this statement as "TF condensates may undergo phase separation in glucose deprived cells." in the revised manuscript.

3. Previous studies have shown that YAP can condensates that TEAD4 condensates; however, the author's studies did not support this. What are the discrepancies among these experiments?

Good point! We now discussed these discrepancies in our revised manuscript.

"Cofactors differentially mediated repressive or active condensation of TFs- Gene transcription is a tightly regulated process as it controls the fate of a cell, organ, and even whole body. The finely tuned regulation of gene expression maintains homeostasis, whereas dysregulation of transcription leads to serious consequences such as tumorigenesis. The regulation of gene transcription involves not only individual TFs but also complexes composed of TFs and cofactors (coactivators and corepressors) which play a pivotal role in modulating TF activity. Transcription cofactors can participate in splicing, gene looping, and phase separation to ultimately fine-tune the process of transcription. In the last decade, multiple TFs have been found to allow for multivalent weak interactions with other TFs and/or cofactors, subsequently leading to the formation of phase-separated condensates to promote transcription activity. In contrast to the well-known condensation-induced transcription activation, the kind of repressive condensation of TF is poorly studied. In this work, we dissected the molecular mechanism of YAP or VGLL4-induced phase separation and proposed a machinery in which active or repressive condensation of TEAD4 play opposite roles for gene transcription (Fig. 4M). Thus cofactor-mediated differential condensation of TFs may act as a switch to control

transcriptional activity.”

4. TEAD4 can be partitioned into YAP-condensates or VGLL4 condensates. What is the potential mechanism for this selectivity?

According to the previous paper (Jiao et al., 2014), we think the binding affinity of TEAD4 with YAP or VGLL4 and the concentration of YAP or VGLL4 in the nucleus might be the potential reasons for the selectivity.

5. The authors state that "Transitions from liquid to gel like states in proteins are often facilitated by weak multivalent interactions between proteins". There is no such data to support this transition.

We corrected this statement in our revised manuscript.

6. The authors used N-STORM to image DNA in cells overexpressing VGLL4 and then claimed there are differences in these cells. Based on my examination, these images are similar. The authors should develop a metric to quantify these differences if they believe these images are different.

We described this image according to previously reported paper (Jiao et al., 2023). In figure 4F, DAPI staining of VGLL4-overexpressing cells showed that chromatin displayed aggregated and compacted structure.

7. The quality of images of co-immunostaining of H3K27me3 and H3K27Ac with GFP-TEAD4 is low. High-quality images should be helpful.

Agree! We now replaced them with high-quality images with quantification.

Figure R46. Fluorescent images and quantification of these fluorescent images. (A) Immunofluorescence staining (upper and middle, representative images with zoom-in; lower, quantification of fluorescent intensity with indicated color scheme) of GFP-TEAD4 and H3K27me3 in VGLL4-overexpressing cells treated with or without 1,6-Hex (n=6). Scale bar, 1 μm. (B) Immunofluorescence staining (upper and middle, representative images with zoom-in; lower, quantification of fluorescent intensity with indicated color scheme) of GFP-TEAD4 and H3K27ac in VGLL4-overexpressing cells treated with or without 1,6-Hex (n=6). Scale bar, 1 μm. (C) Immunofluorescence staining (upper and middle, representative images with zoom-in; lower, quantification of fluorescent intensity with indicated color scheme) of GFP-TEAD4 and H3K27me3 in RFXANK-overexpressing cells treated with or without 1,6-Hex (n=6). Scale bar, 1 μm. (D) Immunofluorescence staining (upper and middle, representative images with zoom-in; lower, quantification of fluorescent intensity with indicated color scheme) of GFP-TEAD4 and H3K27ac in RFXANK-overexpressing cells treated with or without 1,6-Hex (n=6). Scale bar, 1 μm.

8. The length of DNA sequences is less than 40 bp, which is under the resolution of optic microscopy (confocal). Could the authors explain how they measure the morphology of DNA in vitro?

We are sorry for not clearly describing this part. Actually, the DNA sequence length of the pUC-GW-Kan vector containing the inserted M-CAT triple tandem repeat sequence

used in our work was 3 to 4 Kbp.

9. I would recommend that the authors use condensed fraction to describe the condensation capacity of proteins, which allows a systematic comparison among transcription factors and between different conditions.

Good point! We now used condensed fraction to quantify the condensation capacity of proteins. Also, we calculated the fluorescence recovery time (τ) and mobile fraction (R_f) in FRAP assays. The quantification of condensed fraction was added to the revised manuscript.

10. Figure 1F, why are the FRAP results normalized to 240 rather than 1?

Following the reviewer's suggestion, we now normalized FRAP values to 1 (**Figure R32**).

Figure R32. FRAP analysis of TEAD4 condensate in HEK293FT cells upon glucose deprivation. White circles denote the photobleached regions in the spot and nucleoplasm (upper). Three images were gradually taken during pre-bleach, bleaching pulses, followed by fluorescence recovery as indicated on the graph (upper). For each photobleached spot, the fluorescence recovery curve was traced (lower). The graph represents the data collected from 25 cells expressing GFP-TEAD4. $t_{1/2}$: fluorescence recovery time; R_f : mobile fraction. Scale bar, 10 μm .

11. Figure 1H, I would suggest that the authors label YAP and TEAD4 with fluorescence protein or fluorescence dye, which facilitates the qualification. The authors should quantify the phase separation ability.

We now re-performed this assay with fluorescent dye and quantified the phase separation ability using condensed fraction (**Figure R47**). In the *in vitro* LLPS assay, we did observe the formation of YAP droplets. However, the solution containing purified TEAD4 protein of two tested concentrations remained clear during the imaging process. We have put this data into the revised manuscript (Figure 1G).

Figure R47. Droplet formation assay of GFP-tagged YAP or TEAD4 proteins by differential interference microscopy (DIC). Quantification of condensed fraction is showed in the right panel. Scale bar, 10 μm .

12. Figure 2F, the authors should label VGLL4 and TEAD4 with respective fluorescence dyes. I would suggest that the authors vary the concentrations of TEAD4 and test how TEAD4 impacts the condensation capacity of VGLL4.

We now re-performed this assay using fluorescence labelled-VGLL4 or TEAD4 proteins (**Figure R35**). We reproducibly observed that droplets rapidly formed upon mixing VGLL4 and TEAD4 proteins together, whereas higher concentrations of NaCl decreased the condensed fractions of TEAD4/VGLL4. We have put this data in the revised manuscript (Figure 2F).

Figure R35. *In vitro* droplet formation assay by VGLL4-TEAD4 mixtures in the presence of different concentrations of NaCl. NaCl (+), 100 mM; NaCl (++) , 250 mM; NaCl (+++), 500 mM. Quantification of TEAD4 or VGLL4 condensed fraction (right) were shown. The data were analyzed

using one-way ANOVA, followed by the Tukey's post-hoc test. *, $p < 0.05$; ***, $p < 0.001$; ****, $p < 0.0001$. Scale bar, 10 μm .

13. Figure 2E, again, why are the FRAP results normalized to 240 rather than 1?

Following the reviewer's suggestion, we now normalized FRAP values to 1 (**Figure R48**) and calculated the fluorescence recovery time and mobile fraction.

Figure R48. FRAP analysis of TEAD4 condensate in HEK293FT cells transfected with VGLL4. For each photobleached spot, the fluorescence recovery curve was traced (lower). The graph represents the data collected from 16 cells expressing GFP-TEAD4. $t_{1/2}$: fluorescence recovery time. Scale bar, 10 μm .

14. Figure 3B, the authors should quantify the colocalization.

We now provided quantification for the colocalization assay (**Figure R49**). As shown below, strong signals for co-localization of mCherry-tagged TEAD4 and GFP-tagged TEAD4 were observed in glucose deprivation-induced TEAD4 condensates.

Figure R49. Co-localization of GFP-TEAD4 and mCherry-TEAD4 in HEK293FT cells upon glucose deprivation. Scale bar, 10 μm .

15. Figure 3, the current in vitro cross-linking data suggests that TEAD4 forms dimer rather than oligomer. Other alternative mechanisms may exist for the formation of TEAD4 condensates by VGLL4. For instance, an alternative mechanism is scaffold-client one.

Thanks for pointing this out. The reviewer #3 raised the same issue. This was because the upper half of the SDS-PAGE gel image was cropped. We now show the uncropped image of this assay. As shown below, we clearly detected dimers and oligomers of TEAD4 in the presence of VGLL4 (**Figure R38**).

Figure R38. Crosslinking analysis of TEAD4 in the presence of VGLL1, VGLL4, or their mutants with Coomassie blue staining. DSS, a cross-linker reagent.

16. Figure 3J and 4C, again, fluorescent images should be given.

We now re-performed these assays and provided fluorescent images (**Figure R50**) in the revised manuscript. The result showed that only wildtype VGLL4 protein, but not VGLL4^{mut} protein could trigger TEAD4 droplet formation in the *in vitro* droplet formation assay (**Figure R50A**), confirming the necessity of two TDUs for VGLL4 in promoting TEAD4 condensation. Also, in **Figure R50B**, we observed that VGLL4 markedly enhanced the droplet formation of the TEAD4-DNA complex.

Figure R50. Fluorescent image of droplet formation assay. (A) Images and quantification showing TEAD4 droplet formation in the presence of VGLL4 or VGLL4mut. Scale bar, 10 μ m. (B) Images

showing the droplet formation of DNA with or without indicated proteins. Scale bar, 10 μm .

17. Figure 4D, a quantitative description of compaction should be shown.

The reviewer #3 also suggested a quantification. The DNA sequence length of the pUC-GW-Kan vector containing the inserted M-CAT triple tandem repeat sequence used in our work was 3 to 4 Kbp. Figure 4D showed the compaction level of the DNA in a context of VGLL4-induced TEAD4 oligomerization. Following this reviewer's suggestion, we now quantified the DNA compaction level by measuring the half-length of DNA. The staining showed that TEAD4 itself could only slightly lead to DNA aggregation, whereas, VGLL4-TEAD4 complex caused severe aggregation of the DNA (**Figure R40**). We now updated this data in the revised manuscript (Figure 4D).

Figure R40. Representative images of DNA organization and appearance in the presence or absence of TEAD4 and VGLL4. DNA was stained with DAPI. Scale bar, 100 nm.

Three-color images should be shown with proteins labelled with different fluorescent proteins or dyes. Following the reviewer's comments, we now re-performed all the IF assays with different fluorescent dyes/proteins as listed below (**Figure R51**).

Figure R51. Fluorescent image of droplet formation assay. (A) Droplet formation assay of GFP-tagged YAP or TEAD4 proteins by differential interference microscopy (DIC). Quantification of condensed fraction is showed in the right panel. Scale bar, 10 μ m. (B) *In vitro* droplet formation assay by VGLL4-TEAD4 mixtures in the presence of different concentrations of NaCl. NaCl (+), 100 mM; NaCl (++) , 250 mM; NaCl (+++), 500 mM. Scale bar, 10 μ m. (C) *In vitro* droplet formation assay by RFXANK-TEAD4 mixtures in the presence of different concentrations of NaCl. NaCl (+), 100 mM; NaCl (++) , 250 mM; NaCl (+++), 500 mM. Scale bar, 10 μ m. (D) Droplet

formation of mCherry-TEAD4 proteins in the presence of VGLL4/VGLL1. Scale bar, 10 μ m. (E) Images and quantification showing TEAD4 droplet formation in the presence of VGLL4 or VGLL4^{mut}. Scale bar, 10 μ m. (F) Images showing the droplet formation of DNA with or without indicated proteins. Scale bar, 10 μ m. (G) GLUP-induced phase separation of TEAD4. Solubility of TEAD4 proteins after treatment with GLUP (left). Droplet formation of TEAD4-GLUP mixture in different concentrations of NaCl (right). NaCl (+), 20 mM; NaCl (++) , 150 mM; NaCl (+++), 500 mM. Scale bar, 10 μ m.

Minor point

1. Sanger Cloud Platform (www.i.sanger.com) is not accessible.

We now corrected this information on Sanger Cloud Platform (www.majorbio.com).

References

- Brown, K., Chew, P.Y., Ingersoll, S., Espinosa, J.R., Aguirre, A., Espinoza, A., Wen, J.Y., Astatike, K., Kutateladze, T.G., Collepardo-Guevara, R., *et al.* (2023). Principles of assembly and regulation of condensates of Polycomb repressive complex 1 through phase separation. *Cell Reports* *42*.
- Cai, D.F., Feliciano, D., Dong, P., Flores, E., Gruebele, M., Porat-Shliom, N., Sukenik, S., Liu, Z., and Lippincott-Schwartz, J. (2019). Phase separation of YAP reorganizes genome topology for long-term YAP target gene expression. *Nature Cell Biology* *21*, 1578-+.
- Chen, T. (2010). *A practical guide to assay development and high-throughput screening in drug discovery* (Boca Raton: CRC Press).
- Holden, J.K., and Cunningham, C.N. (2018). Targeting the Hippo Pathway and Cancer through the TEAD Family of Transcription Factors. *Cancers (Basel)* *10*.
- Jiao, S., Li, C.C., Guo, F.H., Zhang, J.J., Zhang, H., Cao, Z.F., Wang, W.J., Bu, W.B., Lin, M.B., Lue, J.H., *et al.* (2023). SUN1/2 controls macrophage polarization via modulating nuclear size and stiffness. *Nature Communications* *14*.
- Jiao, S., Wang, H., Shi, Z., Dong, A., Zhang, W., Song, X., He, F., Wang, Y., Zhang, Z., Wang, W., *et al.* (2014). A peptide mimicking VGLL4 function acts as a YAP antagonist therapy against gastric cancer. *Cancer Cell* *25*, 166-180.
- Leihener, A., Slefarska, D., Leja, M., Heinzle, C., Mündlein, A., Kikuste, I., Mezmale, L., Drexel, H., Mayhew, C.A., and Mochalski, P. (2021). The Volatilomic Footprints of Human HGC-27 and CLS-145 Gastric Cancer Cell Lines. *Front Mol Biosci* *7*.
- Lu, Y., Wu, T.T., Gutman, O., Lu, H.S., Zhou, Q., Henis, Y.I., and Luo, K.X. (2020). Phase separation of TAZ compartmentalizes the transcription machinery to promote gene expression. *Nature Cell Biology* *22*.
- Shi, Z., He, F., Chen, M., Hua, L., Wang, W., Jiao, S., and Zhou, Z. (2017). DNA-binding mechanism of the Hippo pathway transcription factor TEAD4. *Oncogene* *36*, 4362-4369.
- Tang, Y., Fang, G., Guo, F., Zhang, H., Chen, X., An, L., Chen, M., Zhou, L., Wang, W., Ye, T., *et al.* (2020). Selective Inhibition of STRN3-Containing PP2A Phosphatase Restores Hippo Tumor-Suppressor Activity in Gastric Cancer. *Cancer Cell* *38*, 115-128 e119.
- Wang, J.X., Zhou, J.F., Huang, F.K., Zhang, L., He, Q.L., Qian, H.Y., and Lai, H.L. (2017). GLI2 induces PDGFRB expression and modulates cancer stem cell properties of gastric cancer. *Eur Rev Med Pharmacol Sci* *21*, 3857-3865.
- Wu, X.M., Shao, X.Q., Meng, X.X., Zhang, X.N., Zhu, L., Liu, S.X., Lin, J., and Xiao, H.S. (2011). Genome-wide analysis of microRNA and mRNA expression signatures in hydroxycamptothecin-resistant gastric cancer cells.

Acta Pharmacol Sin 32, 259-269.

Dear Zhaocai,

Thank you for submitting your revised manuscript to The EMBO Journal. Your manuscript has now been seen by all original reviewers, and you can find their comments below.

As you can see, reviewer #1 indicates remaining issues with the data conclusiveness and interpretation. In particular, he/she finds that the association of phase-separated TEAD4 and its target genes with repressive chromatin marks is not conclusively shown. Furthermore, he/she finds that the provided data do not support the role of GLUP in inducing TEAD4-repndent repressive chromatin phase separation, but rather suggest GLP-dependent removal of TEAD4 from its target genes. In the reviewer cross-consultation session, reviewers #3 and #4 agreed that these are important concerns that would need to be convincingly clarified in a revised version, e.g., by revising the model of TEAD4 phase co-separation with repressive chromatin due to the lack of repressive chromatin marks at TEAD4 target genes in the ChIP data. The model of GLUP action should also be revisited to incorporate the observed removal of TEAD4 from chromatin. Please also add the requested analysis in TEAD4 KO cells to exclude a TEAD4-independent mode of GLUP action.

Please keep in mind that the outcome of the final acceptance of the manuscript will depend on the conclusiveness of your response to these remaining concerns. I would also be happy to discuss this second revision via email or Zoom. Thank you again for the opportunity to consider your work for publication, and I look forward to receiving your revision.

With kind regards,

leva

leva Gailite, PhD
Senior Scientific Editor
The EMBO Journal
Meyerhofstrasse 1
D-69117 Heidelberg
Tel: +4962218891309
i.gailite@embojournal.org

Please remember: Digital image enhancement is acceptable practice, as long as it accurately represents the original data and conforms to community standards. If a figure has been subjected to significant electronic manipulation, this must be noted in the

figure legend or in the 'Materials and Methods' section. The editors reserve the right to request original versions of figures and the original images that were used to assemble the figure.

We realize that it is difficult to revise to a specific deadline. In the interest of protecting the conceptual advance provided by the work, we recommend a revision within 3 months (4th Aug 2024). Please discuss the revision progress ahead of this time with the editor if you require more time to complete the revisions.

Referee #1:

The authors have now addressed several of my major points which I appreciate. However, some things were not addressed or simply ignored (e.g. TEAD4 KO cell line). Furthermore, GLUP seems to strip TEAD4 off chromatin (R19B) which undermines the whole mechanism the authors propose in the manuscript and in the title. I am really sorry, but really cannot follow their conclusions here when it comes to GLUP-mediated repression via H3K27me3.

Major

It is very unusual that the majority of requests were merely addressed by figures for the reviewer. Of course, there are certain restrictions in terms of space etc. However, there is a reason for bringing up these points, namely, making it clearer for the scientific community so that they can better judge the findings of a publication.

1. Regarding points 10/15: I asked the authors to analyze TEAD4 and H3K27me3 (not H3K27ac). Even though K27ac and K27me3 targets the same lysine residue, their genomic localization is VERY different and cannot be used interchangeably => H3K27me3 marks whole gene bodies whereas H3K27ac marks promoters and active enhancers. The analysis R19G,H is very confusing for me: what does "TEAD4-specific motifs" mean. Is this a peak set containing TEAD motifs or are these real experimental TEAD4 peaks?

I have two issues with R19G,H: 1.) the analysis provided is not quantitative and statistics are missing. Since the analysis is not described, I cannot judge what I am looking at but the heatmap certainly contains only very few peaks and why is the number of peaks different between G and H. If the authors have defined TEAD4 peaks, the heatmaps should not differ in terms of peaks (which they clearly do) 2.) having looked at the bigwig files for the H3K27me3 ChIP-Seq: there is no induction of H3K27me3 at CTGF, CYR61 & Co which is not in line with their proposed mechanism.

Specific response to point 15 (Figure R19):

Unfortunately, the authors did not perform the experiment in a TEAD4 KO cell line, and also do not comment on this in the text why they did not perform this essential experiment as requested.

Furthermore, it is crystal clear that GLUP leads to a complete loss of TEAD4 binding at CTGF in their ChIP-Seq analysis (R19B) and (having looked at the bw files uploaded to GEO) also in the new CUT&TAG analysis. This is at odds to the proposed mechanism of the whole paper.

Thus, I have to question the conclusions about GLUP-mediated repression via TEAD4-dependent repressive LLPS, especially since this claim cannot be backed up by genome-wide H3K27me3 data. It seems more likely that GLUP leads to a loss of TEAD binding which can explain the effects on TEAD targets and secondary epigenomic changes.

The CUT&TAG data tracks (R19E) are unfortunately not labelled, but I assume that this is TEAD4?

Referee #2:

The authors took great effort in the revision, and have completed a lot of new experiments to address previous comments. All of my questions are addressed, and I am happy to support its publication.

Referee #3:

The authors have added an impressive amount of work to address the comments of the four reviewers. My comments focused

on the first half of the manuscript and in general they are sufficiently addressed by the new data. I now recommend this manuscript for publication in EMBO Journal.

Please note that the manuscript can benefit from textual editing, for example "As TEAD4 has been well-characterized in forming LLPS for" p7, should be "As TEAD4 has been well-characterized to form LLPS-driven condensates that...", etc.

Referee #4:

The authors have addressed my concerns. I recommend the publication of this manuscript.

A Point-by-point Response to Reviewers' Comments

Referee #1:

The authors have now addressed several of my major points which I appreciate. However, some things were not addressed or simply ignored (e.g. TEAD4 KO cell line). Furthermore, GLUP seems to strip TEAD4 off chromatin (R19B) which undermines the whole mechanism the authors propose in the manuscript and in the title. I am really sorry, but really cannot follow their conclusions here when it comes to GLUP-mediated repression via H3K27me3.

We thank this reviewer for the critical comments regarding GLUP stripping TEAD4 off chromatin. To address this issue, we have now performed additional experiments including Cut&Tag against H3K27me3 in both WT and TEAD4-knockout cell lines. We indeed observed enhanced H3K27me3 signals upon treatment with GLUP, which coincided with reduced gene transcription (**Fig. R1**). Meanwhile, we thoroughly re-analyzed the TEAD4 sequencing data and found its binding to chromatin decreased on a subset of genomic regions upon GLUP treatment. Therefore, GLUP may suppress gene transcription either through TEAD4-mediated DNA entangling or simply by stripping TEAD4 off chromatin. Accordingly, we have modified our description in the revised manuscript including in the "Limitations of this study" section.

Major

It is very unusual that the majority of requests were merely addressed by figures for the reviewer. Of course, there are certain restrictions in terms of space etc. However, there is a reason for bringing up these points, namely, making it clearer for the scientific community so that they can better judge the findings of a publication.

Agreed. We have now re-organized the main text to better elucidate the dual models of suppression of gene transcription, namely GLUP-induced suppression and TEAD4-dependent suppression. Briefly, we have included a new figure (revised Figure 6, Fig. R1) to systemically integrate the Cut&Tag and ChIP-Seq

analyses, and hence clarify the transition between GLUP development (original Figure 5) and in vivo anti-tumor capacity (original Figure 6).

Figure R1. GLUP treatment inhibits gene transcription mainly via TEAD4-induced repressive condensates. (A) Representative live cell images of HGC-27 cells treated with FAM (a fluorescent moiety)-labeled GLUP for indicated periods of time. FAM-GLUP: 10 $\mu\text{g/ml}$ (5 μM). Scale bar, 10 μm . (B) Representative DAPI staining images of DNA organization and appearance in the presence or absence of TEAD4 and GLUP. GLUP: 1 $\mu\text{g/ml}$ (0.5 μM). Scale bar, 100 nm. (C) Representative N-SIM images of chromatin in GLUP-treated cells with DAPI staining. GLUP, 1 $\mu\text{g/ml}$ (0.5 μM). Scale bar, 100 nm. (D) Representative images showing the co-localization between GFP-TEAD4 and H3K27me3 in GLUP-induced HEK293FT cells treated with 1,6-Hex. Scale bar, 1 μm . (E) Average enrichment profiles of H3K27me3 in GLUP-treated WT or TEAD4 KO HGC-27 cells stratified by gene length. Normalization of coverage using RPKM was performed over the

genes and flanking 1 kb region. GLUP: 10 $\mu\text{g/ml}$ (5 μM). (F) Venn analysis of H3K27me3 and TEAD4 peaks reveals a group of co-existed peaks regardless of GLUP treatment. (G) Heatmap showing the chromatin association of H3K27me3 and TEAD4 onto the co-existed peaks identified in figure R1F with or without GLUP treatment. (H) IGV (Integrative Genomics Viewer) snapshot of H3K27me3 or TEAD4 Cut&Tag coverage upon indicated treatment. The interval scale is 10 in both cases.

1. Regarding points 10/15: I asked the authors to analyze TEAD4 and H3K27me3 (not H3K27ac). Even though K27ac and K27me3 targets the same lysine residue, their genomic localization is VERY different and cannot be used interchangeably => H3K27me3 marks whole gene bodies whereas h3K27ac marks promoters and active enhancers.

We understand this reviewer's concern about the genome-wide association of the H3K27me3 modification with GLUP treatment. In our last revision, we had applied ChIP-qPCR to demonstrate that either glucose starvation or VGLL4 or RFXANK overexpression markedly enhanced H3K27me3 signals at promoters of CTGF, CCNA2 and CGB5 (Figure R15 of the last revision, Figs. R2, A-C). Moreover, confocal imaging analysis also revealed that stronger H3K27me3 signals were incorporated into the TEAD4 condensates under glucose starvation or VGLL4/RFXANK overexpression conditions (Figs. R2, D-F). Overall, these pieces of evidence implied that TEAD4 LLPS could trigger development of a transcriptional repression status.

Figure R2. Chromatin immunoprecipitation-quantitative PCR (ChIP-qPCR) analysis of TEAD4-specific motifs such as *CTGF*, *CCNA2* and *CGB5* under glucose deprivation or in VGLL4/RFXANK-overexpressing HEK293FT cells and TEAD4 condensates were correlated with transcriptional repression. (A) ChIP-qPCR analysis of TEAD4-specific motifs such as *CTGF*, *CCNA2* and *CGB5* under glucose deprivation of HEK293FT cells. (B) ChIP-qPCR analysis of TEAD4-specific motifs such as *CTGF*, *CCNA2* and *CGB5* in VGLL4/RFXANK-overexpressing HEK293FT cells. (C) Western blotting to confirm the expression of VGLL4 or RFXANK in HEK293FT cells. Data represents three replicates from one experiment. (D) Fluorescent images of GFP-TEAD4 and H3K27me3 in glucose-deprived HEK293FT cells with or without glucose re-supplement. Scale bar, 10 μ m. (E) Representative images of the GFP-TEAD4 and H3K27me3 in VGLL4-overexpressing HEK293FT cells treated with 1,6-Hex. Scale bar, 10 μ m. (F) Representative images of the GFP-TEAD4 and H3K27me3 in RFXANK-overexpressing HEK293FT cells treated with 1,6-Hex. Scale bar, 10 μ m.

To further demonstrate that treatment with GLUP could similarly induce TEAD4 LLPS-dependent transcriptional repression, we have now performed a Cut&Tag assay against H3K27me3 in both WT and TEAD4-knockout HGC-27 cells treated with or without 5 μ M GLUP (3 replicates/group, **Fig. R3**). First, we calculated the FRiP value for assessing data quality, and these values indicated the CUT&Tag data to be of high quality (**Fig. R3**, FRiP value > 0.03). Principal component analysis (PCA) also showed clustered replicates (**Fig. R3B**),

validating our Cut&Tag data. More importantly, we observed that GLUP treatment dramatically promotes H3K27me3 signals at 3 kb regions flanking gene transcription start site (TSS) in control WT HGC-27 cells but not in TEAD4-depleted HGC-27 cells (**Fig. R3C**). Overall, these new results further validated that GLUP triggers a transcriptional repression signature mainly via TEAD4.

Next, we used Venn plot to specify the peaks shared by both TEAD4 and H3K27me3, which identified 761 and 235 co-existed peaks in control and GLUP groups, respectively (**Fig. R3D**). Among them, 169 peaks were found in both control and GLUP groups, allowing us to compare TEAD4 association with chromatin and H3K27me3 association with chromatin upon GLUP treatment (**Fig. R3D**). This analysis revealed a positive correlation between TEAD4 and H3K27me3 chromatin association, *i.e.*, the GLUP-induced TEAD4-binding peaks also have elevated H3K27me3 signals (**Fig. R3E**), suggesting the involvement of H3K27me3 in the repressive TEAD4 condensates. These results were further validated by confocal imaging (**Fig. R3F**). Of note, we also observed that GLUP treatment reduced TEAD4 chromatin association in a subgroup of co-existed peaks, suggesting that GLUP-induced TEAD4 condensates may sometimes strip TEAD4 off from chromatin (**Fig. R3E**). Nevertheless, the H3K27me3 signals were also decreased in these TEAD4-low peaks, further supporting the co-existence of TEAD4 and H3K27me3 signals.

Figure R3. Multiple analysis of H3K27me3 Cut&Tag data in WT or TEAD4 KO HGC-27 cells. (A) FRiP value for H3K27me3 Cut&Tag assay for triplicates with control or GLUP treatment in TEAD4 KO cells. (B) PCA analysis for H3K27me3 Cut&Tag assay with control or GLUP treatment in TEAD4 KO cells. (C) Normalization of coverage using RPKM was performed over the genes and flanking 3 kb region. (D) Venn analysis of H3K27me3 and TEAD4 ChIP-Seq common motifs upon GLUP treatment. (E) Heatmap for H3K27me3 or TEAD4-binding onto the common motifs. (F) Representative images showing the co-localization between GFP-TEAD4 and H3K27me3 in GLUP-induced HEK293FT cells treated with 1,6-Hex. Scale bar, 1 μ m.

The analysis R19G, H is very confusing for me: what does "TEAD4-specific motifs" mean. Is this a peak set containing TEAD motifs or are these real experimental TEAD4 peaks? I have two issues with

R19G,H: 1.) the analysis provided is not quantitative and statistics are missing. Since the analysis is not described, I cannot judge what I am looking at but the heatmap certainly contains only very few peaks and why is the number of peaks different between G and H. If the authors have defined TEAD4 peaks, the heatmaps should not differ in terms of peaks (which they clearly do) 2.) having looked at the bigwig files for the H3K27me3 ChIP-Seq: there is no induction of H3K27me3 at CTGF, CYR61 & Co which is not in line with their proposed mechanism.

We thank the reviewer for pointing out this issue. First, we defined "TEAD4-specific motifs" from the real experimental TEAD4 peaks and provided representative peaks for demonstration in the last rebuttal. As this reviewer suggested, we have now in an unbiased manner re-analyzed both the H3K27ac and H3K27me3 chromatin associations between TEAD4-specific peaks. Briefly, we used a Venn plot analysis to characterize the overlap of the peaks reproducibly identified using anti-TEAD4 and anti-H3K27ac or anti-TEAD4 and anti-H3K27me3 antibodies regardless of treatment with GLUP, and thus revealed 872 and 169 co-existing peaks (TEAD4-specific peaks) (**Figs. R4A and R4C**). We found that GLUP treatment dramatically reduced the binding of H3K27ac and increased the binding of H3K27me3 to the TEAD4-specific motifs (**Figs. R4B and R4D**).

In addition, we also used a Venn plot analysis to characterize the overlap of the peaks reproducibly identified by TEAD4, H3K27ac and H3K27me3 antibodies regardless of GLUP treatment, and identified 60 TEAD4-specific peaks. Moreover, we observed that GLUP treatment reduced H3K27ac-binding, but promoted H3K27me3-binding, to those TEAD4-specific peaks (**Figs. R4E and R4F**).

Meanwhile, we also looked at the bigwig files for the H3K27me3 ChIP-Seq data using Integrative Genomics Viewer (IGV), and found a little bit more but not significant coverage of H3K27me3 at CTGF or CYR61 upon GLUP treatment when compared to the control group (**Fig. R1H**). As the reviewer pointed out, it is possible that, under certain circumstances, GLUP may strip TEAD4 off the chromatin. Accordingly, we have now modified our interpretation of the data and the related wording in our revised manuscript.

Figure R4. Multiple analysis of TEAD4 ChIP-Seq data. (A) Venn analysis of H3K27ac and TEAD4 ChIP-Seq common motifs upon GLUP treatment. (B) Heatmap for H3K27ac or TEAD4-binding onto the common motifs in figure R5A. (C) Venn analysis of H3K27me3 and TEAD4 ChIP-Seq common motifs upon GLUP treatment. (D) Heatmap for H3K27me3 or TEAD4-binding onto the common motifs in figure R5C. (E) Venn analysis of H3K27ac, H3K27me3 and TEAD4 ChIP-Seq common motifs upon GLUP treatment. (F) Heatmap for H3K27ac and H3K27me3-binding onto the common motifs in figure R5E.

Specific response to point 15 (Figure R19):

Unfortunately, the authors did not perform the experiment in a TEAD4 KO cell line, and also do not comment on this in the text why they did not perform this essential experiment as requested.

Agreed. Following the reviewer's comment, we have now performed a Cut&Tag assay against H3K27me3 in both WT and TEAD4-knockout HGC-27 cells treated with or without 5 μ M GLUP (3 replicates/group, **Fig. R3**). Notably, we observed that the GLUP treatment dramatically promoted the H3K27me3 signals at 3 kb regions flanking gene transcription start sites (TSS) in WT HGC27 cells, whereas we failed to detect such promotion in TEAD4-depleted HGC27 cells (**Fig. R3C**). These new results further validated the idea that GLUP triggers transcriptional repression in a TEAD4-dependent manner.

Furthermore, it is crystal clear that GLUP leads to a complete loss of TEAD4 binding at CTGF in their ChIP-Seq analysis(R19B) and (having looked at the bw files uploaded to GEO) also in the new CUT&TAG analysis. This is at odds to the proposed mechanism of the whole paper. Thus, I have to question the conclusions about GLUP-mediated repression via TEAD4-dependent repressive LLPS, especially since this claim cannot be backed up by genome-wide H3K27me3 data. It seems more likely that GLUP leads to a loss of TEAD binding which can explain the effects on TEAD targets and secondary epigenomic changes.

The reviewer raised a very interesting point! After carefully checking the ChIP-Seq and Cut&Tag data related to TEAD4 ChIP, we indeed observed a subset of target genes with reduced TEAD4 binding upon treatment with GLUP (**Figs. R5A–B**). Thus, on the one hand, we can reproducibly observe that GLUP treatment had marginal effects on the binding of TEAD4 to some genes such as *ETS1* and *DARS2* (**Fig. R5C**); on the other hand, as this reviewer mentioned, GLUP reduced binding of TEAD4 to other genes such as *CTGF* (**Fig. R5A**). Moreover, we also used confocal microscopy to examine the co-localization of H3K27me3 with TEAD4, and found that GLUP treatment led to stronger sH3K27me3 signals in the TEAD4 LLPS droplets, suggesting that GLUP suppresses gene transcription at least

partially by inducing formation of TEAD4-H3K27me3 condensates (**Fig. R5D**). Taken together, GLUP may suppress gene transcription either through TEAD4-mediated DNA entangling or simply by stripping TEAD4 off chromatin.

Figure R5. GLUP leads to the loss of TEAD4 binding. (A) Integrative Genomics Viewer (IGV) snapshot depicting ChIP-Seq signal of the indicated genes in GLUP-treated cells. Signals are plotted on a normalized read per million (RPM) bases. (B) IGV showing the single peak for *CTGF*, *TEAD4* and *BCL2L1* in Cut&Tag assay using TEAD4 antibody. (C) IGV showing the single peak for *ETS1* and *DARS2*. (D) Representative images of the GFP-TEAD4 and H3K27me3 in GLUP-induced HEK293FT cells treated with 1,6-Hex. Scale bar, 1 μ m.

The CUT&TAG data tracks (R19E) are unfortunately not labelled, but I assume that this is TEAD4?

Yes, the previous Cut&Tag assay was performed using an anti-TEAD4 antibody. We thank the reviewer for pointing out this issue

Referee #2:

The authors took great effort in the revision, and have completed a lot of new experiments to address previous comments. All of my questions are addressed, and I am happy to support its publication.

We appreciate this reviewer's comments on the revision of our manuscript.

Referee #3:

The authors have added an impressive amount of work to address the comments of the four reviewers. My comments focused on the first half of the manuscript and in general they are sufficiently addressed by the new data. I now recommend this manuscript for publication in EMBO Journal.

Please note that the manuscript can benefit from textual editing, for example "As TEAD4 has been well-characterized in forming LLPS for" p7, should be "As TEAD4 has been well-characterized to form LLPS-driven condensates that...", etc.

We appreciate this reviewer's constructive comments on the revision of our manuscript. We have now carefully examined the wording in our manuscript.

Referee #4:

The authors have addressed my concerns. I recommend the publication of this manuscript.

We appreciate this reviewer's comments on the revision of our manuscript.

Dear Zhaocai,

Thank you for submitting a revised version of your manuscript. We have now received input from one of the original reviewers, who now finds that their previous concerns have been addressed satisfactorily. Therefore, there now remain only a number of editorial points that need addressing before I can extend official acceptance of the manuscript:

1. Please check the spelling of the authors' names in the manuscript and in our system; specifically, currently there are two spellings of the author's name Yanni/Yani Zhang, please check and correct.
2. We are missing the ORCID iD for the co-corresponding author Shi Jiao. In order to link the ORCID iD to the account in our manuscript tracking system, the author in question has to do the following:
 - Click the 'Modify Profile' link at the bottom of your homepage in our system.
 - On the next page you will see a box halfway down the page titled ORCID*. Below this box is red text reading 'To Register/Link to ORCID, click here'. Please follow that link: you will be taken to ORCID where you can log in to your account (or create an account if you don't have one)
 - You will then be asked to authorise Wiley to access your ORCID information. Once you have approved the linking, you will be brought back to our manuscript system.Unfortunately, we cannot do this linking on the author's behalf for security reasons.
3. Please check that the funding information is correct and identical both in the manuscript and our online system. Currently, grants 82222052, 2017YFA0504504, 19JC1415600, 22ZR1448100, 22QA1407200, 22QA1407300, 23ZR1480400 are missing in our online system.
4. CRediT has replaced the traditional author contributions section because it offers a systematic, machine-readable author contributions format that allows for more effective research assessment. Please remove the Authors Contributions from the manuscript and use the free text boxes beneath each contributing author's name in our online submission system to add specific details on the author's contribution. More information is available in our guide to authors.
5. Please rename "Declaration of interests" section into "Disclosure and competing interests statement" (further info: <https://www.embopress.org/page/journal/14602075/authorguide#conflictsofinterest>).
6. Please update references according to The EMBO Journal style - it should be alphabetically ordered; where there are more than 10 authors on a paper, the first 10 should be listed, followed by 'et al.' Please see further information here: <https://www.embopress.org/page/journal/14602075/authorguide#referencesformat>
7. Please rename the movies into Movie EV1-EV2 and update the callouts accordingly. The legends should be removed from the manuscript text file and zipped with each movie file. Further information is available here: <https://www.embopress.org/page/journal/14602075/authorguide#expandedview>
8. In the "Data availability" section, please add resolvable links to the datasets. More information about the format of this section can be found here: <https://www.embopress.org/page/journal/14602075/authorguide#dataavailability>.
9. Please move Appendix methods and references to the main manuscript.
10. In the final version of the Appendix, the text should be plain (currently textual changes are marked in red). Please also add a table of contents with page numbers to the final file.
11. Please rename Supplementary figures into Appendix Figure S1-etc.
12. Please add Table S4 and its legend to the Appendix file and rename it Appendix Table S1.
13. Please rename tables S1-3 into Dataset EV1- EV3 and upload as individual files. Each dataset will need a legend added to the file.
14. Please submit the completed source data checklist and all source data files as requested by our source data coordinator in her email dated 12 November 2023. Please note that the figure panel numbers refer to the initial version of the manuscript.
15. During our routine text plagiarism check, we noted that numerous sentences in the manuscript shows high similarity to these from other publications - please see the attached screenshots. Please rephrase the text accordingly.
16. Please remove "Statement of Significance" and "Highlights" from the manuscript text file and upload as a separate "Synopsis" file.
17. Please upload Supplementary figure 9 as a Synopsis image. The dimensions should be 550x300-600 pixels (width x height), jpeg or png format.
18. Our data editors have flagged the following issues in figure legends that need correcting:
 - Please define the box plot in terms of minima, maxima, centre, bounds of box and whiskers, and percentile in the legend of figure 7c.
 - Please add information on the nature and number of replicates in the legends of figures 1d, f-g; 2d, f-h; 3e, i-j; 4c-d, i-k; 5g-i; 6i; 7a, c-d, h.
 - Please describe the nature of replicates in the legends of figures 7f, i.
 - Please define the error bars in the legends of figures 1d, f-g; 2d, f-g; 3e; 4j; 5h-i; 7a, d, f.
 - Please define the white arrowheads in the legend of figure 1c.
 - Please define the white circles in the legends of figures 2e; 5j.
 - Please define the red circles and red arrowheads in the legend of figure 6e.
 - Please define the red circles in the legend of figure 7i.

Please let me know if you have any questions regarding any of these points. You can use the link below to upload the revised

files.

With best wishes,

Ieva

We realize that it is difficult to revise to a specific deadline. In the interest of protecting the conceptual advance provided by the work, we recommend a revision within 3 months (13th Nov 2024). Please discuss the revision progress ahead of this time with the editor if you require more time to complete the revisions.

Referee #1:

The authors have now addressed all my concerns, so I would recommend this story for publication.

The authors addressed the minor editorial issues.

Dear Dr. Zhou,

Thank you for addressing most of the final points. I sincerely apologise for the delay in communicating the decision due to the high number of submissions we receive at the moment and my absence from the office at the beginning of the month. I am now pleased to inform you that your manuscript has been accepted for publication.

Before we forward your manuscript to our publishers, there are a couple of points that remain to be addressed:

1. The nature of replicates (technical or biological) still needs to be indicated in figure legends, in particular for panels 1d, f-g; 2d, f-h; 3e, i-j; 4c-d, i-k; 5g-i; 6i; 7a, c-d, h-i.
2. I find that the article title needs to be clarified and made more specific for the broad audience of the journal. I have included three proposals below.

Title options:

A cofactor-induced repressive type of transcription factor condensation can be induced by synthetic peptides

OR:

Cofactor-dependent repressive TEAD4 condensation can be induced to suppress tumorigenesis

OR:

Induction of a new type of cofactor-dependent repressive TEAD4 condensation can suppress tumorigenesis

3. I would like to propose some edits in the manuscript abstract and synopsis (please also see the attached file). I have also written a short blurb that will accompany the title of your manuscript in our online system. Please let me know if any corrections are needed.

Blurb:

Glucose deprivation-dependent co-factor binding induces TEAD4 condensation, suppressing its transcriptional activity and inhibiting gastric tumorigenesis in mice.

Synopsis:

Liquid-liquid phase separation of transcription factors has been shown to activate transcription. This study identifies a repressive type of transcription factor condensation induced by TEAD4 co-factor binding, leading to DNA/chromatin aggregation.

- Binding of transcriptional co-factors VGLL4 and RFXANK induces formation of large condensates of TEAD4 in cells under glucose starvation.
- VGLL4-induced condensation of TEAD4 represses its YAP-dependent transcriptional activation.
- Repressive condensation of TEAD4 causes chromatin aggregation and cell death.
- A "glue peptide" derived from the TEAD4-binding motif of VGLL4 induces TEAD4 condensation and suppresses tumorigenesis in mice.

Finally, we would like to promote your manuscript among the Chinese readership. Therefore, we would like to invite you to prepare a short summary of the manuscript in Chinese (1500-2000 Chinese characters), which we will promote on the WeChat platform 'BioArt' with more than 610,000 followers.

If you are interested in this opportunity, we recommend covering the article very close to its online publication date. Thus, ideally we would very much appreciate if you could send us a draft within the next 7 working days. Please let us know whether or not you would be interested in contributing such a short summary in Chinese.

I have included below some general guidelines on how to prepare a summary and a link to recent examples for your reference. Please let me know if you have any questions about this.

If you have any questions, please do not hesitate to contact the Editorial Office. Thank you for this contribution to The EMBO Journal and congratulations on a nice study!

With best wishes,

Ieva

General WeChat Summary Guidelines

1. These summary articles are meant to be targeting general audience so please limit the use of specialized technical terms, acronyms and jargon.
2. A summary usually starts with brief background information of the reported work, which is followed by explaining the findings in some detail, and ends with a short review of the conclusions as well as the implications of the work and future directions for the research.
3. The summary should at least contain one graphical item, such as a scheme or a figure from the paper.
4. Please provide ONE SINGLE document containing all text and graphical materials, ideally as a Word.docx or .doc file. Please DO NOT provide the document as a .pdf file.
5. Please DO NOT publicly release the document before the paper is officially published online.

Summary Examples

EMBO J | 罗招庆/欧阳松应揭示谷酰胺脱氨酶MvcA的去泛素化功能

EMBO J | 王松灵院士团队揭示组织内应力调控大型哺乳动物乳恒牙替换的新机制
